

# A new phylogenetic analysis of Phytosauria (Archosauria: Pseudosuchia) with the application of continuous and geometric morphometric character coding

Andrew S. Jones and Richard J. Butler

School of Geography, Earth and Environmental Sciences, University of Birmingham, Birmingham, UK

Corresponding author
Andrew S. Jones,
andrew.jones.sp@gmail.com

## ABSTRACT

Phytosauria is a clade of large, carnivorous, semi-aquatic archosauromorphs which reached its peak diversity and an almost global distribution in the Late Triassic (*c.* 230–201 Mya). Previous phylogenetic analyses of Phytosauria have either focused primarily on the relationships of specific subclades, or were limited in taxonomic scope, and no taxonomically comprehensive dataset is currently available. We here present the most taxonomically comprehensive cladistic dataset of phytosaurs to date, based on extensive first-hand study, identification of novel characters and synthesis of previous matrices. This results in an almost twofold increase in phylogenetic information scored per taxon over previous analyses. Alongside a traditional discrete character matrix, three variant matrices were analysed in which selected characters were coded using continuous and landmarking methods, to more rigorously explore phytosaur relationships. Based on these four data matrices, four tree topologies were recovered. Relationships among non-leptosuchomorph phytosaurs are largely consistent between these four topologies, whereas those of more derived taxa are more variable. *Rutiodon carolinensis* consistently forms a sister relationship with *Angistorhinus*. In three topologies *Nicrosaurus* nests deeply within a group of traditionally non-Mystriosuchini taxa, leading us to redefine Mystriosuchini by excluding *Nicrosaurus* as an internal specifier. Two distinct patterns of relationships within Mystriosuchini are present in the four topologies, distinguished largely by the variable position of *Mystriosuchus*. In two topologies *Mystriosuchus* forms the most basal clade in Mystriosuchini, whilst in the others it occupies a highly derived position within the *Machaeroprosopus* clade. '*Redondasaurus*' is consistently recovered as monophyletic; however, it also nests within the *Machaeroprosopus* clade. The greatest impact on tree topology was associated with the incorporation of continuous data into our matrices, with landmark characters exerting a relatively modest influence. All topologies correlated significantly with stratigraphic range estimates. Topological variability in our results highlights clades in which further investigation may better elucidate phytosaur relationships.

## INTRODUCTION

Phytosaurs were a group of large-bodied archosauromorph reptiles that achieved an almost global distribution during the Late Triassic (*c.* 230–201 Mya; *Stocker & Butler, 2013*). In overall morphology, they are highly convergent with modern crocodilians, and this observation, in combination with the common recovery of their fossils from fluvial and lacustrine depositional environments, indicates that phytosaurs may have occupied a semi-aquatic niche, with their dentition suggestive of piscivory and carnivory (*Stocker & Butler, 2013*).

By far the most intensively investigated aspect of Phytosauria is their systematics. The phylogenetic position of phytosaurs within Archosauromorpha remains debated, having been recovered by recent analyses as either the sister group to Archosauria (*Nesbitt, 2011*), or as the earliest diverging clade within the crocodilian total-group Pseudosuchia (*Ezcurra, 2016*). Regardless of their exact phylogenetic position, time-calibration of phylogenies indicates that phytosaurs originated in the Early Triassic, soon after the Permo–Triassic mass extinction, although only one confirmed phytosaur taxon is known prior to the Late Triassic (*Stocker et al., 2017*). Their abundance, rich fossil record and cosmopolitan distribution indicate that phytosaurs were an important component of Late Triassic ecosystems; as a result, aspects of phytosaur palaeobiology such as ontogeny (*Irmis, 2007*) and neurosensory adaptions (*Holloway, Claeson & O'keefe, 2013*; *Lautenschlager & Butler, 2016*), as well as biogeography (*Buffetaut, 1993*; *Brusatte et al., 2013*; *Stocker & Butler, 2013*), have received considerable interest. Furthermore, phytosaurs have featured heavily in biostratigraphical hypotheses for the Late Triassic terrestrial record (*Long & Ballew, 1985*; *Parrish & Carpenter, 1986*; *Lucas & Hunt, 1993*; *Lucas, 2010*; *Martz & Parker, 2017*). An important factor for these analyses and others is a robust understanding of evolutionary relationships within Phytosauria. Phytosaur taxonomy has a long, problematic and convoluted history, adding considerable complication to later attempts at understanding phytosaur evolutionary history (*Hungerbühler, 2002*; *Stocker & Butler, 2013*). However, with the advent and continued improvement of cladistic techniques, a more cohesive picture has begun to form.

Most previous phylogenetic analyses of the ingroup relationships of Phytosauria have primarily focused on elucidating the relationships of individual or specific sets of taxa (Table 1). To achieve this, many analyses opted to reduce their taxonomic scope, and as such have greatly enhanced current knowledge of many areas in phytosaur systematics. However, there is currently no taxonomically comprehensive cladistic dataset which can be used to investigate relationships across all known phytosaur species and clades. The development of such a dataset is an essential prerequisite for carrying out broader evolutionary analyses. To address this gap, this paper has three primary aims:

1. To present the most taxonomically comprehensive phylogeny of Phytosauria to date, including nearly all currently recognized species;
2. To use this phylogeny to investigate the phylogenetic relationships of a number of species and higher-level taxa that have previously been recognized as problematic;

3. To assess the utility of continuous and geometric morphometric (GM) character coding techniques, as tools that can potentially expand the information available to assess phytosaur interrelationships.

## Previous work

### Previous cladistic analyses

The first cladistic analysis of the ingroup relationships of Phytosauria was performed by *Ballew (1989)*. Her analysis included 11 operational taxonomic units (OTUs) and 64 characters with the aim of establishing character polarity and revising the diagnoses and species assignments of the genera *Rutiodon* and '*Pseudopalatus*'. The analysis generated a tree topology which, in its general structure, has changed relatively little in subsequent analyses. '*Paleorhinus*' and *Angistorhinus* were recovered at the base of Phytosauria, and a polytomy of taxa which Ballew synonymized into *Rutiodon* was recovered as the sister taxon to a clade consisting of *Nicrosaurus*, '*Pseudopalatus*' and *Mystriosuchus* (Fig. 1A).

Ballew's phylogeny (Fig. 1A) was used as a basis for *Long & Murry (1995)* to present a comprehensive taxonomic review of Phytosauria, including the erection of three new genera ('*Arganarhinus*', *Smilosuchus*, '*Arribasuchus*') and the identification of numerous new anatomical characters with potential taxonomic or phylogenetic significance. No numerical phylogenetic analysis or phylogenetic tree was presented, but based on the identification of novel characters a taxonomy was constructed, differing from the phylogeny of *Ballew (1989)* most importantly in the separation of the taxa included in *Rutiodon* by Ballew into *Leptosuchus Case, 1922* and the new genus *Smilosuchus*, and in the basal position of *Mystriosuchus* as the sister taxon to '*Paleorhinus*' (previously suggested by *Gregory (1962a)* and *Hunt & Lucas (1989)*).

*Hungerbühler (1998)* increased taxonomic sampling, including 22 species-level OTUs, and presented a largely novel matrix of 49 characters, of which 12 were based on or reused from previous studies (*Ballew, 1989*; *Long & Murry, 1995*). The aims were twofold: to test the concept of a monophyletic '*Paleorhinus*' (*Ballew, 1989*; *Hunt & Lucas, 1991*; *Long & Murry, 1995*), and to more thoroughly assess the phylogenetic position of *Mystriosuchus*. '*Paleorhinus*' was found to be paraphyletic, with the species previously assigned to the genus recovered as a grade of iteratively more derived taxa at the base of Phytosauria. In agreement with *Ballew (1989*; Fig. 1A*)*, *Mystriosuchus* was found in a more derived position than '*Paleorhinus*', but nested as the sister taxon to '*Pseudopalatus*' rather than within this genus (Fig. 1B).

A substantially revised version of *Hungerbühler's (1998)* matrix was used by *Hungerbühler (2002)* to further investigate the relationships of *Mystriosuchus* and assess the phylogenetic position of the newly described species *Mystriosuchus westphali*. Sampling was reduced to 11 taxa and 47 characters (16 taken from the previous study), to focus the analysis on the clade formed of *Nicrosaurus*, *Mystriosuchus* and '*Pseudopalatus*', named 'Pseudopalatinae' by *Long & Murry (1995)*. *Mystriosuchus* was again recovered as the sister taxon to '*Pseudopalatus*'; additionally, the genus '*Redondasaurus*' was found to be monophyletic and outside of '*Pseudopalatus*', contra

**Table 1 Details of all previous cladistic studies of the ingroup relationships of Phytosauria.**

| | Phytosauria OTUs | Characters | Notes on matrix | Purpose of analysis |
|---|---|---|---|---|
| Ballew (1989) | 11 | 64 (39 autapomorphic, five for missing clade) | Novel matrix | First attempt to resolve the ingroup taxonomic relationships of Phytosauria using cladistic methods. |
| Hungerbühler (1998) | 22 | 49 | Novel matrix. Characters and scorings based on first-hand study only of European taxa; others based on literature. | Tests the proposed monophyly of 'Paleorhinus' and clarifies the position of Mystriosuchus. |
| Hungerbühler (2002) | 10 | 47 | Heavily revised matrix of Hungerbühler (1998). All scorings based on first-hand study. | Assesses the taxonomic position of Mystriosuchus generally, and specifically the newly named species M. westphali. |
| Parker & Irmis (2006) | 11 | 47 | Matrix of Hungerbühler (2002), plus Machaeroprosopus jablonskiae. | Establishes the taxonomic position of the newly described species M. jablonskiae. |
| Stocker (2010) | 19 | 43 | Novel matrix. | Clarifies the interrelationships of Leptosuchus and previously associated taxa, and finds the position of the newly described Pravusuchus hortus. |
| Stocker (2012) | 19 | 43 | Matrix of Stocker (2010). | Describes and taxonomically places Protome batalaria. |
| Stocker (2013) | 19 | 43 | Matrix of Stocker (2010). | Identifies and describes Wannia scurriensis as the most basal phytosaur, and discusses the paraphyly of 'Paleorhinus'. |
| Hungerbühler et al. (2013) | 12 | 41 | Novel matrix. | Assesses the interrelationships of Machaeroprosopus and 'Redondasaurus', and provides a description and taxonomic placement for M. lottorum. |
| Butler et al. (2014) | 22 | 46 | Matrix of Stocker (2010) plus Ebrachosuchus neukami, Parasuchus angustifrons & Machaeroprosopus jablonskiae, and three additional characters. | Redescribes Ebrachosuchus neukami and 'Francosuchus' angustifrons; tests and establishes a new monophyletic definition of 'Paleorhinus'. |
| Kammerer et al. (2015) | 24 | 48 | Matrix of Butler et al. (2014) plus Parasuchus hislopi, Leptosuchus imperfecta, and two additional characters. | Redescribes Parasuchus hislopi, demonstrates the seniority of the latter genus over 'Paleorhinus', and overhauls the names of phytosaur sub-family groups. |

Hungerbühler, Chatterjee & Cunningham (2003), but closer to the latter taxon than to Mystriosuchus. Nicrosaurus was recovered as the sister taxon of the Mystriosuchus + ('Redondasaurus' + 'Pseudopalatus') clade (Fig. 1C).

The matrix of Hungerbühler (2002) was subsequently used to test the phylogenetic position of 'Pseudopalatus' jablonskiae by Parker & Irmis (2006). This taxon was the only addition to the matrix and was found to occupy the most basal position in the genus 'Pseudopalatus', with no other changes in tree topology (Fig. 2A).

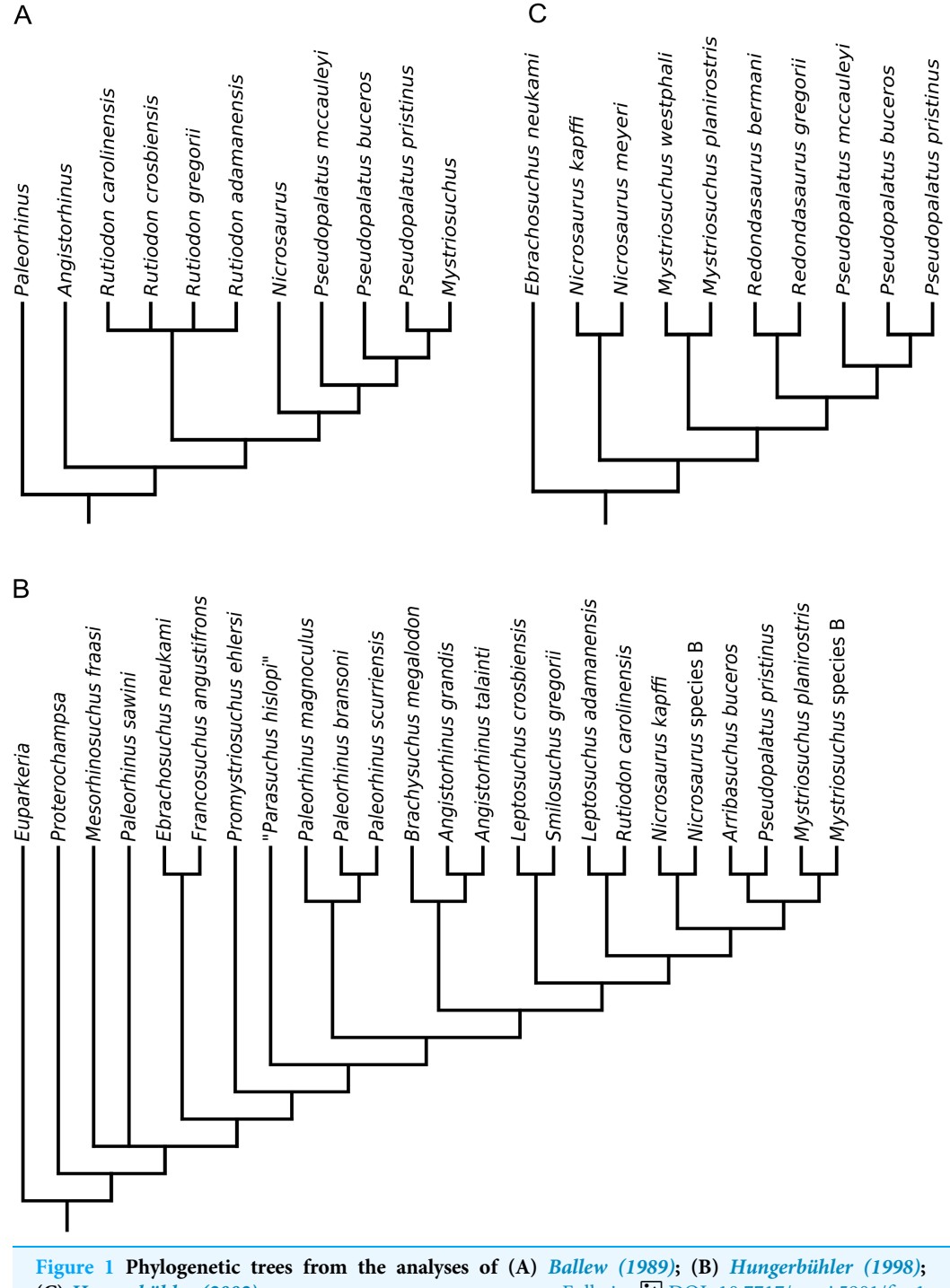

**Figure 1 Phylogenetic trees from the analyses of (A)** *Ballew (1989)*; **(B)** *Hungerbühler (1998)*; **(C)** *Hungerbühler (2002)*.

In order to better resolve the relationships of the stratigraphically important genus *Leptosuchus* (*Camp, 1930*; *Hunt & Lucas, 1991*; *Lucas, 2010*) and other associated taxa (including those that were synonymized into *Rutiodon* by *Ballew, 1989*), *Stocker (2010)* produced a largely novel matrix, incorporating three characters from the matrix of *Sereno (1991)*, and 18 either taken or modified from *Hungerbühler (2002)*. The full matrix

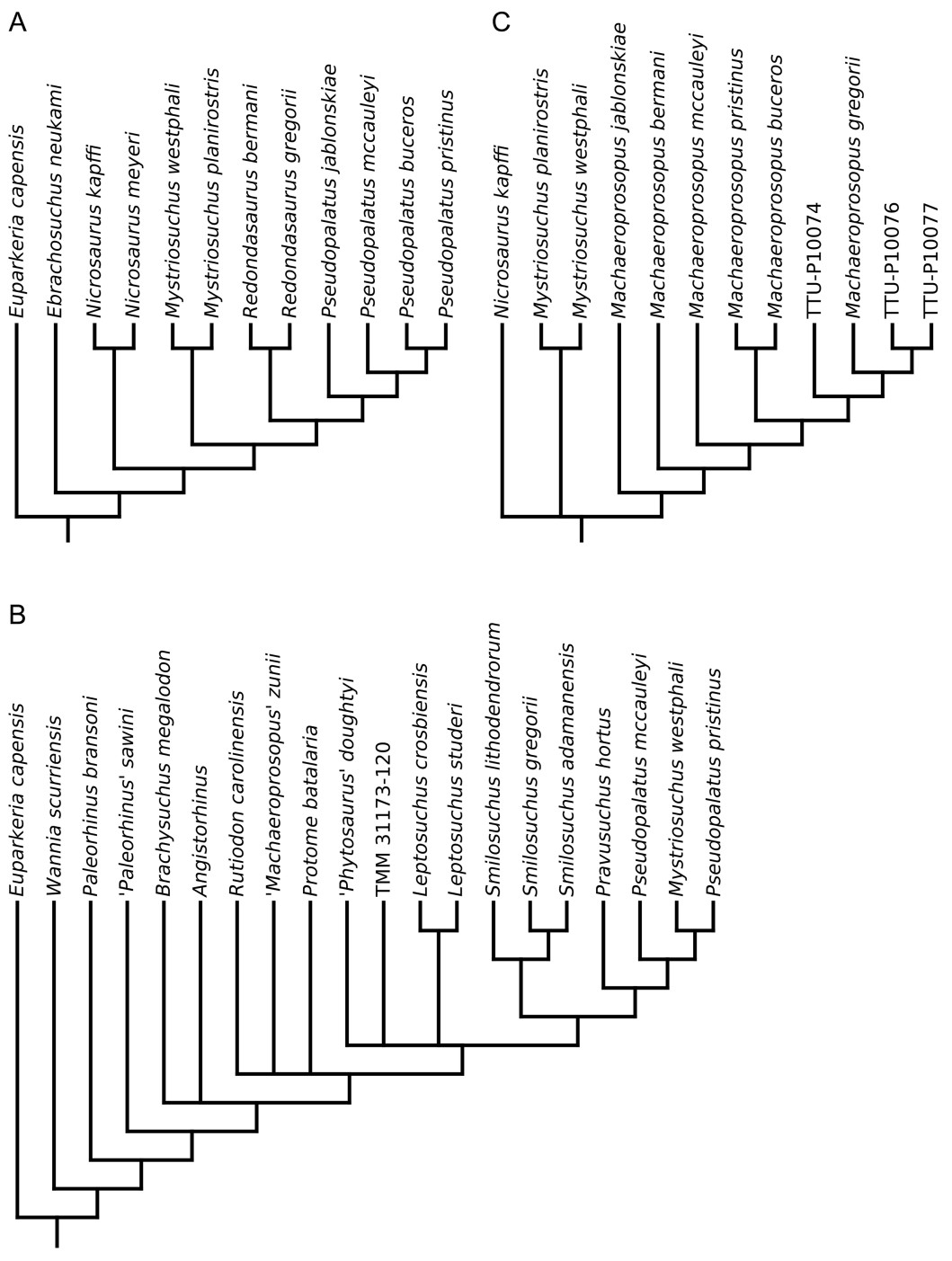

Figure 2 Phylogenetic trees from the analyses of (A) *Parker & Irmis (2006)*; (B) *Stocker (2013)* (topology identical to *Stocker, 2010, 2012*); (C) *Hungerbühler et al. (2013)*.

consisted of 43 characters scored for 24 OTUs and found *Leptosuchus* to be polyphyletic, with '*Leptosuchus*' *adamanensis* forming a monophyletic group with *Smilosuchus gregorii* and '*Machaeroprosopus*' *lithodendrorum* (Fig. 2B). As a result, '*Leptosuchus*' *adamanensis* and '*Machaeroprosopus*' *lithodendrorum* were reassigned to the genus

*Smilosuchus*. *Rutiodon* was not found to be synonymous with *Angistorhinus*, *Brachysuchus* or *Leptosuchus*, contra *Ballew (1989)*, *Long & Murry (1995)* and *Hungerbühler & Sues (2001)*. The new genus and species *Pravusuchus hortus* was recovered as the sister taxon to 'Pseudopalatinae', and '*Paleorhinus*' *scurriensis Langston, 1949* was found to occupy the most basal position within Phytosauria (Fig. 2B).

Following this, *Stocker (2012, 2013)* presented two further studies in which she first described the new taxon *Protome batalaria* and then redescribed '*Paleorhinus*' *scurriensis*, assigning the latter to the new genus *Wannia*. Phylogenetic aspects of both studies were based on the dataset of *Stocker (2010)* with no changes or additions. In the latter study, *Stocker (2013)* provided further discussion questioning the existence of a monophyletic '*Paleorhinus*', supporting the findings of *Hungerbühler (1998*; Fig. 1B).

Although not a phylogenetic study, an important taxonomic alteration was made by *Parker, Hungerbühler & Martz (2012)*. The genus name *Machaeroprosopus* was previously considered invalid because the sole specimen of its presumed type species (*Machaeroprosopus validus*, UW 3807) has been lost (*Gregory, 1962a*); however, *Parker, Hungerbühler & Martz (2012)* established that the holotype specimen of the species *Machaeroprosopus buceros* actually takes priority. The species *Machaeroprosopus buceros* was initially assigned to the genus '*Belodon*', but subsequently made the type species of the genus *Metarhinus* (*Jaekel, 1910*); however, when this genus was found to be preoccupied, a replacement genus, *Machaeroprosopus*, was erected by *Mehl (1915)*. Inexplicably, the species *Machaeroprosopus validus* was long used as the genotype of *Machaeroprosopus* despite *Machaeroprosopus buceros* having priority. As the holotype specimen of *Machaeroprosopus buceros* is readily available to study, the genus *Machaeroprosopus* was considered valid by *Parker, Hungerbühler & Martz (2012)*, with the type species being *Machaeroprosopus buceros*. Furthermore, *Machaeroprosopus buceros* has been recovered frequently as the sister taxon to '*Pseudopalatus*' *pristinus*, the type species of '*Pseudopalatus*', and has taxonomic priority over that species. As a result, all of the species previously assigned to '*Pseudopalatus*' were reassigned to *Machaeroprosopus* by *Parker, Hungerbühler & Martz (2012)*. The clade 'Pseudopalatinae' was, however, retained, as its usage lies outside of the remit of the ICZN, although it has subsequently been replaced by Mystriosuchini (see below, but see *Martz & Parker, 2017*).

The monophyly of the newly diagnosed *Machaeroprosopus* with respect to '*Redondasaurus*' was tested by *Hungerbühler et al. (2013)*; the two species of '*Redondasaurus*' were previously found to nest paraphyletically within *Machaeroprosopus* (*Hungerbühler, Chatterjee & Cunningham, 2003*). The primary purpose of the analysis was, however, to test the phylogenetic position of the newly described species *Machaeroprosopus lottorum*. Taxonomic sampling was restricted to 12 OTUs, focussing on the group 'Pseudopalatinae', and 41 characters of which 21 were to some extent based on characters from previous studies (*Hungerbühler, 1998, 2002*; *Stocker, 2010*). '*Redondasaurus*' was found to be paraphyletic and nest within *Machaeroprosopus* (Fig. 2C), contra *Hungerbühler (2002*; Fig. 1C) and *Parker & Irmis (2006*; Fig. 2A). *Machaeroprosopus lottorum* was also found to nest within *Machaeroprosopus*, bridging the

gap between the more derived species and specimens previously referred to 'Redondasaurus' and the specimens traditionally belonging to *Machaeroprosopus*.

Finally, two further studies were carried out based on the matrix of *Stocker (2010, 2012, 2013)*, both with the aim of redescribing basal phytosaur taxa previously assigned to 'Paleorhinus' and elucidating the relationships of basal phytosaurs. *Butler et al. (2014)* redescribed the taxa 'Paleorhinus' angustifrons (*Kuhn, 1936*) (formerly 'Francosuchus') and *Ebrachosuchus neukami Kuhn, 1936*, and established a robust set of synapomorphies (which were incorporated into the phylogenetic data matrix) to diagnose a revised, restricted definition of 'Paleorhinus' that included the species 'Paleorhinus' bransoni and 'Paleorhinus' angustifrons (Fig. 3A).

*Kammerer et al. (2015)* produced a redescription of *Parasuchus hislopi Lydekker, 1885* and found it to be the sister taxon to 'Paleorhinus' angustifrons, supported by two unambiguous synapomorphies. Given the designation by the ICZN of a neotype for *Parasuchus* (*Chatterjee, 2001*; *ICZN, 2003*), this genus takes priority over 'Paleorhinus' as the senior synonym. As a result, all species in the monophyletic 'Paleorhinus' group were reassigned to the genus *Parasuchus* (Fig. 3B). *Kammerer et al. (2015)* also presented an update to phytosaur family-level and subfamily groups, including the following groups, from most inclusive to most exclusive: Parasuchidae *Lydekker, 1885*, Mystriosuchinae *von Huene, 1915* (formerly Phytosauridae *Jaeger, 1828*), Leptosuchomorpha *Stocker, 2010* and Mystriosuchini *von Huene, 1915* (formerly 'Pseudopalatinae' *Long & Murry, 1995* (sensu *Parker & Irmis, 2006*)). For consistency, the nomenclature used by *Kammerer et al. (2015)* is used henceforth throughout this study, with some minor modification to phylogenetic definitions (Table 2; see below).

### Current consensus

Following the revision conducted by *Kammerer et al. (2015)*, phytosaurs are currently considered to fall into five successively less inclusive groups: Phytosauria, Parasuchidae, Mystriosuchinae, Leptosuchomorpha and Mystriosuchini (Table 2).

Phytosauria *Jaeger, 1828*, is a stem-based clade which encompasses all phytosaurs. Previously the membership of the groups Phytosauria and Parasuchidae overlapped completely (*Kammerer et al., 2015*); however, since the re-evaluation of *Diandongosuchus* (*Stocker et al., 2017*) this taxon has been included within Phytosauria, but excluded from Parasuchidae. However, this placement remains untested in any analysis of ingroup phylogeny to date.

Parasuchidae *Lydekker, 1885* (*Chatterjee, 1978*; *Kammerer et al., 2015*) contains the basal genera *Parasuchus*, *Ebrachosuchus* and *Wannia*, plus all phytosaurs belonging to Mystriosuchinae, Leptosuchomorpha and Mystriosuchini. Following the work of *Stocker (2013)*, *Wannia* has consistently been recovered as the most basal phytosaur within Parasuchidae (Fig. 2B), being distinct from the more derived *Parasuchus* clade defined by *Butler et al. (2014)* and *Kammerer et al. (2015)*. The latter two studies also recovered *Ebrachosuchus* in a more derived position than *Parasuchus* (Figs. 3A and 3B).

Mystriosuchinae *von Huene, 1915* excludes basal phytosaurs, being defined as 'the last common ancestor of *Mystriosuchus planirostris* (*Von Meyer, 1863*) and *Angistorhinus*

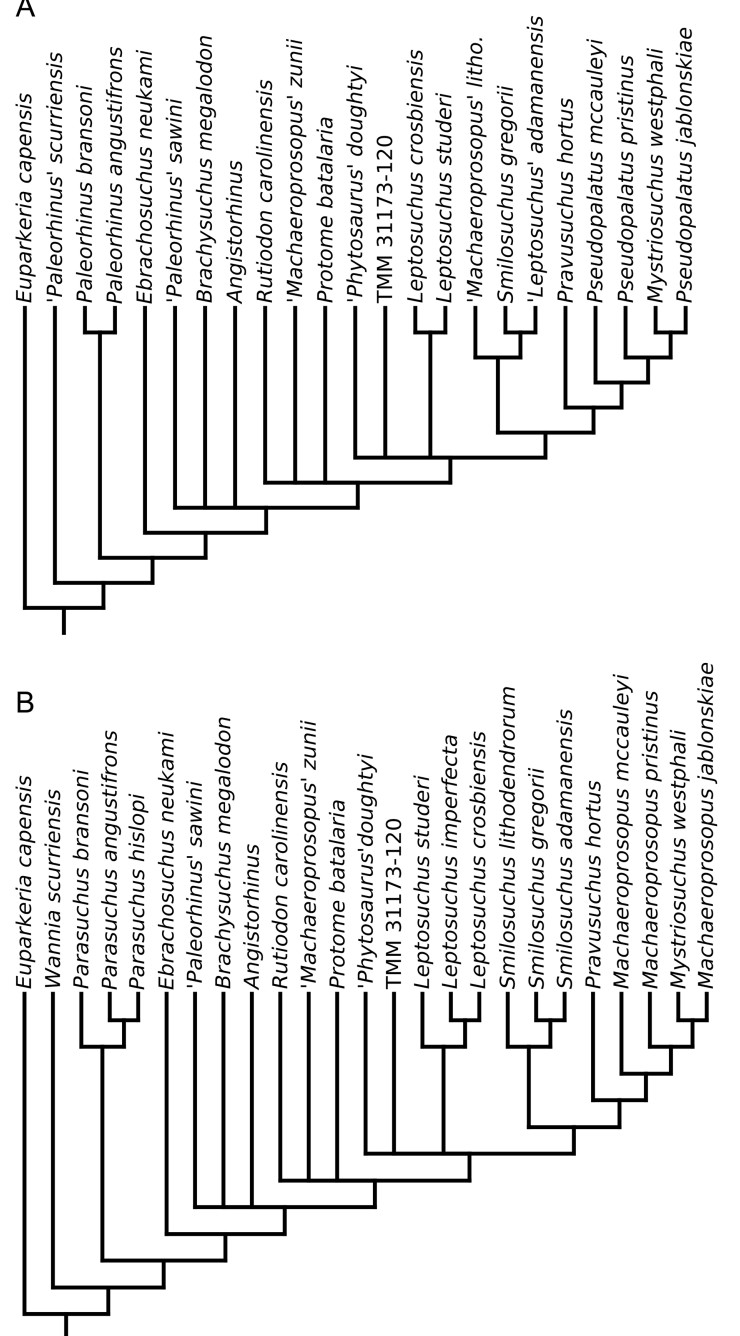

**Figure 3 Phylogenetic trees from the analyses of (A) *Butler et al. (2014)*; (B) *Kammerer et al. (2015)*.**

*grandis Mehl, 1913* and all of its descendants' (*Kammerer et al., 2015*), and is largely equivalent to Phytosauridae of previous analyses. In addition to Leptosuchomorpha and Mystriosuchini, this group may also contain taxa previously synonymized with '*Paleorhinus*', such as '*Paleorhinus*' *sawini*, and other genera, including *Rutiodon*, *Angistorhinus*, *Brachysuchus* and *Protome*. The relationships between *Angistorhinus*,

**Table 2 Higher-level taxonomic changes to family and sub-family group definitions.**

| Stocker & Butler (2013) | Kammerer et al. (2015) | Present study |
|---|---|---|
| Phytosauria *Jaeger, 1828* (stem): *Rutiodon carolinensis* and all taxa more closely related to it than *Aetosaurus ferratus*, *Rauisuchus tiradentes*, *Prestosuchus chiniquensis*, *Ornithosuchus woodwardi* or *Crocodylus niloticus* | Phytosauria *Jaeger, 1828* (stem): unchanged | Phytosauria *Jaeger, 1828* (stem): unchanged |
| (Unnamed node) | Parasuchidae *Lydekker, 1885* (node): *Wannia scurriensis*, *Parasuchus hislopi*, *Mystriosuchus planirostris* and all descendants of their most recent common ancestor | Parasuchidae *Lydekker, 1885* (node): unchanged |
| Phytosauridae *Jaeger, 1828* (node): *Angistorhinus*, *Leptosuchus studeri*, *Mystriosuchus westphali* and all descendents of their most recent common ancestor | Mystriosuchinae *von Huene, 1915* (node): *Mystriosuchus planirostris*, *Angistorhinus grandis* and all descendants of their most recent common ancestor | Mystriosuchinae *von Huene, 1915* (node): unchanged |
| Leptosuchomorpha *Stocker, 2010* (node): *Leptosuchus studeri*, *Machaeroprosopus pristinus* and all descendants of their most recent common ancestor | Leptosuchomorpha *Stocker, 2010* (node): unchanged | Leptosuchomorpha *Stocker, 2010* (node): *Smilosuchus lithodendrorum*, *Leptosuchus studeri*, *Machaeroprosopus pristinus* and all descendents of their most recent common ancestor |
| Pseudopalatinae *Long & Murry, 1995* (node): *Nicrosaurus kapffi*, *Mystriosuchus westphali*, *Machaeroprosopus pristinus*, *Redondasaurus gregorii* and all descendants of their most recent common ancestor | Mystriosuchini *von Huene, 1915* (node): *Nicrosaurus kapffi*, *Mystriosuchus planirostris*, *Machaeroprosopus buceros* and all descendants of their most recent common ancestor | Mystriosuchini *von Huene, 1915* (node): *Mystriosuchus planirostris*, *Machaeroprosopus jablonskiae*, *Machaeroprosopus buceros* and all descendents of their most recent common ancestor |

**Note:**
Included are the two most recent revisions of Phytosauria (*Stocker & Butler, 2013*; *Kammerer et al., 2015*) and the present study.

*Brachysuchus* and 'Paleorhinus' sawini are unresolved, but all of these taxa have been recovered as more derived than *Parasuchus* and basal to *Rutiodon* and *Protome*, with the latter two taxa being placed in a polytomy together with Leptosuchomorpha (Figs. 2B, 3A and 3B).

Leptosuchomorpha *Stocker, 2010*, was previously defined as 'the most recent common ancestor of *Leptosuchus studeri* and *Machaeroprosopus pristinus* and all descendants thereof'. We introduce a slight modification to this definition here (Table 2) in response to our phylogenetic results, and include 'Smilosuchus' lithodendrorum as an additional internal specifier to ensure that minor topological rearrangements between taxa that have consistently been considered as leptosuchomorphs do not jeopardize the stability of the clade. Therefore, in addition to members of Mystriosuchini, Leptosuchomorpha contains all species of *Leptosuchus* and *Smilosuchus*, as well as probably the taxa 'Phytosaurus' doughtyi and *Pravusuchus hortus*. *Leptosuchus* has been supported as monophyletic by recent analyses, though its possible relationship with 'Phytosaurus' doughtyi is unresolved. *Smilosuchus* has also been supported as monophyletic, and recovered as the sister taxon to *Pravusuchus* + Mystriosuchini.

Mystriosuchini *von Huene, 1915*, excludes all but the most derived phytosaurs, and was defined by *Kammerer et al. (2015)* as 'the last common ancestor of *Mystriosuchus*

*planirostris* (*Von Meyer, 1863*), *Nicrosaurus kapffi* (*Von Meyer, 1860*) and *Machaeroprosopus buceros* (*Cope, 1881*) and all of its descendants'. We modify this definition here by excluding *Nicrosaurus kapfii* from the list of internal specifiers and introducing *Machaeroprosopus jablonskiae* as a replacement to maximize the taxonomic stability of Mystriosuchini among the trees recovered here (Table 2; see below). Mystriosuchini is largely synonymous with 'Pseudopalatinae' *Long & Murry (1995)*, defined phylogenetically by *Parker & Irmis (2006)*, with the exception of the inclusion of *Mystriosuchus* and the possible exclusion of *Nicrosaurus*. Although a basal position of *Mystriosuchus* within Phytosauria, such as positioned as the sister taxon to '*Paleorhinus*', has been suggested in multiple studies (*Gregory, 1962a*; *Hunt & Lucas, 1989*; *Long & Murry, 1995*), this hypothesis has not been supported by quantitative cladistic analyses. A derived position for *Mystriosuchus* within Mystriosuchini has been found in all cladistic analyses thus far (*Ballew, 1989*; *Hungerbühler, 1998*, *2002*; *Parker & Irmis, 2006*; *Stocker, 2010*, *2012*, *2013*; *Hungerbühler et al., 2013*; *Butler et al., 2014*; *Kammerer et al., 2015*), and therefore seems relatively uncontroversial. The position of *Mystriosuchus* with respect to other taxa in Mystriosuchini is less well resolved, as discussed below. The European genus *Nicrosaurus* has been included within Mystriosuchini (*Long & Murry, 1995*; *Parker & Irmis, 2006*; *Kammerer et al., 2015*); however, the validity of this is also discussed below. The remainder of Mystriosuchini consists of species referred to *Machaeroprosopus* and '*Redondasaurus*', the relationships of which also differ between studies.

### Current uncertainties

Although *Rutiodon* has been consistently found close to, but in a more derived position than, *Angistorhinus*, this relationship has been tested in only three relatively independent matrices (*Ballew, 1989*; *Hungerbühler, 1998*; *Stocker, 2010*), of which the two earliest contain potential problems, including the use of parsimony uninformative characters, and the outgroup taxon representing homoplastic, rather than ancestral morphology. It has previously been suggested that *Angistorhinus* and *Rutiodon* may be synonymous (*Hungerbühler & Sues, 2001*), although this has never been explicitly tested or fully published.

Aside from the study of *Hungerbühler (1998)*, *Angistorhinus* has only been used as a generic-level OTU, or represented by a single species (*Kammerer et al., 2015*). *Kammerer et al. (2015)* used *Angistorhinus grandis* to score the genus; however, no further discussion of relationships within the genus was presented. The systematics of the genus *Angistorhinus* are another important area which is currently poorly understood within phytosaurs.

*Nicrosaurus kapffi* is generally accepted as the most basal member of Mystriosuchini, and was used as a reference taxon in the previous phylogenetic definition of the group (*Kammerer et al., 2015*; Table 2); however, only the early studies of *Ballew (1989)* and *Hungerbühler (1998)* have tested this position. *Nicrosaurus* has been included in two other relatively independent analyses (*Hungerbühler, 2002*; *Hungerbühler et al., 2013*); however, the aims of these studies did not necessitate the inclusion of taxa from outside of

Mystriosuchini, and therefore the position of the genus within global phytosaur phylogeny was not tested. Therefore, although the position of *Nicrosaurus* has not been contested, it is also not especially well supported by available data.

The position of *Mystriosuchus* within Mystriosuchini remains unclear, having been placed as either sister to the clade of *Machaeroprosopus* + 'Redondasaurus' (*Hungerbühler, 1998*, *2002*; *Parker & Irmis, 2006*; *Hungerbühler et al., 2013*) or nested within *Machaeroprosopus* (*Ballew, 1989*; *Stocker, 2010*; *Butler et al., 2014*; *Kammerer et al., 2015*). As the genus name *Mystriosuchus Fraas, 1896* has priority over *Machaeroprosopus Mehl, 1915*, this later relationship may have extensive taxonomic implications.

In multiple studies 'Redondasaurus' has been found to nest within *Machaeroprosopus* (*Ballew, 1989*; *Hungerbühler, Chatterjee & Cunningham, 2003*; *Stocker, 2010*; *Hungerbühler et al., 2013*; *Butler et al., 2014*; *Kammerer et al., 2015*), whereas in others 'Redondasaurus' is monophyletic to the exclusion of *Machaeroprosopus* (*Hungerbühler, 1998*, *2002*; *Parker & Irmis, 2006*). In the most recent phylogeny of derived phytosaurs (*Hungerbühler et al., 2013*), 'Redondasaurus' was found to nest within *Machaeroprosopus* and the two were tentatively synonymized, but this hypothesis requires further testing.

## MATERIALS AND METHODS

### Material

The analysis presented here uses species as OTUs to facilitate comparison with previous phylogenetic analyses. There has been recent interest in specimen-level phylogenetic analyses in vertebrate palaeontology (*Upchurch, Tomida & Barrett, 2004*; *Tschopp, Mateus & Benson, 2015*), but the validity of this approach and its results remain largely unexplored. We did not use a specimen-level phylogeny here as it would be hampered by the range of intraspecific variation found in most taxa, and would be further compounded by poor preservation in many specimens resulting in high quantities of missing data and widespread polytomies due to unstable terminals.

The OTUs included in this analysis consist of 34 species across 18 genera which are fully detailed in Appendix 1. An additional nine specimen-level OTUs were also included to test their affinities. We attempted to sample all phytosaur species currently regarded as taxonomically valid or potentially taxonomically valid, with the exception of a number of problematic species that were excluded for reasons discussed below. *Euparkeria capensis* was used to root the analysis as it displays a generalized archosauriform cranial morphology (*Sookias, 2016*) which has been used in previous studies for character polarization (*Hungerbühler, 2002*; *Parker & Irmis, 2006*; *Stocker, 2010*, *2012*, *2013*; *Butler et al., 2014*; *Kammerer et al., 2015*). *Diandongosuchus fuyuanensis*, a taxon from the Middle Triassic of China initially identified as a basal poposauroid (*Li et al., 2012*), was recently re-interpreted as the basal-most phytosaur currently known (*Stocker et al., 2017*) and is therefore included in this analysis to verify its basal position within Phytosauria.

Of the 43 OTUs included in this analysis, 39 were scored based on first-hand study of at least one of the referred specimens. Photographs and published descriptions

and figures were also used where available. The remaining four terminals (*Leptosuchus studeri*, *Diandongosuchus fuyuanensis*, *Euparkeria capensis* and *Parasuchus hislopi*) were not studied first hand for the purposes of this study, and were scored from photographs and/or published descriptions and figures.

## Excluded taxa

Although this analysis was designed to be the most comprehensive cladistic dataset for phytosaurs to date, a small number of taxa were excluded for various reasons.

*Angistorhinus gracilis Mehl, 1915*, from the Popo Agie Formation in Wyoming, was only very briefly described in the original paper, and a holotype was not formally designated, despite apparently consisting of a large skull and much of the postcrania of a single phytosaur. When ASJ visited the University of Missouri this material could not be found; however, it may be located in one of many footlockers containing the 'Mehl collection' in the basement of the department (James Schiffbauer, personal communication to Andrew S. Jones, 2016). At present this material is considered lost with no images available other than a line drawing of the antorbital region and two photographs of an anterior thoracic vertebra (*Mehl, 1915*); because the proportion of missing data would likely hinder any analysis more than its inclusion would contribute, we excluded this taxon.

*Angistorhinus maximus Mehl, 1928* is known from the orbital and postorbital portions of a single skull (MU 531) from the top of the Popo Agie Formation in Wyoming. *Long & Murry (1995)* noted apparent similarities between this species and *Angistorhinus talainti* from Morocco, but also suggested this material may represent a more derived taxon, not referable to *Angistorhinus*. They noted that determining the taxonomic affinities would require detailed study and the type material 'may be lost' (*Long & Murry, 1995*:42). This material is also suspected to reside in the 'Mehl collection' of the University of Missouri. As this material is considered lost and no images exist aside from the five line drawings in *Mehl (1928)*, it was excluded from analysis.

*Angistorhinus alticephalus Stovall & Wharton, 1936* is represented by an incomplete skull, nine vertebrae, rib fragments and osteoderms (OMNH 733) from the Dockum Group of Texas. This species is differentiated from other *Angistorhinus* species primarily by the more laterally directed orbits, the shape of the squamosal and the straight mediolateral frontal-parietal suture (*Stovall & Wharton, 1936*). It has been suggested that the direction of the orbits should be used cautiously due to taphonomic distortion (*Gregory, 1962a*; *Hungerbühler, 1998*) and is 'severely restricted' in practical use due to the difficulty in taking measurements and previous scoring subjectivity (*Hungerbühler, 1998*:130); therefore, a more detailed taxonomic analysis of this specimen is required to verify its distinctness, which is beyond the scope of this study. Given the incomplete nature of the type material, the range of better *Angistorhinus* material available to study and the taxonomic uncertainty regarding its validity, *A. alticephalus* was excluded from this study.

*Angistorhinus aeolamnis Eaton, 1965* is known from a single skull, lacking approximately its dorsal 50–80 mm (KU 11659) from the Dockum Group of Texas. As far as can be seen from its original description, the skull does not preserve any of the

features indicative of the genus *Angistorhinus*, such as posterior parietal extensions or the parietal-squamosal bars forming a posterolateral curve when viewed dorsally (*Long & Murry, 1995*). The loss of the dorsal part of the skull also greatly reduces the number of characters for which this specimen could be scored, making it likely to be problematic in phylogenetic analysis; this combined with its unclear taxonomic affinities leads us to exclude this taxon.

*Brachysuchus megalodon Case, 1929* is a very robust taxon, represented by the largely complete, but dorsoventrally crushed holotype skull (UMMP 10336), a likely associated mandible (UMMP 10336a) and a second, well preserved, also largely complete skull (UMMP 14366), from the Dockum Group of Texas. *B. megalodon* has historically been a difficult taxon to interpret, being synonymized with '*Phytosaurus*' (*Gregory, 1962a*) and *Angistorhinus* (*Long & Murry, 1995*) before being provisionally resurrected by *Stocker (2010)* pending a full reanalysis of the taxon. *B. megalodon* is excluded here because the material was unavailable for study due to the redevelopment of the UMMP museum. Although the original description by Case is very detailed and contains many line drawings, it was deemed unfeasible to score such a taxonomically problematic specimen that has been subjected to severe taphonomic distortion from images alone, especially as the less distorted referred specimen has only ever been figured in palatal view (*Case & White, 1934*).

'*Machaeroprosopus validus*' *Mehl, 1916* was erected on the basis of an incomplete skull (UW 3807) from the Chinle Formation of Arizona. This specimen, which has been lost (*Westphal, 1979*), was long considered to be the holotype specimen for the genus *Machaeroprosopus* (*Case, 1920*; *Camp, 1930*; *Colbert, 1947*; *Ballew, 1989*; *Hungerbühler, 1998*). However, the holotype of *Machaeroprosopus buceros* was recently found to take priority (*Parker, Hungerbühler & Martz, 2012*). Considering the loss of the only specimen and its now decreased taxonomic significance and uncertain taxonomic position this taxon is here excluded.

*Mesorhinosuchus fraasi* (*Jaekel, 1910*) was named based on a single partial skull, reportedly from the Middle Buntsandstein of Saxony-Anhalt, Germany. The supposed type locality is dated as Olenekian in age, making this potentially the stratigraphically oldest phytosaur, and predating even *Diandongosuchus* by approximately 10 million years. The specimen, which was housed at the University of Göttingen, was destroyed in WWII and only one photograph exists in the original description by *Jaekel (1910)*; moreover, its stratigraphic provenance has frequently been questioned (*Gregory, 1962a, 1969*; *Hunt & Lucas, 1991*). In any case, this species is excluded due to the loss of the type specimen.

'*Paleorhinus magnoculus*' *Dutuit, 1977* is represented by a single, very small (275 mm anteroposterior length) juvenile skull (MNHN ALM 1) from the Argana Formation of Morocco. It was originally described as a unique species of '*Paleorhinus*' due to (among other features) its proportionately enormous orbits and small antorbital fenestrae; however, these putative autapomorphies were later reinterpreted as a reflection of the early ontogenetic stage of the type specimen (*Fara & Hungerbühler, 2000*) and the species was reclassified as an indeterminate specimen of *Parasuchus*, a view that is shared in this

study (but see *Kammerer et al., 2015*). This taxon is therefore excluded from this study because the inclusion of ontogenetically variable features could affect its phylogenetic placement, as has been extensively reported in dinosaurs (*Rozhdestvensky, 1965*; *Dodson, 1975*; *Sampson, Ryan & Tanke, 1997*; *Scannella & Horner, 2010*; *Tsuihiji et al., 2011*).

*Promystriosuchus ehlersi* (*Case, 1922*) is known from a poorly preserved partial skull from the Dockum Group in Texas (UMMP 7487). The specimen displays extensive dorso-ventral crushing with many elements not retaining their original associations; as such, it is a difficult specimen to interpret. It has previously been referred to 'Paleorhinus' (=*Parasuchus*) (*Gregory, 1962a*; *Hunt & Lucas, 1991*; *Long & Murry, 1995*), but more recently its taxonomic position has been seen as uncertain (*Kammerer et al., 2015*). As with *B. megalodon* the sole specimen of this taxon was unavailable for study, and it represents a taxonomically uncertain specimen with challenging morphology and few images available in the literature; for these reasons *Promystriosuchus ehlersi* is not included in this study.

## Continuous data in cladistics

The use of continuous characters in cladistics has historically been controversial, with many researchers questioning their validity and appropriateness to cladistic methods (*Crisp & Weston, 1987*; *Pimentel & Riggins, 1987*; *Cranston & Humphries, 1988*; *Felsenstein, 1988*; *Stevens, 1991*). The majority of concerns raised have been around the discretization of frequently overlapping taxonomic ranges of continuous measurements into distinct character states using methods often criticized as arbitrary (*Poe & Wiens, 2000*).

Indeed, techniques such as gap-coding (*Mickevich & Johnson, 1976*) and segment-coding (*Thorpe, 1984*; *Chappill, 1989*) do suffer from elements of arbitrariness: in gap-coding the size of the fundamental gap, and in segment-coding the number of segments, must be specified by the researchers (*Rae, 1998*). These metrics may be based on various statistical concepts, such as 95% confidence intervals or standard deviations about the mean, and data may be treated on a linear or logarithmic scale; however, as shown by *Gift & Stevens (1997)* the choice of which metric to use can have a profound effect on the final character states.

Despite the general rejection of continuous data by many authors, continuous ranges of overlapping data have remained common in cladistic matrices, scored via character states with arbitrary 'discrete' cut-offs, which are generally not explained or justified, for example, 'ratio of femoral length to width: <6 [0], $\geq$6 [1]', or 'shape of orbit: circular [0], oval [1]' (*Stevens, 1991*; *Poe & Wiens, 2000*; *Wiens, 2001*). These arbitrary character states have been shown to convey little phylogenetic information compared to identical data ranges coded using gap-weighting (*Garcia-Cruz & Sosa, 2006*). Despite this, these types of characters are frequently found in modern cladistic datasets, including recent analyses of phytosaur phylogeny (*Hungerbühler, 2002*; *Hungerbühler et al., 2013*; *Parker & Irmis, 2006*; *Stocker, 2010*, *2012*, *2013*; *Butler et al., 2014*; *Kammerer et al., 2015*). This study aims to incorporate continuous morphological data, including that of 'shape', characterized in a non-arbitrary manner to increase the quantity of phylogenetically useful

information available to studies of phytosaur systematics, with the goal of increasing their accuracy and resolution.

As expressed above, the main problem with many continuous coding techniques is the arbitrary splitting of range data into discrete character states. The software package TNT overcomes this problem by employing a similar technique to gap-weighting (*Thiele, 1993*) and step-matrix gap-weighting (*Wiens, 2001*). Gap-weighting splits the range of species mean values into as many character states as allowed by the software (32 in PAUP*), thus increasing coding resolution and (as the characters are ordered) ensuring large changes must pass through many steps in comparison to small changes, thus increasing their weight. This technique is, however, hampered by the limits imposed by the software. Step-matrix gap-weighting follows a similar initial procedure, but circumvents the limit on character weighting by using the sizes of the gaps between unique character states, rescaled along a range from zero to the maximum steps allowed by the software (1000 in PAUP*), to create step-matrix values to weight character state changes. Although gap-weighting provides a higher resolution of states into which measured variation can be categorized, the categorization method is still fundamentally arbitrary and, due to this, taxon ranges that are significantly different may be grouped together and those that are statistically identical may be split up (*Farris, 1990*).

The techniques developed in TNT (*Goloboff, Mattoni & Quinteros, 2006*; *Goloboff, Farris & Nixon, 2008b*), and used in this study, remove arbitrary discretization by analysing the taxon range values as they are, that is, without being grouped into character states. This is possible through the use of *Farris' (1970)* down-pass and *Goloboff's (1993)* up-pass algorithms which are designed to use numerical differences between the states being optimized; therefore, the actual intervals between taxon data ranges, being numerical, are treated in the same way as ordered character states (*Goloboff, Mattoni & Quinteros, 2006*). As mentioned in *Goloboff, Mattoni & Quinteros (2006)*, step-matrix gap-weighting would produce the same outcome as the TNT technique; however, this approach becomes difficult with a large number of taxa and is not capable of handling ranges of variation. As the scale of the step changes, and therefore weights, are directly proportional to the measured data, the magnitude on which the original measurements were made could have a large (and often unwarranted) influence on character weighting. *Goloboff, Mattoni & Quinteros (2006)* suggested that implied weighting (re-weighting of characters based on their level of homoplasy) can reduce this issue, however, this was found to be only a partial solution and a combination of implied weighting and re-scaling trait measurement values to unity produced far more satisfactory results (*Koch, Soto & Ramírez, 2015*).

## Geometric morphometric data

Geometric morphometric characters are a relatively new development in cladistics (*Catalano, Goloboff & Giannini, 2010*; *Goloboff & Catalano, 2011*; *Goloboff & Catalano, 2016*). In relation to phylogenetics, the use of geometric morphometrics tends to be equated with phenetic studies and the use of techniques such as principal components analysis to reduce overall morphology to a small number of axes of covariation.

The method presented by *Catalano, Goloboff & Giannini (2010)* avoids this: *x*, *y* and *z* landmark coordinates are used, without transformation, to generate ancestral state reconstructions using a spatial optimization technique which minimizes displacement between individual, or configurations of, landmarks from two descendants. A thorough discussion of the applicability of geometric morphometrics in phylogeny is given by *Catalano, Goloboff & Giannini (2010)* in which previous arguments against its use are also addressed. When integrated into a phylogenetic analysis of Vespinae (*Perrard, Lopez-Osorio & Carpenter, 2016*), landmark characters were generally found to improve tree resolution when combined with a morphological character matrix. Landmark characters still exerted a noticeable effect with the addition of molecular data, though only four of the 10 relationships generated by landmark data were supported in the morphological + landmark + molecular data trees (*Perrard, Lopez-Osorio & Carpenter, 2016*). In these trees the landmark data mostly affected poorly supported nodes—allowing greater resolution, though possibly only due to over-resolution due to the analysis techniques. It was also found that the landmark data alone were insufficient to reliably resolve relationships, likely due to homoplasy arising from the functional unit in which the landmark characters were placed (*Perrard, Lopez-Osorio & Carpenter, 2016*). Although the quantity of information may be increased by using landmark characters, not all information is included, which could lead to important features being excluded.

## Character coding

The character list (Appendix 2) was constructed by combining those used in previous analyses (*Ballew, 1989*; *Hungerbühler, 2002*; *Stocker, 2010*; *Butler et al., 2014*; *Kammerer et al., 2015*) as well as by identifying new characters based on first-hand study of specimens and published literature. In order to compare the effects of different character types on phylogenetic results, all characters (including continuous and GM) were scored and input into one matrix, each character type as a different data block. The resulting matrix contained three blocks of data: discrete scores, continuous ranges and GM coordinates. Many of the continuous and GM characters were based on discrete characters from previous analyses, for which the categorization of character states seemed inappropriate, for example, for relative linear measurements of morphological features, or complex morphologies. Therefore, some characters in the discrete data block are discrete versions of continuous or GM characters. Some continuous and GM characters incorporated here were novel; therefore, discrete versions of these were also created in the discrete data block to ensure that where phylogenies were analysed using different data types, any differences in results would not be affected simply by differences in the exact morphological information included. The different combinations of character types were incorporated into different analyses by setting either the continuous, GM, or both character blocks to 'active' or 'inactive' in the phylogenetic software TNT (see below).

The number of characters and proportion of missing data in each data block are summarized in Table S1. No characters were excluded based on quantity of missing data in scored taxa as including more characters, even if this increases the proportion of missing data, has been shown to increase accuracy in phylogenetic analysis (*Wiens, 1998*).

**Table 3 Summary of the character composition for all four datasets (D, DC, DM and DCM) analysed in this study.**

| | Number of characters | Description | Characters encoded using continuous or GM methods |
|---|---|---|---|
| D | 94 | Discretely encoded characters only | None |
| DC | 94 | Discretely and continuously encoded characters (no characters scored using GM) | 8, 11, 25, 38, 43, 54, 60, 87, 89, 94 |
| DM | 90 | Discretely and GM encoded characters (no characters scored continuously) (Reduced character count, as GM characters often correspond to multiple discrete characters) | 39, 40, 46, 50, 54, 55, 81, 89, 91 |
| DCM | 90 | Discretely, continuously & GM encoded characters (all scoring methods used) (Reduced character count, as above) | 8, 11, 25, 38, 39, 40, 43, 46, 50, 54, 55, 60, 81, 87, 89, 91, 94 |

This technique increases the possibility of long branch attraction (*Swofford et al., 1996*), but is less likely in a dataset where missing data is distributed randomly among all taxa (*Poe & Wiens, 2000*); in our dataset missing data seem more likely to occur in certain taxa and certain characters, therefore the possibility of long branch attraction should be kept in mind when interpreting the results.

A consistent discrete matrix was used as a base for each analysis, into which continuous or GM characters were swapped with their discrete counterparts. The discrete data block consisted of 94 characters, the continuous block 10 characters and the GM block five characters. These were combined in four analyses (Table 3): (1) discrete characters only (D coding treatment) (94 characters, 21 of which are ordered), (2) discrete + continuous characters (DC coding treatment) (94 characters, 21 ordered), (3) discrete + GM (DM coding treatment) (90 characters, as some GM characters encompass variation described by more than one character in the discrete dataset; 19 ordered), (4) discrete + continuous + GM (DCM coding treatment) (90 characters, 19 ordered). A full list of all characters, ordering and the correspondences of continuous and GM to discrete characters is available in Appendix 2. The coding procedures used here for continuous and GM characters are described below, as are the methods of character state distinction for their discretized counterparts.

It is important to note here that when incorporating continuous and GM character scorings for analysis, the format of the TNT data file requires these characters to be presented first in the file. This differs from how the characters are ordered in our character list (Appendix 2). Our character list presents characters in the order in which they occur for the base discrete data block; where a character possesses a continuous or GM variant this is flagged next to that character. It should also be noted that characters in a TNT file begin at zero, whereas we shift our characters such that the list begins at one.

### Continuous characters

Measurements were taken from all referred specimens with the appropriate morphology preserved, either directly, using digital callipers, or from photographs, using the software ImageJ. Standard error was calculated about the mean score of each species, this was then used to calculate min–max species ranges with statistically meaningful differences (*Goloboff, Mattoni & Quinteros, 2006*). Min–max species range values were rescaled

in each character using the formula: $z_i = x_i - \min(x)/\max(x) - \min(x)$ where $z_i$ is the rescaled value, $x_i$ is the original value and $\min/\max(x)$ are respectively the minimum and maximum original values in the range of variation across all taxa for that character. This rescales values onto a 0–1 scale, ensuring that magnitudes of interspecific differences within characters are maintained, whilst between-character weighting is standardized.

The rescaled range values (and where only one specimen is known, the single values) were input into the data matrix file and treated as ordered.

### GM characters

Many features of phytosaur skulls that are appropriate for shape analysis contain few discrete landmark positions, making traditional landmark analysis difficult, and the resolution of the morphology influencing the results would be poor. For example, only two sutures regularly form connections on the border of the antorbital fenestra that could be landmarked in all phytosaurs, and due to the variable shape of the fenestra there are no consistent 'corners' or other morphological features that can be traditionally landmarked on the border, aside from the most anterior and posterior extremities. Conversely, these problems can be resolved by using sliding semi-landmarks to approximate outline shape; this is the technique used here. In techniques such as principal components analysis, semi-landmarks require special treatment, on account of their reduced dimensionality and therefore degrees of freedom (*Bookstein, 1996*; *Zelditch, Swiderski & Sheets, 2012*); however, as TNT does not use such analyses and providing the user employs appropriate Procrustes alignment techniques, nothing precludes their use. Semi-landmarks were digitized from photographs using the 'draw background curves' tool in the software tpsDig2 (*Rohlf, 2015*) to capture a detailed outline of the structure; this was then resampled to contain a consistent number of equally spaced points which were used for alignment. See Fig. S1 for configurations of landmarks in GM characters. Semi-landmarks were subjected to sliding and Procrustes superimposition to minimize distances between configurations using the R package Geomorph (*Adams & Otárola-Castillo, 2013*). In TNT, landmark configurations were scaled to unity using the command '*lmark rescale =\*;*'. Whole configurations of landmarks were used for optimization and to calculate support values, rather than a pairwise approach with each individual landmark, as semi-landmarks define curves and not homologous points.

### Discrete characters

Characters consisting of continuous measurements such as ratios were discretized into character states using primarily quantitative, but also qualitative approaches; all measurements from all referred specimens were sorted numerically and character state divisions were introduced where gaps occurred in their sequence. Where no substantial gaps occurred character states were introduced at points between substantial transitions in the data. For example: in a hypothetical dataset of four taxa, A–D, each represented by four specimens which all occupy a 0–10 continuous scale for one of their characters, if all or a substantial majority of specimens from taxa A and B sit between

zero and five, whereas those of taxa C and D sit between five and 10, the continuous character range would be divided into two character states at number five. This therefore splits the continuous range into discrete states in the absence of gaps.

This treatment was designed to mimic the presumably qualitative techniques for dividing continuous data into discrete states used in previous analyses (although the delimitation technique has never been described in any previous phytosaur phylogeny), and represents a similar treatment to the 'arbitrary' method of *Garcia-Cruz & Sosa (2006)*. Discrete characters used as counterparts to implicitly ordered continuous characters were also treated as ordered. This means that different topologies resulting from different combinations of character types reflect changes in character coding approach rather than differences in the approach to character ordering.

### Implied weighting

Implied weighting (*Goloboff, 1993*) is a method of character weighting in which the number of step changes a character undergoes in its current tree topology is compared to the minimum possible for that character, as a metric for homoplasy. Each character in a tree topology is then weighted in inverse proportion to its level of homoplasy, with a concavity constant ($k$) ascribing the severity of weighting. These weighted scores of 'character fit' are then summed to provide an estimate of character fit for the whole tree; each tree topology in the analysis undergoes the same procedure, with the 'best' overall tree (s) having the best character fit score. We primarily use implied weights here for its apparent advantages in the analysis of matrices high in homoplasy (*Goloboff et al., 2008a*); a problem well-recognized in Phytosauria (*Hungerbühler, 1998*, *2002*). Although implied weighting has been criticized recently (*Congreve & Lamsdell, 2016*) it does also have advantages when using continuous and GM character scorings. Continuous characters may be measured on different scales, and this difference in scaling is transferred to a character's step-matrix (arbitrarily increasing the impact of 'large-scale' characters); accordingly, homoplasy in characters measured on large scales tends to be greater and these characters are thus down-weighted in proportion with this (*Goloboff, Mattoni & Quinteros, 2006*). In this study we further address issues of scaling by standardizing continuous character ranges into a 0–1 range, as described above. Implied weighting also provides a method for weighting landmark-based characters and can be performed either for each individual landmark within a configuration or for whole configurations using the average homoplasy. The latter method is particularly useful in this study as we use semilandmarks; as such the individual landmarks do not necessarily represent homologous points, rather it is the overall structure that is important—it is therefore the whole configuration of landmarks that should be treated as a single character for weighting.

### Analyses

All analyses were performed in the software TNT version 1.5 (*Goloboff & Catalano, 2016*), under extended implied weighting with the concavity constant '$k$' set to vary for each character depending on the quantity of missing entries (using '*xpiwe (*' commands).

Implied weighting requires the minimum possible length for each character coding in order to calculate homoplasy; however, this is problematic in landmark data (*Goloboff & Catalano, 2016*). Therefore, TNT provides an option to find minimum values for each landmark using heuristic searches; this search function was applied before analysing any dataset incorporating GM characters, then the minima were added to the file for use during tree searching. Furthermore, GM characters were each weighted separately according to the average homoplasy of their landmark configuration (using '*xpiwe [*' commands); therefore, weighting was based on entire configurations rather than the sum of component landmarks, which as stated above, may not be individually homologous.

### Analysis parameters

Tree searches were performed using the new technology algorithms in TNT: 10,000 random addition sequences, analysed using TBR swapping with 10 iterations of drift and ratchet, followed by a sectorial search and finally three rounds of tree fusing. The search was performed until the minimum tree length was hit five times. The duration of tree searches dramatically increased with the addition of GM characters; therefore, only 200 random addition sequences were used and minimum length was found only once. Furthermore, because landmark data is relatively unstructured the perturbation phases of ratchet and drifting can produce trees that are 'too suboptimal' and therefore greatly increase the search time (*Goloboff & Catalano, 2016*). We therefore followed the suggestion of *Goloboff & Catalano (2016)* and increased the drift 'xfactor' to 5, decreased the percentage of swapping to be completed to 90%, decreased the number of substitutions to 45, and for ratchet, lowered the probability of reweighting (both up and down) to 3 and decreased the number of substitutions to 30.

Bremer supports were calculated using 10,000 (D and DC) or 1,000 (DM and DCM) trees suboptimal by a fit of 10; branch swapping using TBR was performed and absolute supports were calculated based on the results. Robusticity analysis was carried out using symmetric frequencies, with TBR swapping beginning from 10 Wagner trees and 10,000 (D, DC) or 100 (DM, DCM) replicates. As the matrices including GM data were exceptionally computationally heavy and time consuming, parameters were altered such that trees were accepted without consideration of error margin during landmark searches and that swapping distance for branch swapping was reduced (commands respectively: '*lmark errmarg 0*' and '*bbreak : limit 5*').

### Output processing and comparisons

Where more than one tree of best character fit resulted from an analysis, a strict consensus was generated. With implied weighting in effect, ties in tree length (resulting in multiple best fitting trees) become very uncommon due to the use of floating-point character fit calculations. Additionally, continuous data are analysed as actual numerical differences, rather than categorical steps, also reducing the chance of exact ties. To avoid over-resolution due to the acceptance of a single or few trees showing only an extremely small difference in character fit compared to other topologies, an arbitrary Bremer support cut-off value of 0.08 was implemented, below which nodes were judged to

be poorly supported and were collapsed. In addition a second cut-off value was used (0.11) which was equal to the average step-length of a single character following weighting. This particular number was used in an attempt to emulate the procedure common in phylogenetics, to collapse nodes with a Bremer support of less than one step. These cut-offs were maintained throughout the four treatments, allowing the effects on tree resolution to be compared.

Best character fit trees resulting from each of the four analyses using different combinations of character data types (see above) were compared using several techniques. Consistency index (CI) and retention index (RI) were compared to assess the homoplasy present in the trees resulting from each analysis. Maximum agreement subtrees were constructed for each comparison to compare the number of congruent relationships between the trees; this was supplemented with a strict consensus of the two trees in case lower level congruence was masked in the agreement subtree by higher-level polytomies (*Goloboff, Mattoni & Quinteros, 2006*). Subtree pruning and regrafting (SPR) distances were calculated to find the minimum number of changes under the SPR search algorithm required to convert one tree topology into the other—essentially a numerical description of tree similarity. The rooted Robinson–Foulds (RF) distance, which measures the differential presence/absence of phylogenetic relationships between trees, was also used to measure tree-similarity.

The effect of each coding technique was assessed and compared to its alternative counterparts in several ways. Trees were initially compared using mean and summed frequency and Bremer supports across each collapsed tree, alongside the number of nodes retained after collapsing each tree to get a broad view of any major differences. For a more detailed view of the effects of data type on the nodal support each non-collapsed best fit tree was split into five tree-regions; (1) the most basal portion of the tree, including all non-Mystriosuchinae members of Phytosauria; (2) the clade formed by *Rutiodon* and *Angistorhinus*; (3) *Leptosuchus*-grade taxa, here composed of all *Leptosuchus*, *Smilosuchus* and *Nicrosaurus* species, plus PEFO 34852, '*Phytosaurus*' *doughtyi*, *Pravusuchus hortus* and *Coburgosuchus goeckeli*; (4) all members of *Machaeroprosopus* and '*Redondasaurus*', plus USNM V 17098, NMMNHS P-4256, NMMNHS P-31094 and *Protome batalaria*; (5) the clade composed of named species of *Mystriosuchus* plus NHMW 1986 0024 0001 and MB.R. 2747. The mean frequency and Bremer supports were calculated within each region to investigate the effects of different character coding techniques at a greater resolution.

The support for monophyly of groups/taxa of interest was investigated by placing them in alternative positions in a constraint tree, then re-running the analysis whilst imposing those constraints and observing the effect on character fit in the resulting trees.

The accuracy of trees, as denoted by the various nodal support metrics and comparisons described above, is a measure of internal consistency; regardless of a tree's accuracy it may still be spurious. Stratigraphic congruence was used here as an independent estimate of tree-validity; four metrics were employed which measure stratigraphic congruence differently. (1) The stratigraphic consistency index (SCI) (*Huelsenbeck, 1994*) measures the

proportion of nodes within which the first appearance datum is of the same age or younger than the sister node; these nodes are considered stratigraphically consistent. (2) The relative completeness index (*Benton & Storrs, 1994*) reports the ratio between the sum of ranges for taxa in the tree and the sum of ghost-range length within the tree. (3) The Manhattan stratigraphic measure (MSM*) (*Siddall, 1998*; *Pol & Norell, 2001*) optimizes the difference in age between the first appearances of taxa (Manhattan distance) as a Sankoff character on the proposed tree. The MSM is the ratio between the minimum possible tree length based on taxon ranges (topology determined by the Manhattan distance character), and the tree length when Manhattan distance is optimized to the original topology. The MSM is basically the CI of the distance character (*Pol & Norell, 2006*). *Pol & Norell (2001)* introduced a correction to prevent reversals in the Manhattan distance character 'states', presenting the updated metric, MSM*. (4) The gap excess ratio (*Wills, 1999*) finds the proportion of ghost range in a tree, relative to the minimum and maximum possible sum of ghost ranges for the corresponding dataset. It also optimizes age range differences on the tree in the same manner as the MSM*, but is calculated as the RI for the distance character (*Pol & Norell, 2006*).

The 'strap' package (*Bell & Lloyd, 2014*) for the software R version 3.2.5 (*R Core Team, 2016*) implements all the above metrics, and was used for all analyses of stratigraphic congruence in this study. The strap package also implements a test of statistical significance for each metric, based on random permutations. In calculating significance values we made use of two additional options offered by strap: the first is to generate random trees by swapping OTUs, whilst maintaining tree shape; the second is to fix the outgroup OTU such that it is not randomized. These additions respectively resolve issues of random trees being more symmetrical than commonly found in fossil groups (*Wills, Barrett & Heathcote, 2008*), and the deliberate assignment of the outgroup prior to analysis, removing the need for its position to be tested (*Bell & Lloyd, 2014*). The random trees therefore provide a closer estimate of the original tree topology and a more robust test of significance (*Bell & Lloyd, 2014*). Primarily the *P*-values from the significance tests are used here for comparisons of stratigraphic congruence, rather than the raw metrics, as the latter are strongly influenced by tree balance, the arrangement of taxon stratigraphic ranges and tree size (*Siddall, 1996*; *Wills, 1999*). The results of randomization tests are free from these influences and should therefore be more directly comparable (*Wills, 1999*; *Benton, Hitchin & Wills, 1999*).

In this study significance tests were carried out with 1,000 random permutations. The strict consensus trees resulting from the four data treatments were analysed, as were the three most recent alternative phylogenetic hypotheses of phytosaur relationships (*Parker & Irmis, 2006*; *Hungerbühler et al., 2013*; *Kammerer et al., 2015*). Where a previous analysis included specimen-level OTUs or taxa not present in this study, these terminals were removed; three terminals were removed from the tree of *Kammerer et al. (2015)* and two from *Hungerbühler et al. (2013)*. Three alternate hypotheses of topology were presented by *Hungerbühler et al. (2013)*, though with the two terminals missing from this analysis removed, two of the trees become synonymous; therefore, only two hypotheses are tested here from *Hungerbühler et al. (2013)*.

## Model-based cladistic methods

In palaeontology, parsimony-based methods of phylogenetic analysis have historically dominated the field and continue to be the preferred analysis method for morphological data. Although model-based approaches to phylogenetics, such as maximum likelihood and Bayesian methods, are relatively common in analyses including molecular data, their application to palaeontological datasets has only recently become more widespread (*Lee & Worthy, 2011*). It seems likely that the tardiness with which palaeontologists have taken up probabilistic methods is linked to the ongoing debate over the relative performance of parsimony and probabilistic methods (*Huelsenbeck, 1995*; *Lee & Worthy, 2011*; *Wright & Hillis, 2014*; *O'Reilly et al., 2016, 2017*; *Goloboff, Torres & Arias, 2017*, *Goloboff, Torres Galvis & Arias, 2018*; *Sansom et al., 2018*), especially regarding morphological characters which constitute the vast majority of palaeontological datasets. However, theoretical criticisms have also been made against both model-based (*Kolaczkowski & Thornton, 2004*; *Goloboff & Pol, 2005*; *Livesey & Zusi, 2007*; *Wagner, 2012*) and parsimony approaches (*Felsenstein, 1978*; *Kuhner & Felsenstein, 1994*; *Lewis, 2001*).

Advances in the probabilistic approach to phylogeny, stemming from the MK model of discrete trait evolution (*Lewis, 2001*), have led to a more robust framework with which to analyse morphological datasets (*Ronquist & Huelsenbeck, 2003*; *Wagner, 2012*; *Wright, Lloyd & Hillis, 2015*). Similarly to parsimony methods, recent advances have seen the development of procedures to incorporate both continuous (*Parins-Fukuchi, 2017*) and GM data (*Parins-Fukuchi, 2018*) into probabilistic analyses of phylogeny. Both methods utilise alternative models of evolution to the MK model: Brownian motion (*Felsenstein, 1973, 1985*; *Gingerich, 1993*) and Ornstein–Uhlenbeck (*Hansen, 1997*; *Butler & King, 2004*; *Beaulieu et al., 2012*) models simulate random, normally distributed phenotypic evolution, and stabilising evolution respectively in the analysis of continuous data, while a Brownian motion model is used again, with branch lengths representing morphological variation, to analyse morphometric data (*Parins-Fukuchi, 2017, 2018*).

Probabilistic methods of phylogenetic analysis are not explored further in this study, largely for practical reasons. We note that the use of continuous and morphometric data in probabilistic methods is very new and as such lacks intuitive implementation in software packages, resulting in a requirement for careful documentation and testing of methodological properties, especially for an empirical dataset. Further analyses of this dataset under probabilistic methods could be very illuminating, and a potentially fruitful avenue of future research, but will be explored elsewhere.

## RESULTS

A total of eight best fit trees were found across all four coding variants; in each of the D and DC treatments three equally 'fitting' trees were found, whereas DM and DCM each returned only one best fit tree. Our results are presented as the strict consensus trees of the best fit trees or single best fit trees resulting from each of the four different variants of character coding (D, DC, DM and DCM) with absolute and relative symmetric resampling

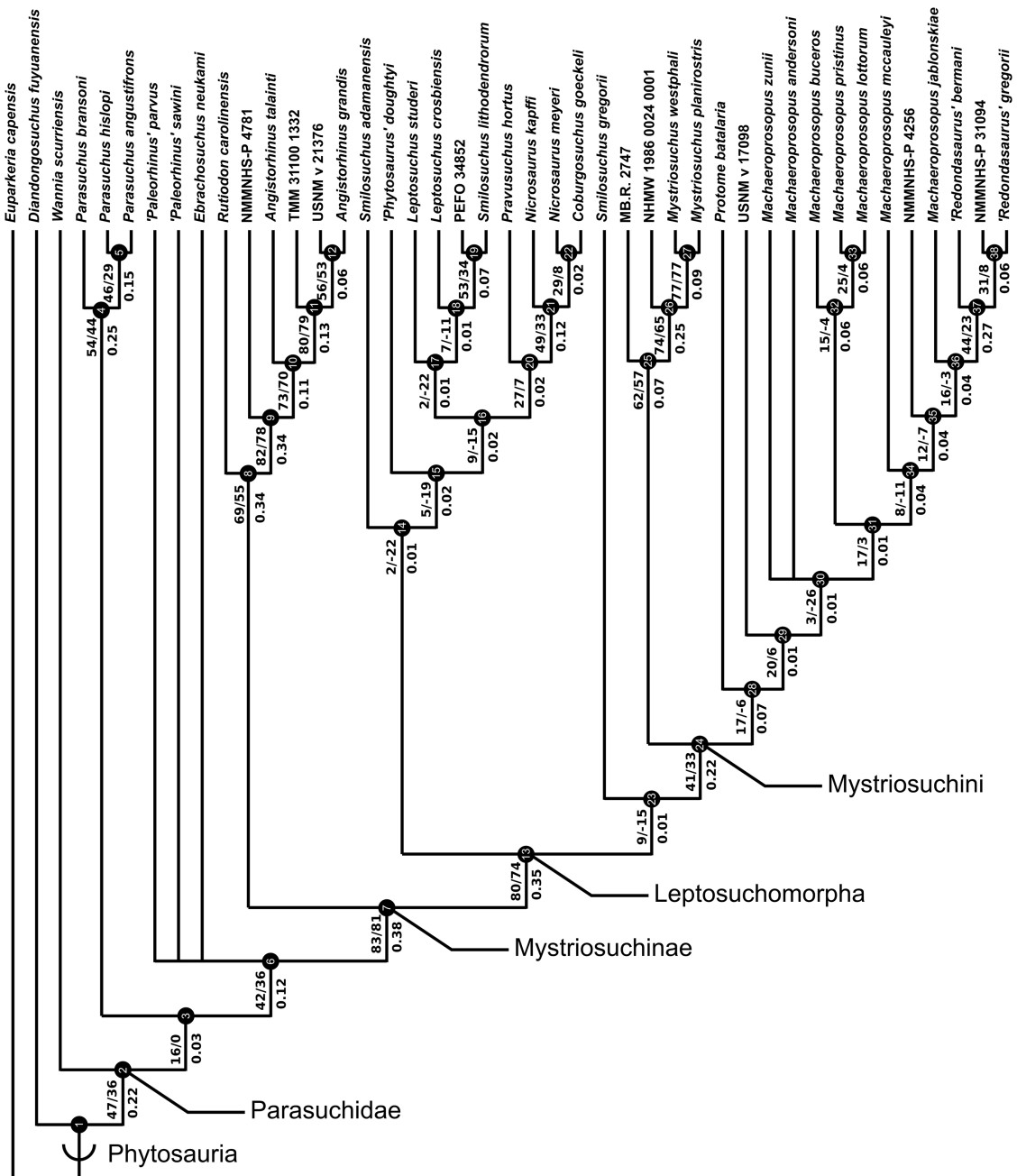

**Figure 4 Strict consensus of the three trees resulting from the analysis treatment incorporating discrete characters only (D).** Node numbers are labelled within black circles. Absolute frequencies/frequency differences are presented to the left of the node's stem; Bremer support values are reported to the right of the node's stem.

frequencies above nodes, and Bremer supports below (Figs. 4–7). We also present the strict consensus and maximum agreement subtree of these four trees, to summarize the most consistent relationships across all coding treatments (Fig. 8 and Fig. S2).

The tree lengths resulting from the four coding treatments are summarized in Table 4, as are the CI and RI. Tree lengths are not directly comparable between treatments including or excluding GM coding; this arises because the morphology encoded in some

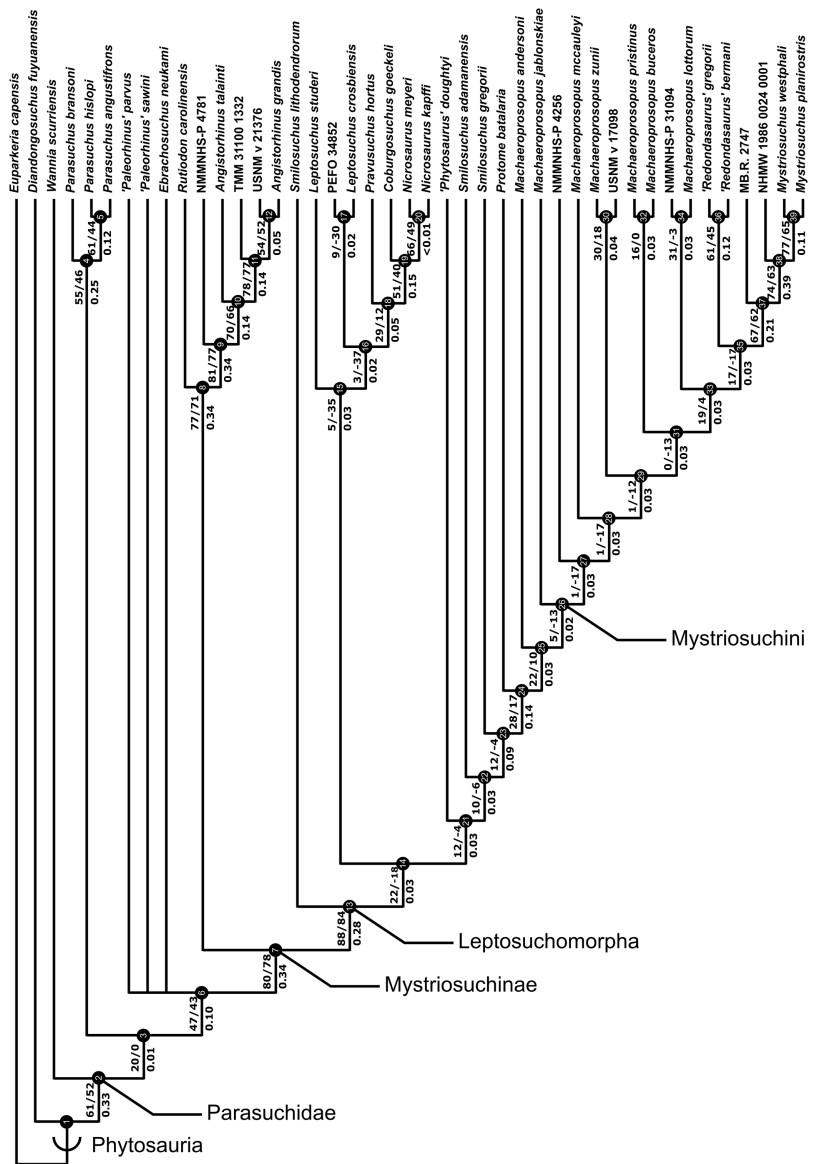

**Figure 5 Strict consensus of the three trees produced by the treatment incorporating discrete and continuous character scoring (DC).** Node labels as in Fig. 4.

GM characters encompasses more than one discretely coded character. Therefore, analyses incorporating GM data contain fewer characters than the other scoring types and will likely show lower tree lengths as a result.

Conversely, providing that continuous characters replace their corresponding discrete characters with one-to-one equivalence (which they do here), their alternative coding method alone should not affect tree length. Continuous characters are here scored as ratios and are transformed to occupy a 0–1 scale; the standard treatment of continuous characters by TNT uses the numerical differences between scores to create the step-matrix. As these values are constantly below 1 it may be expected that the greater proportion

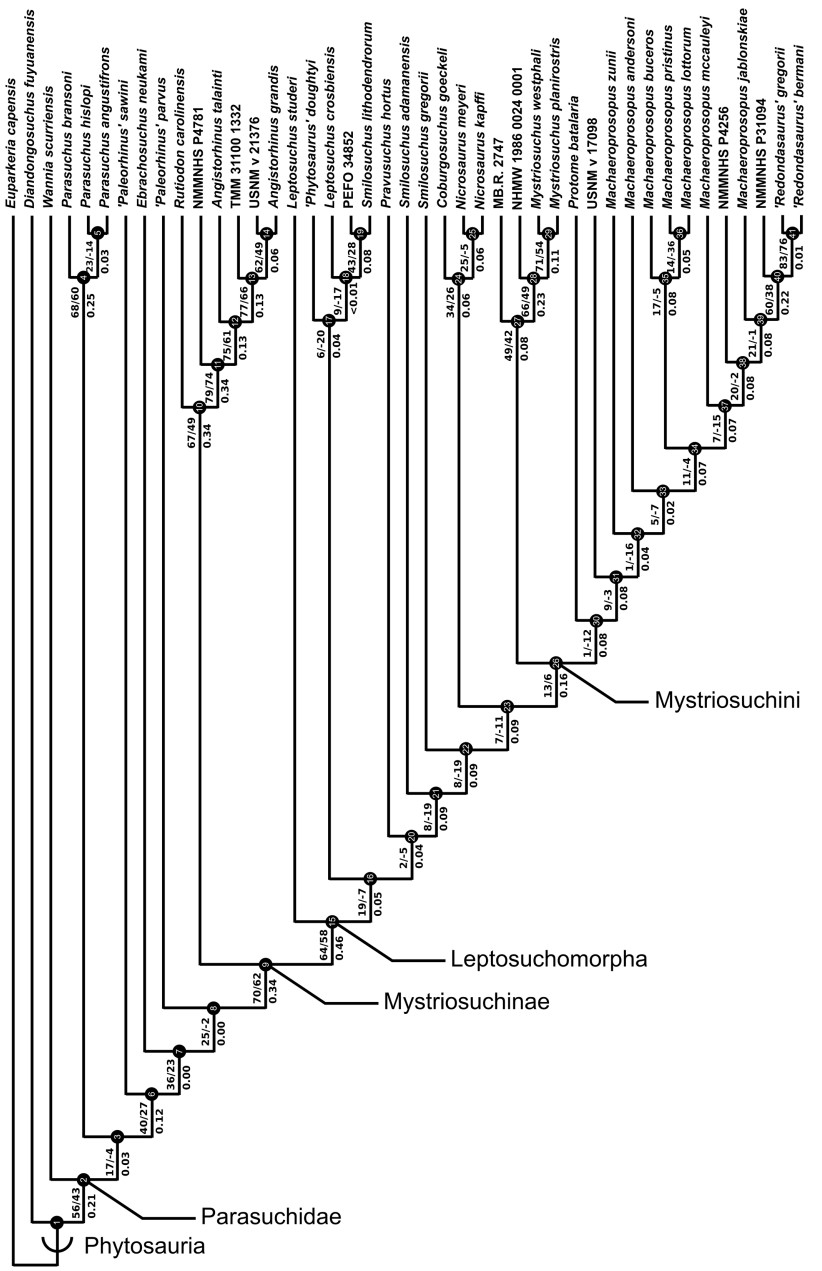

**Figure 6 Single tree resulting from the treatment incorporating discrete and geometric morphometric character scoring (DM).** Node labels as in Fig. 4.

of continuous characters in a dataset would result in lower tree length. However, due to our use of implied weighting this should not present a problem, as tree length is the sum of homoplasy-adjusted character weight. Homoplasy is, in the simplest sense, calculated as a proportion of the minimum length of a character in topology X, and the minimum possible length of a character in any topology. Character weight is then calculated from this proportion (homoplasy) and is then summed across all characters to generate tree length. As character weight is based on a character-specific proportion, the actual size of

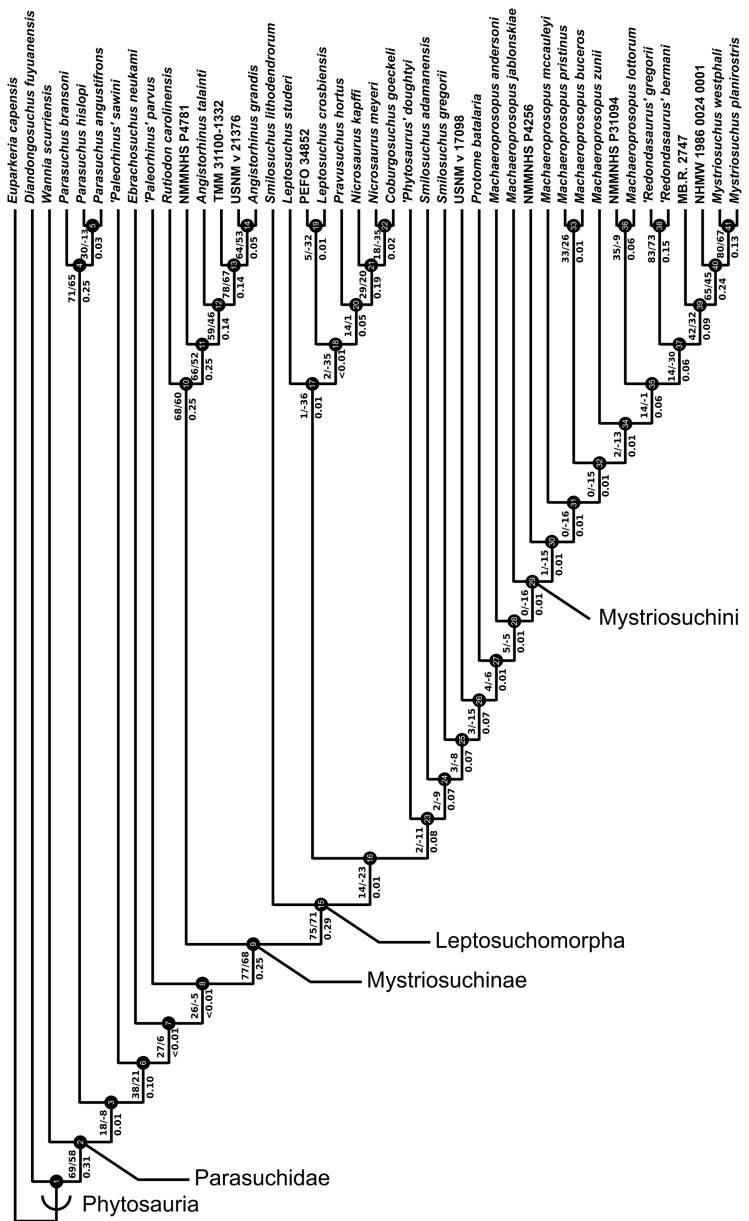

**Figure 7** **Single tree resulting from the treatment incorporating all three character scoring methods; discrete, continuous and geometric morphometric (DCM).** Node labels as in Fig. 4.

changes in the character step-matrix should not affect the final tree length. Simply put, if equivalent discrete and continuous characters share a consistent proportion of homoplasy, their effect on tree-length under implied weighting will be identical regardless of how they are scored.

## Comparisons of similarity

Comparisons of trees are presented in Tables 5 and 6, using the number of taxa retained by maximum agreement subtrees, the SPR distance and the RF distance as metrics of

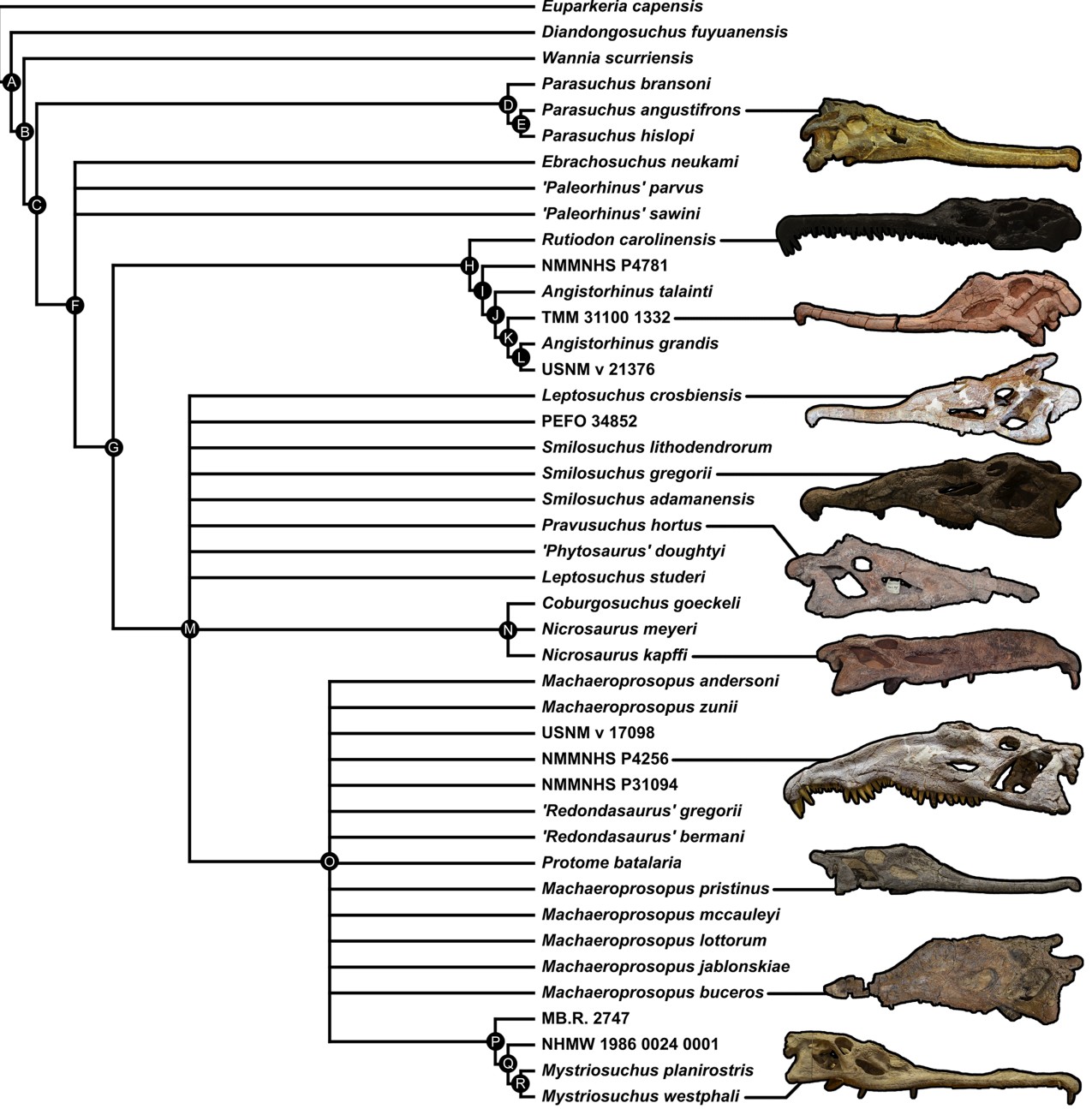

**Figure 8** Strict consensus tree constructed with the four trees presented in Figs. 4–7 (D, DC, DM & DCM). All photographs were taken by Andrew Jones.

similarity. Maximum agreement subtrees essentially produce fully resolved consensus trees by pruning taxa in conflict between the input trees; the number of taxa retained in a maximum agreement subtree can be used as a measure of topological similarity between two or more trees.

All four trees were found to be significantly similar to each other. For all pairwise comparisons between different coding treatments the number of taxa retained in the maximum agreement subtrees was statistically much greater than expected by chance.

**Table 4 Consistency index (CI), retention index (RI) and 'tree-lengths' of the four phylogenetic trees corresponding to the four data treatments.**

|  | Consistency index (CI) | Retention index (RI) | Tree length |
|---|---|---|---|
| Discrete only (D) | 0.383 | 0.689 | 31.90 |
| Discrete + continuous (DC) | 0.409 | 0.684 | 27.46 |
| Discrete + GM (DM) | 0.391 | 0.691 | 30.52 |
| Discrete + continuous + GM (DCM) | 0.420 | 0.689 | 25.44 |

**Table 5 Similarity of the trees from each coding treatment.**

|  | Discrete only (D) | Discrete + continuous (DC) | Discrete + GM (DM) | Discrete + continuous + GM (DCM) |
|---|---|---|---|---|
| Discrete only (D) |  | 27 taxa (62.8%) 39 trees | 33 taxa (76.7%) 18 trees | 26 taxa (60.5%) 27 trees |
| Discrete + continuous (DC) | 13 moves (Similarity: 0.675) |  | 24 taxa (55.8%) 36 trees | **38 taxa (88.4%) 9 trees** |
| Discrete + GM (DM) | 6 moves (Similarity: 0.850) | 11 moves (Similarity: 0.725) |  | 26 taxa (60.5%) 6 trees |
| Discrete + continuous + GM (DCM) | 12 moves (Similarity: 0.700) | **3 moves (Similarity: 0.925)** | 12 moves (Similarity: 0.700) |  |

Note:
Trees are compared using number of taxa retained in their maximum agreement subtree (green), and the number of moves under SPR swapping to move from one tree to the other (blue). The pair of most similar trees are highlighted and emboldened in both comparison techniques. Multiple maximum agreement subtrees are frequently produced where several OTUs are in conflict with each other and when any combination of their pruning results in the same final subtree; the number of trees produced is also included in the table.

**Table 6 Similarity of tree topologies as measured using Robinson–Foulds distance.**

|  | Discrete only (D) | Discrete + continuous (DC) | Discrete + GM (DM) |
|---|---|---|---|
| Discrete + continuous (DC) | 0.45122 |  |  |
| Discrete + GM (DM) | 0.23171 | 0.48780 |  |
| Discrete + continuous + GM (DCM) | 0.45122 | **0.21951** | 0.48780 |

Note:
The most similar combination of topologies, indicated by the shortest distance between input trees, is highlighted and emboldened.

Statistical significance was established using 5,000 agreement subtrees constructed with randomized tree topologies. None of these subtrees retained more than 14 OTUs and subtrees retaining the highest number of OTUs (14) comprised only 0.96% of the data. All pairwise comparisons yielded multiple maximum agreement subtrees of the same length showing alternative prunings (Table 5).

The two coding treatments that utilized continuous data (DC and DCM: Figs. 5 and 7) were consistently found to be the most similar tree topologies using all similarity metrics. The trees generated from discrete and discrete + GM coding treatments (D and DM: Figs. 4 and 6) also showed a high degree of similarity to each other. However, there is greatly reduced similarity when the DC/DCM trees are compared with the D/DM

trees. Broadly speaking, this suggests there are two partially conflicting phylogenetic hypotheses, one represented by the DC and DCM trees (Figs. 5 and 7) and one by the D and DM trees (Figs. 4 and 6). However, the agreement subtrees suggest that the amount of overlap between these hypotheses is still greater than would be expected to occur by chance.

## Consistent relationships

A list of nodal synapomorphies for each tree is presented in Appendix 3. The following relationships were found to be consistent in the trees of all four scoring treatments, and match the topology of the strict consensus tree (Fig. 8).

*Diandongosuchus* is recovered as the most basal phytosaur in every tree. Its position outside of all other phytosaurs is supported well by frequency and Bremer supports, and two consistent synapomorphies supporting Parasuchidae (Fig. 8, node B) to the exclusion of *D. fuyuanensis* in every tree [13: 0→1; 22: 0→1].

*Wannia scurriensis* is consistently found as the most basal member of Parasuchidae, outside the clade that includes *Parasuchus* and Mystriosuchinae. The latter clade (Fig. 8, node C) is, however, poorly supported, with only two synapomorphies supporting *Parasuchus* + Mystriosuchinae to the exclusion of *Wannia* in all four trees [36: 0→1; 69: 0→1].

*Parasuchus* (Fig. 8, node D) is consistently found to include the species *Parasuchus bransoni*, *Parasuchus hislopi*, and *Parasuchus angustifrons*, and is well supported by frequency and Bremer scores, with three synapomorphies common to all trees [23: 0→1; 26: 0→1; 50: 0→1].

'*Paleorhinus*' *parvus*, '*Paleorhinus*' *sawini* and *Ebrachosuchus neukami* are closer to Mystriosuchinae than to *Parasuchus* in all trees; however, the interrelationships of these species and their exact relationships to Mystriosuchinae are variable in the different coding treatments. Mystriosuchinae itself (Fig. 8, node G) is supported by three synapomorphies common to all trees [9: 0→1; 14: 1→2; 80: 0→1].

*Rutiodon carolinensis* and *Angistorhinus* form a clade at the base of Mystriosuchinae that is consistently well supported by frequency and Bremer supports (Fig. 8, node H) and is united by two synapomorphies in all trees [22: 2→1; 92: 0→1]. Within this clade, *Rutiodon* is consistently the sister taxon to *Angistorhinus*; the clade composed of *Angistorhinus* and *Angistorhinus*-like specimens, to the exclusion of *Rutiodon carolinensis* (Fig. 8, node I), is supported by two synapomorphies [56: 0→1; 58: 0→1]. The relationships of the species and specimen-level OTUs within *Angistorhinus* are consistent in all coding treatments: *A. talainti* is the most basal of the two named species and *A. grandis* is more derived, with the specimen-level OTUs representing either potential additional species within the genus, or morphologically diverse representatives of existing *Angistorhinus* species.

Leptosuchomorpha (Fig. 8, node M) possesses two synapomorphies common to all tree topologies that separate it from the more basal taxa [16: 1→0; 25: 0→1]. Within Leptosuchomorpha the four phylogenies are more variable (Fig. 8, node M). Among the leptosuchomorph OTUs not included in Mystriosuchini there is only one clade

common to all tree topologies: the clade which unites *Nicrosaurus kapffi* and *Nicrosaurus meyeri* with *Coburgosuchus goeckeli*, although the relationships between these three species are variable in the different coding treatments (Fig. 8, node N). This clade is supported by a single synapomorphy [57: 1→2].

Although there are conflicting relationships, the majority of the leptosuchomorph taxa that have been excluded from Mystriosuchini by previous analyses (*Kammerer et al., 2015*) are also consistently excluded from Mystriosuchini as defined in the current analysis (with *Mystriosuchus planirostris*, *Machaeroprosopus jablonskiae* and *Machaeroprosopus buceros* as exemplars of the clade; see Table 2). Non-Mystriosuchini leptosuchomorphs in this analysis include all members of *Smilosuchus*, *Leptosuchus* and *Nicrosaurus* plus 'Phytosaurus' doughtyi, *Pravusuchus hortus*, *Coburgosuchus goeckeli* and PEFO 34852, as well as *Protome* in the DC and DCM coding treatments (Figs. 5 and 7) (see below).

There is only one synapomorphy of Mystriosuchini common to all trees (Fig. 8, node O) [43: 2→0]. Much like the non-Mystriosuchini leptosuchomorphs, interrelationships within Mystriosuchini are generally inconsistent across the different coding treatments; however, as in previous analyses, the clade includes all named species of *Machaeroprosopus*, 'Redondasaurus' and *Mystriosuchus*, as well as USNM V 17098, NMMNHS P-4256, NMMNHS P-31094, MB.R. 2747 and NHMW 1986 0024 0001.

*Protome batalaria* has been placed close to *Rutiodon* by previous studies (*Stocker, 2012*; *Butler et al., 2014*; *Kammerer et al., 2015*). In this study it is consistently found to be either nested just inside Mystriosuchini (Fig. 4, node 28 and Fig. 6, node 30) or as the sister taxon to this clade (Fig. 5, node 24 and Fig. 7, node 27). In both trees in which *Protome* is recovered within Mystriosuchini (D and DM trees) the node is supported by the presence of 'parietal prongs' [65: 0→1]; additionally, in the DM tree the node is supported by the presence of a small elongate depression on the postorbital bar just posterodorsal to the orbit [29: 0→1], as well as all GM characters. Parietal prongs are exclusive only to *Protome* and members of *Machaeroprosopus* and 'Redondasaurus', whereas the groove on the descending process of the postorbital is common to many taxa throughout Parasuchidae.

Within Mystriosuchini, *Mystriosuchus* (Fig. 8, node P) is the only consistently supported clade. Within this clade MB. R. 2747 and NHMW 1986 0024 0001 form successive sister taxa to *Mystriosuchus planirostris* and *Mystriosuchus westphali*; it is likely that these two specimen-level OTUs also represent unnamed species of *Mystriosuchus*. *Mystriosuchus* and its internal nodes are statistically well supported. The basal node of the clade and the internal nodes are each supported by single synapomorphies common to all trees (Fig. 8, node P) [85: 1→0], (Fig. 8, node Q) [2: 1→2], (Fig. 8, node R) [88: 0→1].

## Conflicting relationships

As discussed above, relationships among the non-Mystriosuchinae taxa are almost entirely consistent across all four trees with the exception of 'Paleorhinus' parvus, 'Paleorhinus' sawini and *Ebrachosuchus neukami* (Fig. 8). The relationships between these taxa are poorly supported statistically and variable, and the three form a polytomy together with

Mystriosuchinae in the strict consensus trees of the D and DC analyses (Figs. 4 and 5). In the DM and DCM analyses (Figs. 6 and 7) the relationships are consistent, if not well supported. 'Paleorhinus' parvus is the sister taxon of Mystriosuchinae, with Ebrachosuchus neukami and 'Paleorhinus' sawini forming successively more distant sister groups.

Those non-Mystriosuchini members of Leptosuchomorpha in this analysis comprise species assigned to the genera Smilosuchus, Leptosuchus, 'Phytosaurus', Pravusuchus, Nicrosaurus and Coburgosuchus. Relationships between these taxa are entirely consistent in the DC and DCM trees (Figs. 5 and 7). However, the D and DM trees each show different topologies (Figs. 4 and 6). In the DC and DCM trees, 'Smilosuchus' lithodendrorum is the most basal taxon in Leptosuchomorpha. Within Leptosuchomorpha there are two clades: one containing all species of Leptosuchus and Nicrosaurus, in addition to Pravusuchus hortus, Coburgosuchus goeckeli and PEFO 34852; and one containing 'Phytosaurus' doughtyi, Smilosuchus adamanensis, Smilosuchus gregorii, Protome and Mystriosuchini.

In the D tree, all the aforementioned taxa with the exception of Smilosuchus gregorii form an unnamed clade (Fig. 4, node 14), which forms a sister relationship within Leptosuchomorpha with Smilosuchus gregorii + Mystriosuchini. The basalmost taxon within this unnamed clade is Smilosuchus adamanensis, which in the other three trees presented here is recovered as a branch just basal to S. gregorii; the next taxon in the clade, 'Phytosaurus' doughtyi, also falls closer to S. gregorii than Leptosuchus in the DC, DM and DCM trees.

Above 'Phytosaurus' doughtyi, two distinct clades are present as sister taxa. One of these (Fig. 4, node 17) contains Leptosuchus spp., plus 'Smilosuchus' lithodendrorum and PEFO 34852; the second (Fig. 4, node 20) contains Pravusuchus hortus, Nicrosaurus spp. and Coburgosuchus goeckeli. Relationships in both clades have weak Bremer support, with the exception of the node uniting Nicrosaurus kapffi, N. meyeri and C. goeckeli (Fig. 4, node 21), in which frequency supports are generally better.

The topology for this region of the DM tree is very different from that of the D tree (to which it is very similar in most other respects). The taxa that form a distinct clade in the D tree (Fig. 4, node 14) instead form a largely pectinate series of outgroups to Mystriosuchini in the DM tree (Fig. 6, nodes 15–25). The most basally branching taxon is Leptosuchus studeri, which falls outside of Leptosuchomorpha in this tree. At the base of Leptosuchomorpha is a relatively poorly supported (according to frequency supports) clade including 'Phytosaurus' doughtyi, Leptosuchus crosbiensis, and a sister taxon relationship between 'Smilosuchus' lithodendrorum and PEFO 34852 (Fig. 6, node 17). Pravusuchus hortus, Smilosuchus adamanensis and Smilosuchus gregorii form a series of outgroups to a clade consisting of Mystriosuchini and the Nicrosaurus + Coburgosuchus clade. In the DM tree the Nicrosaurus species are sister taxa (Fig. 6, node 25). In this topology, Nicrosaurus occupies a position consistent with that recovered in previous analyses of Mystriosuchini (Hungerbühler, 2002; Hungerbühler et al., 2013) and with the group's previous definition (Kammerer et al., 2015).

The main inconsistency within Mystriosuchini is the fluctuating position of the *Mystriosuchus* clade (*Mystriosuchus* spp. plus NHMW 1986 0024 0001 and MB. R. 2747). In both trees incorporating continuously scored data (DC, DCM) this group is recovered as highly derived within Mystriosuchini (Figs. 5 and 7), as has previously been found by *Stocker (2010, 2012, 2013), Butler et al. (2014)* and *Kammerer et al. (2015)* (Figs. 2B, 3A and 3B). In the D and DM coding treatments, however, the *Mystriosuchus* clade forms the sister group to *Protome batalaria + Machaeroprosopus* (Figs. 4 and 6), as has been found by *Hungerbühler (2002), Parker & Irmis (2006)* and *Hungerbühler et al. (2013)* (Figs. 1A, 2A and 2C).

Relationships among other species within Mystriosuchini are highly variable, though the general pattern is of a highly laddered series of sequentially more derived terminals. Although the order of OTUs varies considerably, there are some similarities across different coding treatments; taxa in the less derived positions are generally *Protome batalaria* and *Machaeroprosopus andersoni*, which are then followed by *Machaeroprosopus pristinus*, *Machaeroprosopus buceros* and *Machaeroprosopus lottorum* and then a clade containing both species of '*Redondasaurus*' (Fig. S2).

As previously mentioned, the two conflicting hypotheses regarding the position of *Mystriosuchus* (basal or derived within Mystriosuchini) split the results of the four coding methods into two alternative topological hypotheses (respectively D, DM (Figs. 4 and 6) and DC, DCM (Figs. 5 and 7)). The positions of *Machaeroprosopus mccauleyi* and *Machaeroprosopus jablonskiae* also consistently differ between these topologies. In the trees in which *Mystriosuchus* occupies a derived position within Mystriosuchini (DC, DCM), *Machaeroprosopus mccauleyi* and *Machaeroprosopus jablonskiae* form successive sister taxa, basal to the clade comprising *Machaeroprosopus pristinus*, *Machaeroprosopus buceros* and *Mystriosuchus*. In topologies where *Mystriosuchus* is recovered basal to *Machaeroprosopus* (D, DM), *Machaeroprosopus mccauleyi* and *Machaeroprosopus jablonskiae* are more derived than the clade composed of *Machaeroprosopus pristinus*, *Machaeroprosopus buceros* and *Machaeroprosopus lottorum*, forming successive sister taxa to '*Redondasaurus*'.

The position of *Machaeroprosopus zunii* is more consistent; in three trees (D, DC and DM) it is recovered basal to the clade composed of *Machaeroprosopus pristinus*, *Machaeroprosopus buceros*, and all more derived taxa. In the DCM results *Machaeroprosopus zunii* is placed more derived than *Machaeroprosopus pristinus* and *Machaeroprosopus buceros*, but less derived than *Machaeroprosopus lottorum* (Fig. 7, node 34).

*Machaeroprosopus lottorum* is another taxon which varies consistently between the two broad topological hypotheses presented. In the trees incorporating continuously scored data, in which *Mystriosuchus* is highly derived (DC, DCM), *Machaeroprosopus lottorum* forms a clade with NMMNHS P-31094 (Figs. 5 and 7), closely related to '*Redondasaurus*' and *Mystriosuchus*, as was found by *Hungerbühler et al. (2013)*. In the alternative topologies (D, DM) *Machaeroprosopus lottorum* nests with *Machaeroprosopus pristinus*, to the exclusion of *Machaeroprosopus buceros* (Figs. 4 and 6). Both positions are similarly poorly supported by Bremer analyses, but possess relatively good

**Table 7 Bremer (above) and frequency (below) supports resulting from each of the four data conditions.**

| | Discrete (D) | Discrete + continuous (DC) | Discrete + morphometric (DM) | Discrete + continuous + morphometric (DCM) | |
|---|---|---|---|---|---|
| Number of nodes retained | 15 [14] | 17 [**15**] | **23** [13] | 15 [12] | <0.08 [<0.11] Bremer collapsed |
| Total Bremer support | 3.34 [3.25] | 3.59 [**3.40**] | **3.86** [3.04] | 2.86 [2.59] | |
| Mean Bremer support | **0.22** [0.23] | 0.21 [**0.23**] | 0.17 [**0.23**] | 0.19 [0.22] | |
| Number of nodes retained | 29 | **30** | 29 | 26 | <10 Frequency collapsed |
| Total frequency support | 1,337 | **1,416** | 1,305 | 1,207 | |
| Mean frequency support | 46.10 | **47.20** | 45.00 | 46.42 | |

Note:
Values below which nodes were collapsed, were set as 0.08 and [0.11] for Bremer, and 10 for frequency supports. Largest values in each category are shown in bold.

**Table 8 Mean Bremer supports calculated in five tree-regions within each of the four data conditions.**

| | Discrete (D) | Discrete + continuous (DC) | Discrete + morphometric (DM) | Discrete + continuous + morphometric (DCM) |
|---|---|---|---|---|
| Region 1 Bremer mean | 0.15 | **0.16** | 0.09 | 0.11 |
| Region 2 Bremer mean | **0.23** | **0.23** | 0.20 | 0.17 |
| Region 3 Bremer mean | 0.06 | 0.07 | **0.10** | 0.09 |
| Region 4 Bremer mean | 0.06 | 0.05 | **0.07** | 0.05 |
| Region 5 Bremer mean | 0.14 | **0.24** | 0.14 | 0.16 |
| Total | 0.64 | **0.75** | 0.60 | 0.58 |

Note:
Largest values in each region are shown in bold. The data condition with the highest summed mean Bremer support is highlighted with a border.

frequency scores. In this topology NMMNHS P-31094 is consistently found within 'Redondasaurus', as the sister taxon of 'Redondasaurus' gregorii, to the exclusion of 'Redondasaurus' bermani.

## Accuracy and validity

### Bremer supports

With poorly supported nodes collapsed below Bremer values of 0.08, the DM condition produced greatest tree resolution, retaining 23 nodes; however, its mean Bremer score is one of the lowest among the four trees, suggesting that the additional nodes supported in this tree only exceed the cut-off by a small amount (Table 7). When using the mean step length of a single character (0.11) as a cut-off for node-collapsing, the DM and DCM conditions were found to perform more poorly than the D and DC conditions in terms of nodes retained and total Bremer support. Mean Bremer values for the retained nodes remained almost consistent across all trees (Table 7).

When broken down into regions, it appears that the extra support in the DM tree is added in regions three and four, which are almost consistently the worst-supported regions in all trees. Despite this extra support, relationships within these regions are still relatively poorly supported in the DM condition, and the support for region one also becomes among the poorest in both GM trees (DM and DCM) (Table 8).

The best condition for overall Bremer support was the DC tree (Fig. 5), achieving the highest, or equal highest support in all regions except three and four, with a sum of

**Table 9 Mean frequency supports calculated in five tree-regions within each of the four data conditions.**

|  | Discrete (D) | Discrete + continuous (DC) | Discrete + morphometric (DM) | Discrete + continuous + morphometric (DCM) |
|---|---|---|---|---|
| Region 1 freq. mean | 41.00 | **48.80** | 37.86 | 39.86 |
| Region 2 freq. mean | **73.83** | 73.33 | 72.00 | 67.00 |
| Region 3 freq. mean | 24.73 | **27.91** | 20.55 | 15.00 |
| Region 4 freq. mean | 18.91 | 17.85 | **20.75** | 14.92 |
| Region 5 freq. mean | 71.00 | **72.67** | 62.00 | 62.33 |
| Total | 229.47 | **240.56** | 213.16 | 199.11 |

Note:
  Largest values in each region are shown in bold. The data condition with the highest summed mean frequency support is highlighted with a border.

mean support equalling 0.75. Conversely, despite maximizing support in the poorest regions of the tree, the DM condition scored second worst for overall support, with a sum total of 0.60; this was followed by the DCM condition with a score of 0.58 (Table 8).

### Frequency supports

With a cut-off for node-collapsing of <10, symmetric frequency support produced broadly similar results for all the trees, with the DC condition producing a marginally higher resolution and mean support value. Conversely to the results from Bremer supports, the DM condition was the poorest supported topology based on symmetric resampling, although the difference between 'best' and 'worst' is minor (Table 7).

Split into regions, the overall sum of mean supports follows the same trend as that of the Bremer supports; DC is best, with a sum of 240.56, then D (229.47), DM (213.16) and finally DCM (199.11). The DC tree holds the highest mean support compared to the other trees in regions one, three and five. The DM tree only holds the highest support value in region four; however, this is one of the two poorest supported regions (three and four), and is therefore important in achieving the best possible resolution in all parts of the tree (Table 9).

### Stratigraphic congruence

All tree topologies recovered under the four data conditions tested in this analysis were found to be significantly better correlated with stratigraphy than would be expected of random data. Among the raw results from each correlation metric, there is no consistent trend indicating one or more of the four topologies are optimal. The SCI metric suggests the D and DM topologies (in which *Mystriosuchus* is basal to *Machaeroprosopus*) to be better stratigraphically correlated than the DC and DCM topologies (in which *Mystriosuchus* is the most derived member of Mystriosuchini); however, this finding is not borne out by any other metric. Among the other three metrics the only consistent trend is the slightly worse performance of the two datasets incorporating GM characters (DM and DCM); however, the difference in fit is almost negligible (Table 10).

The previous phylogenetic analyses of *Parker & Irmis (2006)* (Fig. 2A) and *Kammerer et al. (2015)* (Fig. 3B) (based respectively on the original matrices of *Hungerbühler (2002)* and *Stocker (2010)*), also correlate well with the stratigraphic data used in this study,

**Table 10 Stratigraphic consistency metrics for each of the four tree topologies, compared with those of three previous phylogenetic analyses of ingroup phytosaur relationships.**

|  | SCI | RCI | GER | MSM* | P. Sig. SCI | P. Sig. RCI | P. Sig. GER | P. Sig. MSM* |
|---|---|---|---|---|---|---|---|---|
| D | 0.737 | −9.272 | 0.777 | 0.150 | <0.0001 | <0.0001 | <0.0001 | <0.0001 |
| DC | 0.615 | −9.369 | 0.777 | 0.150 | <0.0001 | <0.0001 | <0.0001 | <0.0001 |
| DM | 0.732 | −10.343 | 0.775 | 0.149 | <0.0001 | <0.0001 | <0.0001 | <0.0001 |
| DCM | 0.634 | −11.032 | 0.773 | 0.148 | <0.0001 | <0.0001 | <0.0001 | <0.0001 |
| *Kammerer et al. (2015)* | 0.750 | −16.042 | 0.805 | 0.291 | <0.0001 | – | 0.0002 | <0.0001 |
| *Hungerbühler et al. (2013)* | 0.400 | 19.878 | 0.163 | 0.258 | 0.20 | – | 0.96 | 0.77 |
| *Hungerbühler et al. (2013)* | 0.429 | 25.841 | 0.247 | 0.278 | 0.08 | – | 0.81 | 0.58 |
| *Parker & Irmis (2006)* | 0.700 | −57.830 | 0.797 | 0.389 | <0.0001 | – | <0.0001 | <0.0001 |

Note:
   Raw analysis output values are displayed on the left and significance values generated via random permutations are presented on the right.

generally achieving significance values equal to those of the current study. The topologies of *Hungerbühler et al. (2013)* (Fig. 2C) were found to correlate poorly with stratigraphy and were not statistically differentiable from random data; however, the analysis of *Hungerbühler et al. (2013)* focuses only on one area of the tree, roughly corresponding to 'region four' in this study. This region is poorly supported in terms of accuracy and robusticity. The poor stratigraphic correlation of the analysis of *Hungerbühler et al. (2013)* may indicate that this region has poor stratigraphic support, but this is masked in the stratigraphic correlations of other studies by good correlation overall in other areas of the tree.

## Tree choice

In order to carry out further investigations into the effects of alternative, or previously reported topologies, it was decided to select only two of the four topologies presented above to avoid unnecessarily long comparisons of fit between multiple alternative taxonomic relationships within multiple tree topologies. As there is a general dichotomy within the four trees, it would be arbitrary to favour one topology over the other, so a representative of each was chosen.

   The DC condition (discrete + continuous characters (Fig. 5)) exhibits an almost identical topology to the DCM condition (discrete + continuous + GM characters (Fig. 7)), but consistently outperforms the latter in the various robusticity analyses described above. Comparisons of topological similarity do not assist in selecting one of these topologies over the other as they are shown to be almost identical, with neither being more representative of all topologies.

   The D and DM conditions (respectively discrete characters and discrete + GM characters (Figs. 4 and 6)) are less similar to each other than are the DC and DCM conditions, though they show largely the same topology. Between the Bremer and frequency analyses the D and DM conditions outperform each other in various aspects; when the trees are regionalized the DM condition generally provides slightly better support in the worst-supported areas of the tree, but is poorly supported in most other areas. The sum of RF distances for the D tree in comparison to all others suggests

that it is the most representative topology of the four trees recovered in this study; this was never found to be the case with the DM topology.

Ultimately the D and DC trees (Figs. 4 and 5) were selected for further analysis based partially on the above metrics, but partially due to the relative difficulty of undertaking multiple further GM analyses. Continuous and discrete characters boast substantial advantages in analysis duration, and the comparative simplicity of data acquisition and processing, over GM characters. Because of these reasons continuous and discrete data are far more accessible and provide a better basis on which future studies can build.

## Alternative taxonomic relationships

The consistent recovery of a sister-relationship between *Rutiodon carolinensis* and the genus *Angistorhinus* makes the decision of whether or not to synonymize these taxa entirely arbitrary (see below); therefore, to test for their synonymy would also be meaningless and as such these taxa were excluded from these analyses.

*Nicrosaurus* was previously found as the basal-most member of Mystriosuchini (*Hungerbühler, 2002*; *Hungerbühler et al., 2013*) and was therefore used as an internal specifier for the most recent phylogenetic definitions of the clade, preceding this study (*Parker & Irmis, 2006*; *Kammerer et al., 2015*); however, as described in the introduction little data has been provided to support this. Here, we find *Nicrosaurus* to group closer to *Leptosuchus* than to *Mystriosuchus* or *Machaeroprosopus*, and thus outside of Mystriosuchini according to our redefinition of the clade (Table 2).

We tested the previously proposed position of *Nicrosaurus*, that is, as the most basal group within Mystriosuchini (*Kammerer et al., 2015*). To achieve this, the clade of *Nicrosaurus* and *Coburgosuchus* was constrained to its previous position in relation to Mystriosuchini, such that all members of *Machaeroprosopus* and *Mystriosuchus* fell in more derived positions. Additionally, *Pravusuchus hortus* was constrained as the basal sister taxon to *Nicrosaurus*, *Coburgosuchus* and Mystriosuchini, to replicate the previous hypothesis that *Pravusuchus* is the immediate sister taxon to Mystriosuchini (*Stocker, 2010*). Under these topological constraints tree character fit worsened by 0.693 in the D condition, and 1.013 in the DC condition.

The tree topology resulting from the D condition places *Mystriosuchus* as the sister clade to *Machaeroprosopus*; for this analysis we constrained *Mystriosuchus* to nest within *Machaeroprosopus* as found by *Stocker (2010)*, although its exact position within the clade was left flexible. Under this condition the tree-fit worsens by 0.584. In contrast, in the DC condition *Mystriosuchus* was found to occupy a position within the *Machaeroprosopus* clade; therefore, we constrained it as sister to this clade, leading to a decline in tree fit by 0.714.

Unlike the findings of *Hungerbühler et al. (2013)*, in our phylogenies the two species of 'Redondasaurus' do appear to form a sister taxon relationship; however, in accordance with their findings and those of other studies (*Ballew, 1989*; *Hungerbühler, Chatterjee & Cunningham, 2003*; *Stocker, 2010*; *Butler et al., 2014*; *Kammerer et al., 2015*) 'Redondasaurus' remains nested within *Machaeroprosopus*. When the two genera are

forced into a sister group relationship the tree fit deteriorated considerably by a score of 0.857 under the D condition, and 1.004 in the DC condition.

## DISCUSSION

### Higher-level taxonomy

The recently revived family-level name Parasuchidae *Lydekker, 1885* (*Kammerer et al., 2015*) was suggested by *Stocker et al. (2017)* to exclude the proposed basal phytosaur *Diandongosuchus fuyuanensis*. Our analysis corroborates the hypothesis of *Stocker et al. (2017)* that *Diandongosuchus* is the most basal phytosaur, and the only taxon to fall outside of Parasuchidae but within Phytosauria using current definitions (Fig. 8).

The taxonomic content of Mystriosuchinae *von Huene, 1915*, defined as the last common ancestor of *Angistorhinus grandis* and *Mystriosuchus planirostris* and all its descendants by *Kammerer et al. (2015)* (Table 2), is largely compatible between the phylogenetic hypotheses presented here and that presented by *Kammerer et al. (2015)*. However, in the phylogeny of *Kammerer et al. (2015)* (Fig. 3B) '*Paleorhinus*' *sawini* falls within Mystriosuchinae whereas here it is excluded from this clade.

*Stocker (2010)* erected the clade Leptosuchomorpha, defined as the most recent common ancestor of *Leptosuchus studeri* and *Machaeroprosopus pristinus*, and all descendants thereof (Table 2). In the D and DM trees presented here (Figs. 4 and 6) this definition is perfectly compatible with previous definitions of the clade; however, in the DC and DCM conditions (Figs. 5 and 7) '*Smilosuchus*' *lithodendrorum* is recovered in a more basal position than all other previous members of Leptosuchomorpha, and would thus be excluded from the group based on the definition of *Stocker (2010)*, despite exhibiting numerous similarities with other members. We therefore redefine Leptosuchomorpha such that it includes the latest common ancestor of '*Smilosuchus*' *lithodendrorum*, *Leptosuchus studeri* and *Machaeroprosopus pristinus*, and all of its descendants (Table 2). In addition, *Protome batalaria* and '*Machaeroprosopus*' *zunii* are consistently recovered within Leptosuchomorpha in the analyses presented here, whereas they were previously excluded (*Stocker, 2010*; *Butler et al., 2014*; *Kammerer et al., 2015*).

The definition of Mystriosuchini *von Huene, 1915* proposed by *Kammerer et al. (2015)* (Table 2) is problematic with regard to the results presented here, due to our general result that *Nicrosaurus* is deeply nested with taxa such as *Leptosuchus* and *Smilosuchus* that are traditionally excluded from Mystriosuchini. This problem is especially pronounced in the D tree (Fig. 4), in which the previous definition of Mystriosuchini renders the group entirely synonymous with Leptosuchomorpha; the DC and DCM trees produce a very similar result, though excluding '*Smilosuchus*' *lithodendrorum* from Mystriosuchini (Figs. 5 and 7). In the DM tree (Fig. 6) the taxonomic content of Mystriosuchini using the previous phylogenetic definition is essentially the same as in previous studies, with the inclusion of a few additional taxa such as *Protome batalaria*.

To resolve this taxonomic issue, we propose that *Nicrosaurus kapffi* is removed from the definition of Mystriosuchini due to its conflicting phylogenetic position, and is replaced with *Machaeroprosopus jablonskiae* to stabilize the taxonomic content of the clade

(see above; Table 2). Without the addition of *Machaeroprosopus jablonskiae* as a specifier, *Machaeroprosopus mccauleyi* and *Machaeroprosopus jablonskiae* would be variably excluded from Mystriosuchini, despite consistent previous findings of their inclusion in the clade. A number of other taxa would also be variably included in Mystriosuchini, leading to increased instability of the clade.

*Machaeroprosopus jablonskiae* is recovered in a similar position to that found by previous phylogenetic analyses (*Parker & Irmis, 2006*; *Hungerbühler et al., 2013*; *Butler et al., 2014*; *Kammerer et al., 2015*) in all of our trees. In the DC and DCM trees *Machaeroprosopus jablonskiae* is recovered as one of the most basal taxa within *Machaeroprosopus* (Figs. 5 and 7), as in the studies of *Parker & Irmis (2006)* and *Hungerbühler et al. (2013)* (Figs. 2A and 2C). In the D and DM trees *Machaeroprosopus jablonskiae* is placed in a more derived position in the *Machaeroprosopus* clade (Figs. 4 and 6), similar to the findings of *Butler et al. (2014)* and *Kammerer et al. (2015)* (Figs. 3A and 3B); however, as this coincides with the migration of *Mystriosuchus* to a more basal position with respect to *Machaeroprosopus*, the taxa retained in Mystriosuchini remain largely identical among our four trees. Crucially, *Machaeroprosopus jablonskiae* consistently nests within Mystriosuchini in previous studies (*Parker & Irmis, 2006*; *Hungerbühler et al., 2013*; *Butler et al., 2014*; *Kammerer et al., 2015*), and in this sense our proposed definition errs on the side of caution in ensuring the definition of Mystriosuchini used here is as compatible as possible with the phylogenetic topologies of previous studies.

This being the first investigation of this dataset, it seems likely that future analyses of this data could disagree with our findings, in which case a definition that maximizes compatibility between recent studies may be the most useful. We therefore tentatively suggest Mystriosuchini should henceforth be defined as the most recent common ancestor of *Mystriosuchus planirostris*, *Machaeroprosopus jablonskiae* and *Machaeroprosopus buceros*, and all its common ancestors (Table 2).

## Lower-level taxonomy

### Synonymy of Rutiodon and Angistorhinus

The results of this analysis depart from both previously proposed hypotheses of the relative phylogenetic positions of these taxa. *Hungerbühler & Sues (2001)* found *Rutiodon* to occupy a derived position within the monophyletic clade of *Angistorhinus*, whereas other studies recovered *Rutiodon* in a more derived position than *Angistorhinus*, closer to *Leptosuchus* (*Hungerbühler, 1998*; *Stocker, 2010*). Supporting character data were not provided for the proposal of synonymy made by *Hungerbühler & Sues (2001)*, which was published in an abstract only. In our results the two taxa form a monophyletic group, supported by two synapomorphies common to all four best-fit trees [22: $2 \rightarrow 1$; 92: $0 \rightarrow 1$]. However, the fact that *Rutiodon* consistently forms the sister group to *Angistorhinus* makes the decision of whether or not to synonymize the genera entirely arbitrary. Unfortunately, we were unable to study any material of *Brachysuchus megalodon*, which has been suggested to be synonymous with *Angistorhinus* (*Long & Murry, 1995*), but which was also found to be distinct by *Stocker (2010)*.

### Angistorhinus

In her discussion of the relationships of *Angistorhinus*, *Stocker (2010)* advocated the necessity for future in-depth analysis of *Angistorhinus* and its affinities. We do not present a detailed analysis or redescription of any species within *Angistorhinus*; however, our analysis is only the second to include more than one species (*Hungerbühler, 1998*), and the first to incorporate further specimens that have been identified previously as *Angistorhinus*. Our results provide a stable and consistently well-supported phylogenetic position for *Angistorhinus* that future descriptive and taxonomic work can build on. Furthermore, we provide additional synapomorphies for both the *Angistorhinus* clade, and relationships within it.

The *Angistorhinus* clade (Figs. 4 and 5, node 9; Figs. 6 and 7, node 11) is distinguished by two unambiguous synapomorphies common to all trees, pertaining to the parietal/squamosal bars being medially convex, and at least as wide as the postorbital/squamosal bars [56: 0→1; 58: 0→1]. Both of these characters have previously been suggested to be diagnostic features of *Angistorhinus Mehl, 1913* (*Mehl, 1915*; *Gregory, 1962a*; *Stocker, 2010*) or 'Angistorhininae' *Camp, 1930* (*Long & Murry, 1995*).

The next most inclusive clade contains *Angistorhinus talainti*, *A. grandis*, TMM 31100-1332 and USNM V 21376. This group is distinguished by the presence of a sulcus running longitudinally along the postorbital/squamosal bar [42: 0→1], and the partial or total squaring of the medial rim of the postorbital/squamosal bar and posterior process [51: 0→1].

The next most inclusive clade excludes *A. talainti*, leaving only *A. grandis*, TMM 31100-1332 and USNM V 21376. This clade is well supported by four unambiguous synapomorphies, though within the clade the basal-most member (TMM 31100-1332) shows no autapomorphies and the sister grouping of *A. grandis* with USNM V 21376 is supported by only one synapomorphy [69: 2→1] and displays poor support values. Given the strong support for the wider clade, but the relatively poor differentiation of the OTUs within it, there may be a case for referring both TMM 31100-1332 and USNM V 21376 to *A. grandis*. The synapomorphies of this clade are: the division of the narial openings into an anterior 'anteriorly opening' section and a posterior 'dorsally opening' section [12: 0→1]; the raising of the external nares above the level of the skull roof [17: 0→1]; the posttemporal fenestra being moderately wide and dorsoventrally compressed [66: 0→1]; and the presence of an anteroposteriorly oriented ridge on the midline of the basioccipital between the basitubera [70: 0→1].

Based on these results we suggest *A. grandis* to be one of the most derived members of *Angistorhinus*, and *A. talainti* to be less derived. At face value, there does not appear to be any clear relationship between palaeogeography and phylogeny; *A. talainti*, from Morocco, nests amongst the specimens known from the west and south central USA. This finding should be expected as these locations were placed at broadly similar palaeolatitudes and were closely connected in the Late Triassic.

### Monophyly of Leptosuchus

*Stocker (2010)* found a strongly supported monophyletic relationship between *Leptosuchus crosbiensis* and *Leptosuchus studeri*; in our analysis, we found almost all nodes

relating to *Leptosuchus*-grade taxa were extremely poorly supported in each tree. Only in the D tree did we find an arrangement approaching a monophyletic *Leptosuchus* (Fig. 4, node 17), though with the addition of 'Smilosuchus' *lithodendrorum* and PEFO 34852 as a sister clade to *L. crosbiensis*. In the DC and DCM trees *Leptosuchus studeri* forms the sister group to a clade containing *Leptosuchus crosbiensis*, but also *Pravusuchus*, *Coburgosuchus* and *Nicrosaurus* (Fig. 5, node 15 and Fig. 7, node 17). Support values are generally poor. In the DM tree *Leptosuchus*-grade taxa occur as a paraphyletic grade of sequentially more derived branches (Fig. 6).

*Stocker (2010)* found one synapomorphy to support the monophyly of *Leptosuchus* and one further potential apomorphy under DELTRAN optimization.

*Distal end of paroccipital process of opisthotic rounded, distal edge is curved rather than straight (36: 1→2).* This character was excluded from analysis here as the associated morphology appears to be variable both inter- and intraspecifically, is often subject to damage, and scoring may change depending on small differences in viewing angle.

*Jugal contributing to antorbital fenestra (4: 0→1) (potential apomorphy under DELTRAN).* In our analysis this character state is optimized as basal to the entire tree, and is found in the vast majority of taxa. In this position the character does not provide unambiguous support for the monophyly of *Leptosuchus*.

### Monophyly of Smilosuchus

The previously proposed taxonomic content of *Smilosuchus* is not monophyletic in any of our best-fit trees. In the D tree (Fig. 4) all three species are found in different locations: *S. adamanensis* forms the basal-most taxon in a clade containing all leptosuchomorph taxa excluded from Mystriosuchini except *S. gregorii* (Fig. 4, node 14); *S. lithodendrorum* is deeply nested within this group, forming a close relationship with *Leptosuchus crosbiensis* (Fig. 4, node 18); *S. gregorii* forms its own distinct branch forming a sister relationship with Mystriosuchini (Fig. 4, node 23).

In none of the trees presented here does 'Smilosuchus' *lithodendrorum* form a close relationship with any other member of *Smilosuchus*. Instead, its relationships are divergent, being recovered in two trees as the most basal member of the newly defined Leptosuchomorpha (DC and DCM; Figs. 5 and 7) and in the other two nesting closely with *Leptosuchus crosbiensis* (D and DM; Figs. 4 and 6). The similarity to *Leptosuchus crosbiensis* has previously been noticed, leading *Long & Murry (1995)* to regard *S. lithodendrorum* as a junior synonym of the former taxon, though without a written justification (see Appendix 1 for more details). We do not here revise the taxonomy of *S. lithodendrorum*, as the instability of its position does not allow any consistent hypothesis of its relationships to be reached. Instead, we consider the phylogenetic position of this taxon as uncertain pending a more detailed investigation into its similarity to *L. crosbiensis*.

In *Stocker's (2010)* analysis, the monophyly of *Smilosuchus* was supported on the basis of two synapomorphies and a further possible apomorphy under ACCTRAN optimization.

*Ventral margin of squamosal gently sloping anteroventrally from posterior edge of posterior process to opisthotic process (28: 1→0).* In contrast to the scorings of *Stocker (2010)*, we found no specimen of *S. lithodendrorum* with a gently sloping posteroventral squamosal

margin. This state was, however, found to be present in both *S. adamanensis* and *S. gregorii*. The latter taxon displays polymorphism for this character as AMNH FR 3060 displays a morphology that is neither a gentle slope, nor a sharp shelf, but sits somewhere between.

In the D and DM trees (Figs. 4 and 6) *S. adamanensis* and *S. gregorii* apparently gain this character state (0) independently, though because the latter taxon is polymorphic for this character, the ancestral state (1) is partially retained. In the DC and DCM trees (Figs. 5 and 7) the ancestral state is polymorphic; therefore, depending on the tree in question this character is either partially consistent or inconsistent with the hypothesis of monophyly between *S. adamanensis* and *S. gregorii*.

Interestingly, if *S. gregorii* is scored as '0' rather than as polymorphic, both taxa consistently form a monophyly in the D tree, whereas they were previously relatively distant phylogenetically from each other. This was also tested in the DC tree (which shares the same relative phylogenetic positions of *S. adamanensis* and *S. gregorii* as in the DM and DCM trees). However, the phylogenetic positions of these two taxa were not modified, and state '0' was also reconstructed as ancestral to the clade including *Protome batalaria* and Mystriosuchini.

*Squamosal fossa extends to posterior edge of squamosal (30: 1→0).* The scores for this character in the current analysis are inconsistent with those of *Stocker (2010)*; we observed a polymorphic state in both *S. lithodendrorum* (TMM 31173-121: 0; UCMP 26688: 1) and *S. gregorii* (UCMP 27200: 0; AMNH FR 3060: 1). Our character optimization is inconsistent with the hypothesis of a monophyletic *Smilosuchus*, given that character state '0' is ancestral to the majority of taxa (excluding many basal taxa for which the character is inapplicable and most species of *Machaeroprosopus*) in all four of our trees.

*Lateral border of posttemporal fenestra formed by the contact of the parietal process of the squamosal and the paroccipital process of the opisthotic (37: 1→0) (potential apomorphy under ACCTRAN).* Our scoring for this character differs from that of *Stocker (2010)*; we concur that *S. lithodendrorum* displays state '0', whereas both *S. adamanensis* and *S. gregorii* are scored as possessing a thin lamina of squamosal that slightly undercuts the border of the fenestra ventrolaterally (character state '2'). The latter condition is ancestral to both species of *S. adamanensis* and *S. gregorii*, all species of *Machaeroprosopus* and closely related taxa in all four trees (though in the D tree the ancestral state is polymorphic '0, 2'). In trees D, DC and DCM character state '2' independently characterizes the clade formed by *Nicrosaurus* and *Coburgosuchus*. Character state '0' is the ancestral condition for '*Smilosuchus*' *lithodendrorum* in all four trees presented here. None of the optimizations of this character presented here support the monophyly of *Smilosuchus*.

### Position of Pravusuchus hortus

*Pravusuchus hortus* has previously been indirectly implied to potentially form a close relationship with *Nicrosaurus*: *Pravusuchus* was found to form the immediate outgroup to Mystriosuchini by *Stocker (2010)*, while *Nicrosaurus* has long been hypothesized to form a close relationship with *Mystriosuchus* and *Machaeroprosopus* (*Ballew, 1989*) as the most basal taxon within Mystriosuchini (*Long & Murry, 1995*; *Hungerbühler, 2002*;

*Parker & Irmis, 2006*; *Hungerbühler et al., 2013*). Thus our a priori assumption was that these taxa would be closely related. Our results corroborate this view, with *Pravusuchus* forming the outgroup to a clade containing *Nicrosaurus* and *Coburgosuchus* in three of the four analyses (D, DC, DCM); however, these taxa are found here to nest deeply within a clade of non-Mystriosuchini leptosuchomorph taxa in all but the DM analysis.

The analysis of *Stocker (2010)* identified a single synapomorphy in support of a clade containing *Pravusuchus*, *Machaeroprosopus mccauleyi*, *Machaeroprosopus pristinus* and *Mystriosuchus westphali*. In the three trees in which *Pravusuchus* is the immediate outgroup of *Nicrosaurus* we found two consistent synapomorphies supporting the clade of *Pravusuchus, Nicrosaurus* and *Coburgosuchus*: presence of an infranasal recess, and absence of a furrow or ridge on the lateral surface of the squamosal/post-orbital bar [21: 0→1; 29: 1→0]. The synapomorphy identified by *Stocker (2010)* is discussed below.

*Subsidiary opisthotic process of the squamosal present (29: 0→1).* Our scores for this character are partially inconsistent with those of *Stocker (2010)*; we found *Pravusuchus* to be polymorphic for this character (PEFO 31218: 0 (however an alternate view to that presented here is that the absence may be attributable to poor preservation (William Parker, personal communication to Andrew S. Jones, 2018)); AMNH FR 30646: 1), as was the case in *Machaeroprosopus mccauleyi* (UCMP 126999: 0; PEFO 31219: 1), *Machaeroprosopus pristinus* (PEFO 382: 0; MU 525: 1; AMNH FR 7222: 1; NMMNHS P-50040: 1) and *Mystriosuchus westphali* (AMNH FR 10644: 0; GPIT 261/001: 1).

In all four trees presented here, the most exclusive clade that contains *Pravusuchus* is not supported by the synapomorphy of *Stocker (2010)*; instead, character optimization finds the absence of the subsidiary opisthotic process [47: 0] to be symplesiomorphic for this clade. Here, we find that the presence of a subsidiary opisthotic process of the squamosal [47: 1] primarily optimizes in two alternative positions depending on tree topology. In the D and DM trees (in which *Mystriosuchus* is basal within Mystriosuchini) (Figs. 4 and 6), the presence of this character is a synapomorphy of the clade formed by USNM V 17098 and all more derived taxa. This clade includes *Machaeroprosopus mccauleyi* and *Machaeroprosopus pristinus*, but excludes *Mystriosuchus westphali* and *Pravusuchus hortus*. Therefore in these topologies, this synapomorphy is partially consistent with the aforementioned clade of *Stocker (2010)*, though fundamentally excludes *Pravusuchus* and therefore does not provide support for its position in our trees.

In the DC and DCM trees (in which *Mystriosuchus westphali* occupies a more derived position within the *Machaeroprosopus* clade) (Figs. 5 and 7), the presence of a subsidiary opisthotic process is optimized as polymorphic for the clade that includes *Machaeroprosopus mccauleyi* and all more derived taxa (including *Machaeroprosopus pristinus* and *Mystriosuchus westphali*, but excluding *Pravusuchus*). At the node one step more derived, (thus excluding *Machaeroprosopus mccauleyi*) the character is optimized as 'present' (1) however cannot be regarded as a synapomorphy due to the uncertain optimization of the previous node. This is also partially consistent with the optimization of this character by *Stocker (2010)*; however, the clade supported by this character state excludes *Pravusuchus*, and is inconsistent with Stocker's phylogenetic hypothesis.

### Position of Nicrosaurus

The most recent novel cladistic analysis to investigate the position of *Nicrosaurus* was that of *Hungerbühler (2002)*. The analysis found *Nicrosaurus* as the sister taxon to a clade formed by *Mystriosuchus*, 'Redondasaurus' and *Machaeroprosopus*—congruent with the later definition of Mystriosuchini by *Kammerer et al. (2015)*; however, no synapomorphies were reported in support of this clade.

In three of the four trees identified in this study (D, DC, DCM) *Nicrosaurus* groups more closely with a number of non-Mystriosuchini leptosuchomorph taxa than with Mystriosuchini. *Nicrosaurus* differs from Mystriosuchini in all trees due to the possession of a relatively long free-section of the postorbital/squamosal bar, rather than a short bar as is synapomorphic for the latter clade [43: 2→0] (although *Nicrosaurus meyeri* independently acquires a short postorbital/squamosal bar). Character optimization suggests that the relatively long 'free-section' is plesiomorphic to almost all phytosaurs. This character therefore provides no support for the hypothesized position of *Nicrosaurus* suggested by *Hungerbühler (2002)*.

### Position of Mystriosuchus

The dichotomy of topologies regarding the position of *Mystriosuchus*, as presented in the results section, reflects the dichotomy seen in the literature. The two most recent hypotheses of the position of *Mystriosuchus*, based on independent datasets, are those of *Hungerbühler (2002)* and *Stocker (2010)*, which respectively place *Mystriosuchus* in the less and more derived positions found in this analysis.

### Less derived position

In the analysis of *Hungerbühler (2002)*, the clade in which *Mystriosuchus* is the basal member is diagnosed with three synapomorphies.

*Presence of a pre-infratemporal shelf (18: 1)*. We find this character in three trees (D, DC, DCM) (Figs. 4, 5 and 7) to be a synapomorphy of the clade containing *Mystriosuchus*, 'Redondasaurus' and many members of *Machaeroprosopus*—generally matching the clade membership of Mystriosuchini as it was previously defined in both *Hungerbühler (2002)* and *Stocker (2010)*. This character is therefore largely unaffected by the placement of *Mystriosuchus*, and thus supports both hypotheses.

Presence of the pre-infratemporal shelf is restricted in our analysis almost exclusively to the clade discussed above, however, this character state independently arises as a polymorphic state in *Nicrosaurus* and *Pravusuchus*, and also in *Parasuchus hislopi*.

*Presence of a parietal ledge (21: 2)*. This character was not included in this analysis as the morphology described is dependent on the morphology of the depressed squamosal processes of the parietal, which is scored elsewhere (character 75). The morphology of this area of the skull is partially considered in character 74, which scores the ratio of width to length of the parietals between the supratemporal fenestrae. Regardless, this morphology appears to be present in all leptosuchomorph phytosaurs, and would thus be unlikely to support the clade detailed above.

*Parieto/squamosal bar is strongly depressed (23: 2).* We find this character to be synapomorphic for a more inclusive group than that of *Hungerbühler (2002)*, consisting of *Smilosuchus gregorii*, *Mystriosuchus planirostris*, their common ancestor and all its descendants [49: 1→2]. In three of the trees presented here (D, DC and DCM) (Figs. 4, 5 and 7) this transformation independently occurs in *Nicrosaurus* and *Coburgosuchus*, whereas in the DM tree *Nicrosaurus* and *Coburgosuchus* are included in the clade described above. This character distribution therefore is not found here to support the clade described by *Hungerbühler (2002)*.

No synapomorphies were listed by *Hungerbühler (2002)* for the clade from which *Mystriosuchus* was immediately excluded; therefore, we are unable to comment of the consistency of our synapomorphies with those of *Hungerbühler (2002)*, for a clade containing *Machaeroprosopus* and 'Redondasaurus' but excluding *Mystriosuchus*. The characters supporting this phylogenetic arrangement in our study are detailed in the results section.

### More derived position

*Stocker (2010)* identified eight synapomorphies (and two potential synapomorphies under ACCTRAN) supporting a clade consisting of *Machaeroprosopus mccauleyi*, *Machaeroprosopus pristinus* and *Mystriosuchus westphali*, which, in her analysis, represented Mystriosuchini.

*Interpremaxillary fossa present—narrow slit (8: 1→2).* Here this character state is restricted only to *Mystriosuchus* and NHMW 1986 0024 0001, which consistently sits within the same clade as *Mystriosuchus* (and probably represents an unnamed species within this this genus), and is a synapomorphy of the node uniting these taxa in all four trees [2: 1→2]. It therefore does not provide support in our analysis for the topology hypothesized by Stocker.

*Alveolar ridges not visible in lateral view (9: 0→1).* We find this character to optimize as a synapomorphy in multiple locations across our four trees; however, these are mostly inconsistent with Stocker's hypothesis of this character's optimization.

In both trees which present the same topological hypothesis of the relationships of *Mystriosuchus* as *Stocker (2010)* (DC, DCM) (Figs. 5 and 7), this character is found as a synapomorphy of a clade containing *Machaeroprosopus pristinus*, *Machaeroprosopus buceros*, *Machaeroprosopus lottorum*, both species of 'Redondasaurus', and *Mystriosuchus* [3: 0→1]. This synapomorphy, however, describes a morphological reversal, that is, state 1→0, rather than 0→1 as suggested by Stocker. In the two trees in which *Mystriosuchus* occupies a more basal position (D, DM) (Figs. 4 and 6), this character is optimized as a 0→1 synapomorphy, as suggested by *Stocker (2010)*, of a clade similar to that described above, though differing by containing all members of *Machaeroprosopus* and excluding *Mystriosuchus*. In summary, we find this character to contradict the hypothesized optimization of *Stocker (2010)*, in that a 0→1 change is only found when *Mystriosuchus* is one of the sister taxa to *Machaeroprosopus*, rather than nesting within the clade.

*Postorbital squamosal articulation approximately transverse (22: 1→2).* The distribution of character state (2) is here restricted to members of *Machaeroprosopus*,

*Mystriosuchus* and '*Redondasaurus*', though it twice arises independently in the *Leptosuchus*-grade OTUs PEFO 34852 and *Coburgosuchus*. Despite its restricted occurrence, this trait change [33: $1 \rightarrow 2$] is not optimized as a synapomorphy here, though the change from $0 \rightarrow 1$ is optimized in two trees (DC, DCM) (Figs. 5 and 7) as a synapomorphy of the node linking *Smilosuchus adamanensis* with all more derived members of Leptosuchomorpha. In the DM tree a $0 \rightarrow 1$ change is a defining feature of the most recent node linking the clade of *Nicrosaurus* and *Coburgosuchus* with all more derived members of Leptosuchomorpha.

Although not optimized as a synapomorphy, the distribution of this character state is broadly supportive of not only the hypothesis of *Stocker (2010)*, but also that of *Hungerbühler (2002)*, as in both topologies, character state (2) is optimized as being plesiomorphic to the clade containing *Machaeroprosopus* and *Mystriosuchus*.

*Lateral ridge from postorbital/squamosal bar continues strongly on lateral surface of squamosal as two raised ridges (23: $1 \rightarrow 2$).* This character state was removed from the analysis as it could not be reliably identified in any species of phytosaur. A similar character state was added by *Butler et al. (2014)*, referring specifically to the bifurcation of the lateral ridge in species of *Parasuchus*, though this state has not been observed in any other phytosaurs. Here, we find the presence of a ridge to occur sporadically throughout the tree, though with a greater frequency in more derived members of *Machaeroprosopus*. In *Mystriosuchus* a ridge is only found as a polymorphism within *Mystriosuchus westphali*, and it is otherwise entirely absent within the genus. In topologies in which *Mystriosuchus* is a sister group of *Machaeroprosopus* the absence state is plesiomorphic to the group. When *Mystriosuchus* is found within *Machaeroprosopus*, the clade containing *Machaeroprosopus mccauleyi*, *Machaeroprosopus pristinus* and *Mystriosuchus westphali* is plesiomorphically polymorphic for this character. Furthermore, the presence of any form of ridge is only found as a synapomorphy of derived members of *Machaeroprosopus* in the D tree; in this topology *Mystriosuchus* is in any case excluded from the *Machaeroprosopus* clade.

*Posterior process of squamosal dorsoventrally expanded in lateral view (25: $2 \rightarrow 1$).* This character was altered to use the terminology of *Ballew (1989)* and *Hungerbühler (2002)* for the 'knob-like' posterior process found in *Machaeroprosopus pristinus*, *Machaeroprosopus buceros* and some specimens of *Machaeroprosopus mccauleyi*; this was done to reduce ambiguity in character scoring.

This character is not optimized as a synapomorphy of any node close to either the base of *Mystriosuchus* or *Machaeroprosopus* in any of the trees presented here. State (1) (which here refers to the same morphology as Stocker's character) is here found to be more frequent in derived members of *Machaeroprosopus*, (excluding *Machaeroprosopus pristinus* and *Machaeroprosopus buceros* which are characterized by a state change of $1 \rightarrow 2$) and is plesiomorphic for the clade. Although the general character distribution generally supports *Stocker's (2010)* topological hypothesis for all other members of Stocker's '*Pseudopalatus*' clade, this character does not convey any information regarding the position of *Mystriosuchus* as the taxon lacks a posterior process and optimization of this character at the base of *Mystriosuchus* relies entirely on its position in the

phylogeny. This character therefore provides no support for the inclusion of *Mystriosuchus* within *Machaeroprosopus*.

*Supratemporal fenestrae fully depressed, posterior process of parietal and entire parietal/squamosal bar below level of skull roof (32: 1→2)*. Rather than forming a synapomorphy of only the Mystriosuchini clade used by *Stocker (2010)*, we find this character to be synapomorphic for the node uniting *Smilosuchus gregorii* with all more derived taxa (D, DC (Figs. 4 and 5): node 23; DM (Fig. 6): node 22; DCM (Fig. 7): node 25) [49: 1→2]. *Mystriosuchus* is included within this clade regardless of its position with respect to *Machaeroprosopus*, thus this character does not provide any support for the inclusion of *Mystriosuchus* within *Machaeroprosopus*.

*Border of posttemporal fenestra formed laterally and slightly ventrally by process of squamosal that extends onto paroccipital process (37: 1→2)*. *Mystriosuchus* is scored here as polymorphic for this character. In the trees in which it occupies a more derived position *Mystriosuchus* forms a sister group to 'Redondasaurus', which consistently displays character state (0); the plesiomorphic state is, in this situation, also polymorphic—providing only limited support for the hypothesis of a derived placement for *Mystriosuchus*. This character is more consistent here with the hypothesis that *Mystriosuchus* is sister to *Machaeroprosopus*, as character state (2) alone is plesiomorphic for *Mystriosuchus* in this position, and forms a synapomorphy in three of our trees (DC, DM and DCM) (Figs. 5–7) for the clade formed by all descendants of the common ancestor of *Smilosuchus adamanensis* and *Mystriosuchus planirostris* [67: 0→2].

*Skull shape boxy in posterior view, width across squamosals approximately equal to width across ventral edge of quadrates (38: 1→0)*. This character was excluded in this analysis as it is extremely sensitive to taphonomic distortion, and is highly subjective. The most basal taxon in Mystriosuchini identified by *Stocker (2010)* is *Machaeroprosopus mccauleyi*, which contrary to Stocker's scoring would here be considered to possess a trapezoidal skull shape, as would *Machaeroprosopus buceros* and all taxa in 'Redondasaurus', none of which were included in Stocker's analysis. Despite the exclusion of this character, the inclusion of multiple additional taxa in this analysis may have affected the optimization of synapomorphies in the clade.

*Rostral crest present, continuous and sloping steeply anteroventrally from nares to terminal rosette (19: 0→1) (Possibly additional apomorphy under ACCTRAN)*. The above character was altered slightly in this analysis (Appendix 2); however, character state (1) of *Stocker (2010)* is still represented by character state (2) here. We find a wide range of synapomorphy optimizations of this character in our trees, none of which are consistent with the results of *Stocker (2010)*.

In the DCM tree (Fig. 7) a clade containing *Mystriosuchus*, 'Redondasaurus' and more derived members of *Machaeroprosopus* are partially defined by this character as a synapomorphy; however, *Machaeroprosopus mccauleyi* is excluded from the group and the state transformation is from the presence of a steep, continuous slope posteriorly from the terminal rosette, to the presence of a narial crest—the relatively abrupt rise from a thin, tubular snout to the nares [7: 2→1]. Within this clade, 'Redondasaurus' undergoes a state reversal back to the morphology of a steep, continuous crest [7: 1→2].

The D and DM trees (Figs. 4 and 6) both optimize this character as a synapomorphy of a clade including all species of *Machaeroprosopus* and '*Redondasaurus*'; in these trees, the state transformation is from the presence of a narial crest, to the presence of a partial rostral crest [7: 1→4]. A more exclusive clade within the former, containing *Machaeroprosopus mccauleyi*, *Machaeroprosopus jablonskiae* and '*Redondasaurus*' again features this character as a synapomorphy, with a state change from a partial rostral crest, to presence of a continuous steep slope [7: 4→2]; however, this feature is not preserved in *Machaeroprosopus jablonskiae*—its presence is inferred by the analysis based on the morphology present in *Machaeroprosopus mccauleyi* and '*Redondasaurus*' (the holotype of *Machaeroprosopus mccauleyi* is missing the anterior end of its rostrum, however, the presence of a continuous crest is verified by referred specimens, for example, PEFO 31219). *Mystriosuchus*, however, occurs in none of these clades in the two trees and this character is not found to support any relatively exclusive clade containing *Mystriosuchus*.

In the DC tree (Fig. 5) this character is not found to define any clade in which *Mystriosuchus* is placed; within close proximity to *Mystriosuchus* the only clade featuring this as a synapomorphy is '*Redondasaurus*', displaying a change from a narial crest to a continuous, steep rostral crest [7: 1→2].

*Supratemporal fenestrae mostly covered/completely closed dorsally, at most only anteromedial corners of supratemporal fenestrae visible in dorsal view (33: 1→2) (Possible additional apomorphy under ACCTRAN).* In the trees in which *Mystriosuchus* is recovered in a derived position this character was only found as a synapomorphy of the clade of *Mystriosuchus* + NHMW 1986 0024 0001 + MB.R. 2747; specifically, the synapomorphy denotes a character transformation from state (2) to state (1) [57: 2→1]. This does not provide support for the hypothesis of relationships within Mystriosuchini proposed by *Stocker (2010)*; however, the majority of nodal optimizations and scorings for this character in the other members of Mystriosuchini (and for all those included in Stocker's analysis), display character state (2). The state change observed by *Stocker (2010)* is likely not found here due to a polymorphic optimization of states (1) and (2) at the base of the *Machaeroprosopus* clade (*Machaeroprosopus andersoni* and all more derived taxa); at the node one step more derived (*Machaeroprosopus jablonskiae* and all more derived taxa) the character is optimized as state (2), as are the majority of following nodes. It is therefore likely that a state change of (1) to (2) [57: 1→2] is synapomorphic at the base of *Machaeroprosopus* which, in the DC and DCM topologies (Figs. 5 and 7), is consistent with the phylogenetic hypothesis of Stocker's 'Mystriosuchini' clade.

In the D and DM trees (Figs. 4 and 6) (i.e., where *Mystriosuchus* occupies a less derived position), this character is only optimized as a synapomorphy of '*Redondasaurus*' + NMMNHS P-31094, as a state change from the supratemporal fenestrae being mostly covered (state 2), to being fully covered (state 3). However, the synapomorphy suggested by *Stocker (2010)* is probably again suppressed due to two nodes optimized as polymorphisms bracketing the base of *Machaeroprosopus*. Using the D tree as an example: the two nodes directly basal to *Machaeroprosopus* (Fig. 4, nodes 28, 29) are

optimized as state (1) and (1 or 2) respectively; in the following node (the most basal in *Machaeroprosopus* (node 30)), this character is again optimized as (1 or 2). In the next node (node 31) the character is optimized as state (2), as are the majority of other nodes within the clade. Therefore, we suggest that the topology in the D and DM trees is also mostly consistent with the reduction in supratemporal fenestra visibility identified by *Stocker (2010)*, except that *Mystriosuchus* is excluded from the supported clade in the D and DM trees.

Relatively few of the synapomorphies identified in previous analyses to support particular clades containing *Mystriosuchus* are corroborated here, despite the dichotomy of tree topologies presented in this analysis being broadly consistent with each of the previous studies discussed above.

### Implications from the position of Protome

A common component of previous phylogenetic and biostratigraphic work relating to phytosaurs was the distinction between taxa (in particular *Leptosuchus* and *Machaeroprosopus*) on the basis of isolated squamosals (*Ballew, 1989*; *Long & Murry, 1995*; *Parker & Irmis, 2006*). This was possible because all non-Mystriosuchini members of Leptosuchomorpha possessed generally '*Leptosuchus*-like' squamosals and all members of Mystriosuchini possessed more '*Machaeroprosopus*-like' squamosals (with the exception of *Mystriosuchus* which possesses a distinctive squamosal morphology of its own), providing a clear distinction between two 'grades' of phytosaur with no overlap.

The implication of *Protome batalaria* being recovered in our study within Mystriosuchini (D and DM trees) (Figs. 4 and 6), and possessing a '*Leptosuchus*-like' squamosal, is to preclude the use of isolated squamosals to distinguish between Mystriosuchini and non-Mystriosuchini leptosuchomorphs. Conversely, in the DC and DCM trees *Protome* remains outside of Mystriosuchini (Figs. 5 and 7) and is thus consistent with the distinction of Mystriosuchini from non-Mystriosuchini taxa using only squamosal morphology.

Regardless of which tree topology is chosen, the usefulness of isolated squamosals for phylogenetic or biostratigraphic purposes is far from lost; *Protome* was never recovered within the *Machaeroprosopus* or *Mystriosuchus* clades in any of our trees, and as such has no bearing on the identification of members of *Machaeroprosopus* or *Mystriosuchus* based on isolated squamosals. Likewise, the presence of a *Machaeroprosopus*-like or *Mystriosuchus*-like squamosal is still consistent only with members of Mystriosuchini. The placement of *Protome* in all trees in this analysis (i.e., excluded from the main *Leptosuchus* clade, and distinct from *Smilosuchus*) does, however, suggest that isolated '*Leptosuchus*-like' squamosals should no longer be automatically assigned to *Leptosuchus* or *Smilosuchus*, and depending on which phylogeny is used, they should not be used as unequivocal evidence of non-Mystriosuchini taxa. In essence, this means that *Machaeroprosopus*- and *Mystriosuchus*-like squamosals remain indicative of Mystriosuchini, whereas *Leptosuchus*-like squamosals may not be indicative of non-Mystriosuchini Leptosuchomorpha, and further details should be investigated, such

as presence/absence of parietal prongs, which are exclusive to *Protome*, *Machaeroprosopus* and '*Redondasaurus*'.

### Relationship of Machaeroprosopus pristinus and Machaeroprosopus buceros

*Machaeroprosopus pristinus* and *Machaeroprosopus buceros* have previously been suggested to be conspecific, representing sexual dimorphs (*Zeigler, Lucas & Heckert, 2002*); a good summary of the current state of this debate is given in *Hungerbühler et al. (2013)*. This hypothesis was supported by the phylogenies of *Hungerbühler (2002)*, *Parker & Irmis (2006)* and *Hungerbühler et al. (2013)* in that they recovered the two taxa as monophyletic, although the hypothesis is inconsistent with the analysis of *Ballew (1989)*. Once again, we find a divergence in our results, with the DC and DCM trees (Figs. 5 and 7) supporting the hypothesis of conspecificity by resolving the taxa as sister taxa within the largely pectinate clade of *Machaeroprosopus*. The D and DM trees (Figs. 4 and 6) do not support this grouping—each finding a clade within *Machaeroprosopus* consisting of *Machaeroprosopus buceros* as the sister taxon to a clade of *Machaeroprosopus pristinus* and *Machaeroprosopus lottorum*.

The monophyly of the two taxa is supported at node 32 in the DC tree by two synapomorphies: [39: 1→2] the posterior process of the squamosal is modified into a 'terminal knob', and [90: 0→1] the presence of an additional ridge on the lateral surface of the posterior process of the squamosal, ventral to the ridge or rugosity from the po/sq bar. The relationship in the DCM tree was also supported by character 90, but character 39 was incorporated into the GM character defining the lateral shape of the squamosal; this landmark character does support the node, though all other landmark characters also support the node, and almost every other node in the tree. The node was also supported in the DCM tree by a continuously measured synapomorphy: [38: 0.442–0.457→0.077–0.319] representing an elongation of the posterior process of the squamosal.

No synapomorphies were given in support of the sister relationship between *Machaeroprosopus pristinus* and *Machaeroprosopus buceros* in any of the three studies that found this relationship; however, it was proposed by *Zeigler, Lucas & Heckert (2002)* that the two taxa (or sexual variants) differed only in the lengths and robustness of the premaxillae and septomaxillae. Our scoring generally agrees with this assertion. In fact, of the 94 characters included in our analysis, *Machaeroprosopus pristinus* and *Machaeroprosopus buceros* were found to differ in only five characters—four of which are directly related to snout shape, robustness or length (7: rostral crest; 8: relative transverse width of rostrum; 11: ratio of rostral to narial + postnarial length; 79: dorsal surface of snout—cross-sectional shape). The remaining character (89) scores the diagonal aspect ratio of the infratemporal fenestra, with *Machaeroprosopus pristinus* possessing an infratemporal fenestra with a higher aspect ratio, that is, the fenestra is more compressed/acute.

In the D and DM trees, the relationship between *Machaeroprosopus pristinus* and *Machaeroprosopus lottorum* to the exclusion of *Machaeroprosopus buceros* is supported consistently by three characters: [7: 4→1] presence of a narial crest; [8: 2→1] relative

transverse width of the rostrum—moderate; and [89: 0 → 1] infratemporal fenestra has a high aspect ratio (although this is represented as a GM character in the DM tree). As is the case for almost all nodes of the trees containing GM data, all landmark-based characters also support this node. It would appear to be a justifiable hypothesis to suggest that in these trees *Machaeroprosopus buceros* and *Machaeroprosopus pristinus* are only separated due to rostral morphology, as suggested by *Zeigler, Lucas & Heckert (2002)*. It would seem likely that if rostral robusticity was assumed to be a sexually dimorphic character, and excluded from analysis, that *Machaeroprosopus buceros* and *Machaeroprosopus pristinus* would be recovered together in all tree topologies, thus providing support to the hypothesis of conspecificity.

### Monophyly of 'Redondasaurus'

'*Redondasaurus*' was originally diagnosed by *Hunt & Lucas (1993)* solely on the basis of the lack of visibility of the supratemporal fenestrae in dorsal view. The genus was re-diagnosed by *Spielmann & Lucas (2012)* with a broader complement of characters: (1) supratemporal fenestrae concealed in dorsal view; (2) reduced antorbital fenestrae; (3) a prominent pre-infratemporal shelf at the anteroventral margin of the lateral temporal fenestra; (4) septomaxillae wrap around the outer margin of the external narial opening; (5) thickened orbital margin; (6) inflated posterior nasal behind the external narial opening; (7) thickened dorsal osteoderms.

*Hungerbühler et al. (2013)* were unable to recover '*Redondasaurus*' *gregorii* and '*Redondasaurus*' *bermani* as a monophyletic group in any of their trees; however, we find a monophyletic '*Redondasaurus*' (albeit nested within *Machaeroprosopus*) in all of our trees. A possible contributory factor in this difference is that species in the analysis of *Hungerbühler et al. (2013)* were scored only with reference to holotype specimens— resulting in an increased proportion of missing data in some taxa. Total proportions of missing character data in *Hungerbühler et al. (2013)* are unavailable, and therefore cannot be compared with those of taxa surrounding '*Redondasaurus*' in the present study; a further difference between these studies, however, that may be discussed is the inclusion of different characters.

Many of the characters proposed by *Spielmann & Lucas (2012)* were not implemented in the analysis of *Hungerbühler et al. (2013)*; however, in this analysis we included some of these characters that were used in previous phylogenetic studies and independently identified others which overlap to a considerable extent with those proposed synapomorphies of '*Redondasaurus*'. The consistency of the characters included in our analysis with the hypothesis of a monophyletic '*Redondasaurus*' are discussed below.

*Supratemporal fenestrae concealed in dorsal view.* As was briefly mentioned above, this character is found as a synapomorphy of the '*Redondasaurus*' clade in all trees presented in this study [57: 2 → 3], and is therefore entirely consistent with the hypothesis of *Hunt & Lucas (1993)*. This character state occurs in no other taxon, though is found in NMMNHS P-31094 (referred to '*Redondasaurus*' *gregorii* by *Spielmann & Lucas, 2012*), which in the D and DM trees is included within the '*Redondasaurus*' clade,

but in the other trees is recovered as the sister taxon of *Machaeroprosopus lottorum*, the character state having arisen independently of 'Redondasaurus'.

*Anteriormost border of pre-infratemporal shelf terminates anterior of the posteriormost corner of the antorbital fenestra.* The presence of this character state is restricted almost entirely to 'Redondasaurus' and *Mystriosuchus*; unsurprisingly, where these two groups form a clade this character is consistently optimized as a synapomorphy. However, in the D and DM trees, where *Mystriosuchus* is placed basally, distant from 'Redondasaurus', the character only constitutes a synapomorphy for *Mystriosuchus* rather than 'Redondasaurus'; this may be due to the polymorphic condition of 'Redondasaurus' *gregorii* for this character. Despite this inconsistency between trees the distribution of this character still broadly supports a monophyletic 'Redondasaurus'.

The diagnostic characters proposed by *Spielmann & Lucas (2012)* for 'Redondasaurus' but not included in our analysis are discussed briefly below. We agree that several of these support a sister taxon relationship between 'Redondasaurus' *gregorii* and 'Redondasaurus' *bermani*, and are therefore consistent with our results.

*Reduced antorbital fenestrae.* Whether or not the antorbital fenestrae are substantially reduced may be subjective; in more robust specimens of 'Redondasaurus' (NMMNHS P-4256) the antorbital fenestra does appear smaller than in closely related taxa. However, in more gracile specimens (YPM 3294) the fenestra appears similar in proportions to those of other phytosaurs such as *Mystriosuchus*. The antorbital fenestrae do appear to exhibit a unique shape in most specimens of 'Redondasaurus'; the general shape is roughly triangular, as is common in *Mystriosuchus* and *Machaeroprosopus*, but the anterior- and posterior-most corners of the fenestra are sharp angles, rather than smooth curves.

*Septomaxillae wrap around the outer margin of the external narial opening.* No taxon studied was observed to possess 'septomaxillae' that extend onto the lateral surface of the external nares. *Stocker (2010)* noted the presence of this character state in 'Redondasaurus' and suggested it may also occur in *Pravusuchus hortus*; however, upon inspection of the holotype and referred specimens of *Pravusuchus hortus* it seems equally likely that the morphology described by Stocker pertains to cracks on the holotype, with the true sutures covered by iron oxide. Rather than a lateral extension of the 'septomaxillae' the feature identified in 'Redondasaurus' and *Pravusuchus* may represent the paranasals, identified in *Machaeroprosopus lottorum* by *Hungerbühler et al. (2013)*.

*Thickened orbital margin.* We here concur with *Spielmann & Lucas (2012)*; in all specimens of 'Redondasaurus' examined by us, the descending process of the postorbital appears to be greatly thickened to an extent not seen in any other group. For this particular character *Spielmann & Lucas (2012)* suggested it is also shared with *Coburgosuchus*; however, we see no observable expansion of the postorbital in the latter taxon to distinguish it from the condition present in most other phytosaurs.

The descending process of the postorbital in *Coburgosuchus* has a roughly rectangular cross-section, with the external face relatively thin, but facing anterolaterally. If *Spielmann & Lucas (2012)* measured this feature in *Coburgosuchus* diagonally between the anterolateral and posteromedial corners (i.e., the full width observable in direct lateral

view), this could account for the increased width, especially given the oblique angle of the process in direct lateral view.

An alternative possibility is that the character is intended to describe a general thickening of the circumorbital bones, resulting in a more blunt appearance and the elevation of the orbital rim; the orbital rim in *Coburgosuchus* is dorsally elevated, but shows no other evidence of thickening (Axel Hungerbühler, personal communication to Andrew S. Jones, 2018). This interpretation would put this character partially in conflict with character 31, 'Medial margins of orbits', and given that this morphology is measured in all other taxa based on only the flat lateral-most face of the descending process, this procedure was also applied here to preserve homology within the character.

*Inflated posterior nasal behind the external narial opening.* Although this entire area of skull is missing in the type specimen of '*Redondasaurus*' *gregorii* (YPM 3294), it is common to a variable extent in many other specimens referred to the genus by *Spielmann & Lucas (2012)*. This feature is not, however, restricted to '*Redondasaurus*', as the morphology of specimens from other taxa frequently overlap with the range of variation observed in '*Redondasaurus*'. Examples include: *Nicrosaurus kapffi* (SMNS 4379), *Machaeroprosopus mccauleyi* (PEFO 31219) and *Machaeroprosopus lottorum* (TTU-P 10076). It may be valid to say that '*Redondasaurus*' is the only taxon in which this character state consistently occurs; however, its variability makes the taxonomic utility of this feature unclear. Given the variable presence of this character in more than one species of *Machaeroprosopus*, this character is likely to support the hypothesis that '*Redondasaurus*' is nested within *Machaeroprosopus*, though verification would require the inclusion of this character in phylogenetic analyses.

*Thickened dorsal osteoderms.* The osteoderms of some large phytosaur taxa are also strongly thickened, for example, *Smilosuchus gregorii* (AMNH FR 3060); however, we have not carried out any sufficiently detailed study of osteoderms to fully assess this proposed synapomorphy. Until more detailed work emerges on phytosaur osteoderm variation we tentatively accept this character, though emphasize that potential size correlation should be borne in mind.

### Synonymy with *Machaeroprosopus*

*Hungerbühler et al. (2013)* presented three lines of reasoning in support of the synonymization of '*Redondasaurus*' into *Machaeroprosopus*. First, they argued that *Machaeroprosopus lottorum* 'bridges the morphological gap' between other members of *Machaeroprosopus* and '*Redondasaurus*' *gregorii* and '*Redondasaurus*' *bermani* in a number of features, and possesses a combination of characters formerly considered exclusive to one or other group.

Second, in all trees recovered by *Hungerbühler et al. (2013)*, both species of '*Redondasaurus*' were found within the clade of *Machaeroprosopus*; in analyses that were constrained to recover '*Redondasaurus*' as a monophyletic sister group to *Machaeroprosopus*, tree fit lengthened by five extra steps.

Third, they did not find '*Redondasaurus*' *gregorii* and '*Redondasaurus*' *bermani* to form a clade to the exclusion of species of *Machaeroprosopus* in any of their trees; instead,

the two taxa were interspersed with members of *Machaeroprosopus*, with 'Redondasaurus' *gregorii* being recovered in a substantially more derived position than 'Redondasaurus' *bermani* in every tree.

The first two points are consistent with our results; however, with regard to their third point we find the exact opposite—that these taxa are always monophyletic to the exclusion of species of *Machaeroprosopus*. In all trees this clade is supported by two to four synapomorphies, with one (supratemporal fenestra completely obscured in dorsal view) consistently present in all trees. One exception is NMMNHS P-31094, which was previously referred to 'Redondasaurus' (*Heckert et al., 2001*), yet in half of our trees is found to form a monophyly with *Machaeroprosopus lottorum*. This observation corresponds well with the findings of *Hungerbühler et al. (2013)*. On this basis, and due to 'Redondasaurus' consistently being resolved within *Machaeroprosopus*, we agree with the suggestions of *Long & Murry (1995)* and *Hungerbühler et al. (2013)* that 'Redondasaurus' should be synonymized with *Machaeroprosopus*. It is clear, however, from our phylogenies, differences in cranial morphology and the general difference in stratigraphic age, that the species attributed to 'Redondasaurus' represent some of the most derived taxa within *Machaeroprosopus*.

### Specimen-level OTUs

A number of specimens were included as individual OTUs in our analyses in order to test their affinities. Here, we report on those of particular importance and those which occupy an interesting phylogenetic position.

**NMMNHS P-4781.** This specimen was originally assigned to *Angistorhinus* by *Hunt, Lucas & Bircheff (1993)* (see Appendix 1); our analyses corroborate that view, recovering the specimen as the most basal member of the *Angistorhinus* clade in all analyses, with the node supported by some of the highest Bremer and frequency support scores in the entire tree. The node is additionally supported by a number of synapomorphies: two synapomorphies were consistent between the D, DC and DM trees [56: $0 \rightarrow 1$] parietal-squamosal bars curved medially before attaching to squamosal; [58: $0 \rightarrow 1$] parietal-squamosal bars wide—approximately the same width as the po/sq bar. One further synapomorphy was only present in the DC and DCM trees, being scored continuously [87: 0.106–0.110 (0.106–0.132 in DCM tree) $\rightarrow$ 0.103] relatively robust jugal—becoming slightly more robust. Despite being less derived than all other members of the *Angistorhinus* clade, this specimen is potentially younger than the others—being from the early Norian (225–218 Mya), rather than the Carnian to early Norian (232–225 Mya).

**PEFO 34852.** This specimen has previously been identified as 'Smilosuchus' *adamanensis* (*Griffin et al., 2017*); however, we disagree with their diagnosis (see Appendix 1), suggesting the specimen shares more similarities with *Leptosuchus crosbiensis*. Our analysis in part supports our hypothesis, as PEFO 34852 forms a monophyletic group with *L. crosbiensis* in half of our trees (DC and DCM). In the other half (D and DM), PEFO 34852 forms a relationship with 'Smilosuchus lithodendrorum'; however, this group's closest sister taxon is *L. crosbiensis* in both trees, and in

addition 'S. lithodendrorum' has previously been suggested to be synonymous with L. crosbiensis (Long & Murry, 1995).

The specimen's relationship with L. crosbiensis in the DC and DCM trees was supported by two consistent synapomorphies: [48: 0→1] squamosal fossa does not reach posterior edge of squamosal; [89: 0.457→0.462] increase in the aspect ratio of the infra-temporal fenestra, however, the latter synapomorphy is represented as a GM character in the DCM tree, and all other landmark-based characters also support the node. Griffin et al. (2017) scored character 48 in the opposite way to which we did here and could represent some subjectivity with regards to the delimitation of this character's states. This was one of the characters that was used to identify the specimen as S. adamanensis; however, only very few verifiable specimens of S. adamanensis exist and all other 'Smilosuchus' taxa were observed to be polymorphic with regards to this character, so this may not be an ideal species identifier.

In the D and DM trees, PEFO 34852 forms a relationship with 'S. lithodendrorum' to the exclusion of L. crosbiensis—this relationship is supported by two consistent synapomorphies: [3: 0→1] alveolar ridges inconsistently visible or entirely hidden in lateral view; [33: 0→2] sutural articulation of postorbital and squamosal in dorsal view—approximately transverse. In the D tree only a further synapomorphy is given: [7: 4→2] a straight, steep slope from the nares to the premaxilla; although the crest does undulate to an extent, we feel that this scoring was most appropriate, given the continuous anterior slope of the rostrum, with no horizontally level portions. In the DM condition all landmark-based characters are also found to support the node.

PEFO 34852 is the first phytosaur specimen that has been recorded as possessing three sacral vertebrae (Griffin et al., 2017); however, in their discussion the authors make it clear that this apparent novel morphology is likely widespread throughout Phytosauria, being present in members of non-leptosuchomorph Mystriosuchinae (Angistorhinus), other non-Mystriosuchini leptosuchomorphs (holotype of S. adamanensis) and members of Mystriosuchini ('Machaeroprosopus' zunii, 'Redondasaurus'), and was previously misinterpreted in past studies. The morphologies only appeared to differ in the extent to which various sutures had fused, which may be due to ontogenetic factors. Given this feature is seemingly homogeneous within phytosaurs, the different taxonomic position for this specimen that we propose here should not affect the conclusions drawn by Griffin et al. (2017).

**MB.R.2747.** This specimen represents the only substantial Rhaetian phytosaur material from Europe, and by a considerable margin is also the largest phytosaur currently known from that continent. We consistently recovered this specimen as the basalmost taxon in the Mystriosuchus clade in all our trees; given the two conflicting positions in which the Mystriosuchus clade has been found in our trees, we also find two independent suites of synapomorphies supporting MB.R.2747 at the base of this clade.

Only one character was found consistently in all trees: [85: 1→0] the pre-infratemporal shelf merges dorsally into the lateral face of the jugal, rather than continuing as a ridge to contribute to the descending process of the postorbital. In the D and DM trees in which Mystriosuchus occupies a more basal position, there are two additional consistent

synaptomorphies: [59: 1, 2→3] dorsal edge of the parietal/squamosal bar entirely, or in parts vertical; [84: 0→1] the pre-infratemporal shelf extends anteriorly past the posteriormost corner of the antorbital fenestra. In addition, in the DM condition all GM characters support this node.

In the DC and DCM trees the *Mystriosuchus* clade is also supported by two additional synaptomorphies: [57: 2→1] supratemporal fenestrae mostly visible in dorsal view; [61: 2→1] lateral wall of the supraoccipital shelf is low and continuously thin. The node is additionally supported by all landmark-based characters in the DCM tree.

Similarly, to *Angistorhinus* and NMMNHS P-4781, MB.R. 2747 is basal to the *Mystriosuchus* clade, yet is younger in age (Appendix 1), suggesting a ghost range for this taxon extending to the middle—late Norian. This specimen was originally referred to 'Angistorhinopsis ruetimeyeri'; however, the type specimen of that genus and species contains no diagnostic material and may be chimaeric, and as such has been widely accepted as a nomen dubium. The placement of this specimen at the base of the *Mystriosuchus* somewhat corroborates the speculative cranial reconstruction of the specimen by von Huene (1922), in which the depressed temporal arcade and the posterior processes of the squamosals are modelled after *Mystriosuchus*. Further investigation into this specimen, including a thorough redescription, is currently underway.

**USNM V 17098.** This specimen was referred to *Leptosuchus* sp. by Long & Murry (1995), yet in our analyses it is constantly recovered in a more derived position than other members of non-Mystriosuchini Leptosuchomorpha; in three of our trees it is recovered within Mystriosuchini. The specimen is labelled as 'Machaeroprosopus zunii' without any written justification, though in support of this we recover this specimen in a similar position to the holotype of 'Machaeroprosopus zunii' in three of our trees, in one of which the two OTUs form a clade.

The single clade that supports the identification of 'Machaeroprosopus zunii' (DC tree, Fig. 5, node 30) has a relatively good frequency support score, and possesses two synaptomorphies: [39: 1→0] posterior process of the squamosal greatly dorsoventrally expanded; [53: 1→2] dorsally expressed ridge present around anterior and medial edge of the supratemporal fenestra. Whether or not USNM V 17098 is referable to 'Machaeroprosopus zunii' does, however, remain problematic due to the erection and description of the species based on many cranial and mandibular fragments, grouped on the basis of their geographic area of discovery, rather than morphological similarity (Camp, 1930). To produce a comprehensive reanalysis of this taxon would require intensive study of all material referred by Camp (1930); however, this is well beyond the scope of this study.

The consistently derived position of 'Machaeroprosopus zunii' does suggest that it may not be simply a species of *Leptosuchus* as suggested by Long & Murry (1995). The phylogenetic position of USNM V 17098, and of the holotype of 'Machaeroprosopus zunii' (where they occupy a similar placement (Figs. 4–6)) is supported by five consistent synaptomorphies in the D and DM trees (Figs. 4 and 6): [3: 0→1] alveolar ridges inconsistently visible or entirely hidden in lateral view; [4: 0→1] presence of a ventral alveolar bulge between the premaxilla and maxilla; [7: 1→4] rostral crest extends

horizontally from nares for less than half the length of the nares, then becomes tubular; [46: 1→0] (incorporated into landmark character in DM tree) ventral margin of squamosal slopes continuously anteroventrally from the posterior edge of the posterior process to the opisthotic process, without any horizontal edge; [47: 0→1] presence of a subsidiary opisthotic process of the squamosal.

In the DC tree the position of the clade of USNM V 17098 and the holotype is supported by two different synapomorphies: [22: 3→2] reduced antorbital fossa in which the lacrimal, maxillary and jugal fossae are not touching; [75: 0→1] presence of a prominent, sharp palatal ridge.

The character states scored in USNM V 17098, regarding characters 3, 4 and 46 are almost exclusively restricted to members of 'Smilosuchus' and robust taxa in Machaeroprosopus. The states in characters 47 and 75 are almost exclusively limited to most members of Machaeroprosopus, though these states also occur in 'Smilosuchus' gregorii and Mystriosuchus (in character 47), and in Rutiodon, Nicrosaurus and Mystriosuchus (in character 75). The character states recorded for characters 7 and 22 occur frequently in many taxa throughout the tree, including many members of Leptosuchus and 'Smilosuchus'. Further study is required to ascertain the taxonomic validity of 'Machaeroprosopus zunii', but we do find a reasonable quantity of evidence to support it as being distinct from Leptosuchus.

**NMMNHS P-4256 & P-31094.** These specimens have both been previously referred to 'Redondasaurus' (Hunt, 1994; Heckert et al., 2001); however, their positions in our trees produce some uncertainty regarding this. NMMNHS P-31094 was found to form a clade with both species of 'Redondasaurus' in two of our trees (D and DM) (Figs. 4 and 6), whilst in the same trees NMMNHS P-4256 falls basal to Machaeroprosopus jablonskiae, and slightly more derived than Machaeroprosopus mccauleyi. The supratemporal arcade of Machaeroprosopus jablonskiae is robust, and also possesses some of the features of 'Redondasaurus', such as broad and proportionately short postorbital/squamosal bars and the lack of a knob-like posterior process of the squamosal; its phylogenetic proximity to 'Redondasaurus' is therefore unsurprising, furthermore NMMNHS P-4256 has also been referred to Machaeroprosopus mccauleyi (Hunt, Lucas & Spielmann, 2006), so the proximity of this specimen to Machaeroprosopus mccauleyi is also understandable.

Conversely, in the DC and DCM trees (Figs. 5 and 7) NMMNHS P-4256, alongside Machaeroprosopus mccauleyi and Machaeroprosopus jablonskiae are recovered at the base of the Machaeroprosopus clade, suggesting that rather than being closely associated with 'Redondasaurus', NMMNHS P-4256 appears to be more closely linked to Machaeroprosopus mccauleyi and Machaeroprosopus jablonskiae. In these tree topologies, NMMNHS P-31094 remains one of the more derived members of the Machaeroprosopus clade; however, it forms a clade with Machaeroprosopus lottorum exclusive to the slightly more derived clade formed by the two species of 'Redondasaurus'.

The node uniting NMMNHS P-31094 and Machaeroprosopus lottorum is supported by five synapomorphies in the DC tree (Fig. 5), and three synapomorphies (plus all five landmark-based characters) in the DCM tree (Fig. 7), with all synapomorphies at this

node in the DC tree overlapping with those in the DCM tree. The synapomorphies are as follows: [25: 0.439–0.514 (DCM: 0.325–0.439)→0.113] antorbital fenestra is relatively short anteroposteriorly; [34: 1→2] anteroventral corner of infra-temporal fenestra in front of anterior rim of the orbit; [53: 1→0] absence of a dorsally expressed ridge around the anterior or medial edges of the supratemporal fenestra; [54: 0.180–0.197→0.039–0.095 (DCM: landmark-based)] relatively wide postorbital/ squamosal bar; [89: 0.550–0.620→0.710 (DCM: landmark-based)] relatively high aspect-ratio of the infra-temporal fenestra.

In the D and DM trees (Figs. 4 and 6) the 'Redondasaurus' clade (if defined to include both conventional species plus NMMNHS P-31094), is consistently supported by only four synapomorphies (and all landmark-based characters in the DM tree): [19: 1→0] interorbital-nasal area is flat in lateral view; [57: 2→3] supratemporal fenestra completely obscured in dorsal view; [59: 2→3] dorsal edge of the parietal/squamosal bar is either entirely or in parts vertical; [63: 1→0] supraoccipital shelf is shallow and its longitudinal axis is predominantly vertical.

Martz & Parker (2017) in part defined the base of the Apachean biozone (207–202 Ma) as the stratigraphically lowest occurrence of 'Redondasaurus'; NMMNHS P-31094 is dated to the Rhaetian (208.5–201.3 Mya), which largely overlaps the Apachean biozone; however, NMMNHS P-4256 does not, being instead dated to the late Norian (c. 218–208 Mya). If the latter specimen was found to be consistently recovered within 'Redondasaurus' it would extend the age range of this taxon, thus invalidating the definition of the Apachean given by Martz & Parker (2017). Given that NMMNHS P-4256 was never recovered within 'Redondasaurus' in our analysis, and was separated from 'Redondasaurus' by Machaeroprosopus jablonskiae, we find no reason to doubt the biozones of Martz & Parker (2017).

## Effects of scoring method

### Consistency index and retention index

The CI calculated for the four character coding variables (D, DC, DM and DCM) were broadly similar; though as noted above, those which incorporated continuous data produced slightly better scores than the others. Regardless, all CI values displayed a significantly higher consistency than expected of random data (for a dataset of 43 taxa and between 90 and 94 characters), based on comparisons with simulated data in Klassen, Mooi & Locke (1991). Differences in the RI were marginal between all conditions, indicating that despite the increased homoplasy in GM datasets, the same proportion of synapomorphic information was retained as in datasets excluding GM data. As the RIs of the continuous and non-continuous datasets are almost identical, it is unlikely that the difference in homoplasy indicated by CI between the datasets can be ascribed to a greater proportion of uninformative or autapomorphic characters in the continuous dataset.

### Tree length

When comparing the tree-length (weighted homoplasy) produced by datasets with equal numbers of characters, trees that incorporate continuous data are consistently

shorter than those which exclude it. The D tree (94 characters) produced a tree-length of 31.90, whereas the DC tree (94 characters) produced a length of 27.46. Likewise, the DM tree (90 characters) recovered a length of 30.52, while the DCM (90 characters) tree-length was 25.44.

The effects of including GM data cannot be interpreted in the same way as above; the base D dataset contains more characters than the DM dataset, and we would therefore naturally assume that the DM tree would be shorter just by virtue of having fewer characters. It is, however, possible to say that the continuous characters in this study do have a shortening effect on tree-length when compared to the standard discrete data tree (D vs DC tree-length). Furthermore, the incorporation of continuous data into the DM dataset (DM vs DCM tree-length) resulted in a greater reduction in tree length than was produced by the combined effect of incorporating GM data into the D dataset and the associated reduction in the number of characters (D vs DM). This may indicate that the continuous characters in this dataset produced a stronger influence on tree length than the GM characters. Additionally, as extended implied weighting was in effect the shorter tree lengths equate to reduced homoplasy. Considering the higher CI of the continuous datasets, it is unsurprising that the continuous datasets also produce the shortest tree lengths when compared to D and DM, as under implied weighting, the 'length' of each character is partially calculated using the same technique as the CI. The overall tree-length is an ensemble score of estimated homoplasy within the dataset—similarly the CI measures ensemble consistency.

### Topological similarity

In analyses of topological similarity (maximum agreement subtrees, SPR distances and RF distance) the DC tree differed from the base discrete data tree by 37.2%, 32.5% and by 0.45122 in each respective metric, whereas the DM tree only differed from the base tree by 23.3%, 15% and 0.23171 respectively. This suggests that the incorporation of continuous characters into the base dataset altered the topology of the output tree to a greater extent than by incorporating GM characters.

Within our overall dataset, continuous characters appear to exert a stronger influence on tree topology and tree length than GM characters, and the incorporation of continuous rather than GM characters produces a tree that is found to be slightly less homoplastic by CI and implied weighting.

It should be noted that the elevated influence of continuous data may be related to variations in our dataset rather than an inherent property of the scoring method. For example, in the DC condition continuous data accounted for 10.64% of the characters used, but in the DM condition GM data only accounted for 5.56% of the total characters; therefore, continuous data may have more influence as it constituted a greater proportion of the data. Alternatively, it is possible that the characters scored as continuous data may, by chance, have been less homoplastic than those scored using GM techniques. It should also be noted that these two influences are not mutually exclusive.

*Support metrics*

A slightly different finding to the above was obtained when investigating Bremer and frequency supports. When collapsing nodes with Bremer scores less than that of the average character step length (0.11), the datasets incorporating GM data (DM and DCM) produced consistently poorer total Bremer support for the collapsed tree, and retained less nodes than the non-GM datasets (D and DC). The mean Bremer support values for nodes exceeding the cut-off were almost entirely consistent between all four data treatments, whereas at the lower cut-off (0.08) these means were more variable. This suggests that the cut-off of 0.11 largely retained the nodes for which the Bremer support values were more resistant to the effects of data treatment.

In contrast to Bremer scores, frequency supports performed more consistently between scoring techniques in terms of number of nodes retained; however, similarly to the results of Bremer supports, the DCM treatment produced the worst results. The pattern of summed frequency values matched the general trend of the Bremer supports, that is, the GM conditions produced lower total support for the collapsed tree; although, the mean frequency supports across the four collapsed trees were again relatively constant.

When the Bremer and frequency support values were averaged in five tree-regions and summed within each tree, in both metrics the DC condition produced the best values and the two GM conditions produced the worst.

## CONCLUSIONS

Our analyses, split between D+DM and DC+DCM trees, broadly support the partially conflicting phylogenetic relationships recovered by previous studies (*Hungerbühler & Sues, 2001*; *Hungerbühler, 2002*; *Parker & Irmis, 2006*; *Stocker, 2010, 2012, 2013*; *Hungerbühler et al., 2013*; *Butler et al., 2014*; *Kammerer et al., 2015*). In particular, a close relationship between *Rutiodon* and *Angistorhinus*, suggested by *Hungerbühler & Sues (2001)* was recovered in all of our trees, as was the paraphyly of 'Paleorhinus' (*Wroblewski, 2003*; *Parker & Irmis, 2006*; *Butler et al., 2014*), the monophyly of *Parasuchus* (*Kammerer et al., 2015*), the placement of *Wannia* as the basalmost member of Parasuchidae (*Stocker, 2013*; *Kammerer et al., 2015*), the recovery of *Diandongosuchus* as the most basal member of Phytosauria (*Stocker et al., 2017*), and the synonymy of 'Redondasaurus' and *Machaeroprosopus* (*Hungerbühler et al., 2013*). The conflicting positions of *Mystriosuchus* were also recovered in our trees; the D and DM trees found *Mystriosuchus* as a clade basal to *Machaeroprosopus* (*Hungerbühler, 2002*; *Parker & Irmis, 2006*; *Hungerbühler et al., 2013*), whereas the DC and DCM trees placed *Mystriosuchus* as one of the most derived clades within Mystriosuchini, and within the *Machaeroprosopus* clade (*Ballew, 1989*; *Stocker, 2010, 2012, 2013*; *Butler et al., 2014*; *Kammerer et al., 2015*).

Some relationships observed in our trees are inconsistent with previous analyses, such as our recovery of *Nicrosaurus* nested deeply within leptosuchomorph taxa, as opposed to being a basal member of Mystriosuchini; this led us to redefine Mystriosuchini, by excluding *Nicrosaurus* as an internal specifier.

*Machaeroprosopus pristinus* and *Machaeroprosopus buceros* were found to form a monophyly in half of our trees (DC and DM), supporting the hypothesis of conspecificity (*Zeigler, Lucas & Heckert, 2002*). Furthermore, in the trees in which *Machaeroprosopus pristinus* instead formed a relationship with *Machaeroprosopus lottorum* to the exclusion of *Machaeroprosopus buceros* (D and DM), the relationship was predominantly supported by characters pertaining to rostral morphology, which have previously been proposed as signals of gender rather than species (*Abel, 1922*; *Colbert, 1947*; *Zeigler, Lucas & Heckert, 2002*; *Hunt, Lucas & Spielmann, 2006*; *Kimmig, 2009*), and if so would be irrelevant to phylogenetic analysis.

*Protome* was recovered in a far more derived position than before, being consistently found within Leptosuchomorpha, and in two trees within Mystriosuchini. This has wide ranging implications for the use of isolated phytosaur squamosals in biostratigraphy; *Leptosuchus*-type squamosals have previously been restricted to non-Mystriosuchini taxa, however, the potential inclusion of *Protome* (which possesses *Leptosuchus*-type squamosals) in Mystriosuchini indicates this distinction between phytosaur 'grades' may not be justified. *Machaeroprosopus*-type squamosals, however, remain a potential identifier for membership of Mystriosuchini.

We found a great deal of inconsistency in the relationships of leptosuchomorph phytosaurs, especially non-Mystriosuchini leptosuchomorphs. In contrast to the findings of *Stocker (2010)*, we found inconsistent support for a monophyletic *Leptosuchus* (present only in condition D), and no support for the monophyly of *Smilosuchus*. The inconsistency among these phytosaurs is reminiscent of their difficult taxonomic and phylogenetic history, in which these taxa were shuffled between genera including *Rutiodon*, *Leptosuchus*, *Smilosuchus* and *Machaeroprosopus*, in various combinations (*Camp, 1930*; *Colbert, 1947*; *Gregory, 1962b*; *Ballew, 1989*; *Long & Murry, 1995*; *Stocker, 2010*). Although the overall consensus tree (Fig. 8) suggests most derived phytosaurs could be classified as numerous monospecific genera, it seems more likely that the true phylogenetic relationships are masked by wildcard taxa/specimens and uncertainties in intra/interspecific variation. Further work, investigating the relationships of leptosuchomorph phytosaurs, building on the studies of *Stocker (2010)* and *Hungerbühler et al. (2013)* and including predominantly well-known, morphologically complete taxa would be very useful.

To broadly summarize our findings regarding character use—for our dataset it appears that continuous characters consistently exert a greater influence over the results than GM characters, and in comparison to datasets excluding continuous characters, they also appear to reduce homoplasy. GM characters in this study produced trees with generally worse nodal support values, and despite the lack of polytomies within the best-fit trees, when collapsing nodes to adjust for over-resolution of the tree the GM datasets retained fewer nodes at a reasonable cut-off value than the continuous and discrete trees.

A potential drawback of using GM data in particular is the relative difficulty, in comparison to discrete characters, of interpreting morphological changes in a way that is useful for producing written diagnoses. For synapomorphic continuous characters it is

possible to express the character 'state' of a taxon or group as a numerical range and transformations as shifts from one range to another; however, describing subtle, but apparently phylogenetically relevant changes in shape according to geometric morphometrics necessitates either multiple diagrams of landmark displacements at supported nodes, or long breakdowns of morphology, and an elevation of analytical complexity for relatively little gain (at least in the case of this dataset). An example of the perplexity caused by GM data may be seen in the nodal synapomorphies in the treatments which incorporate GM data (Appendix 3); in both trees (DM and DCM) almost all GM characters are optimized as synapomorphies for almost every node.

A further obstacle to incorporating substantial amounts of GM data into phylogenetic analyses is that in palaeontological datasets, and especially with phytosaurs, it is relatively uncommon to find the pristine, non-deformed morphologies necessary for GM comparisons. Furthermore, GM characters may inherently encompass multiple discrete characters; if one aspect of a morphological feature is deformed (thus rendering the feature unusable for GM), all associated morphological features to be scored by the same configuration of landmarks would also have to be excluded from the analysis. In this sense, the addition of GM characters into a dataset may actually increase the quantity of missing data in a dataset where the characters could be alternatively scored with discrete or continuous methods.

For the various reasons outlined above we prefer the D and DC trees as they either incorporate continuous data, exclude GM data, or both. These trees are also representative of the two conflicting topologies found in this study and are generally consistent with previous analyses of ingroup Phytosauria. On the basis of statistical comparisons and similar nodal support values we suggest that either tree would be equally valid for use in further study; however, we find the D tree to be potentially less computationally and systematically problematic.

First, the D matrix uses only discrete characters and is therefore more easily implemented into a broad range of phylogenetic software packages, allowing new data to be easily added and analysed in the future, rather than the DC matrix which (in its current form) is restricted to the software TNT. Additionally, whilst neither tree recovers high support values for non-Mystriosuchini Leptosuchomorpha, the D tree does retain a monophyletic *Leptosuchus* clade (with the addition of '*Smilosuchus lithodendrorum*'). The D tree also recovers *Mystriosuchus* as distinct from *Machaeroprosopus*, thus maintaining a valid distinction between genera respectively endemic to Europe or the USA. The genus '*Redondasaurus*' forms a clade within *Machaeroprosopus* in both trees, but in the D tree the specimen NMMNHS P-31094, (diagnosed as '*Redondasaurus*' on the basis of possessing many of the group's synapomorphies) is also a member of the clade, as expected, rather than forming a sister-group with *Machaeroprosopus lottorum* as in the DC tree. The position of *Machaeroprosopus lottorum* in the D tree, in a sister-group with *Machaeroprosopus pristinus*, supports the validity of the latter taxon, suggesting it is not a sexual dimorph of *Machaeroprosopus buceros*. For these reasons, we tentatively suggest the use of the discrete-character tree for further analyses.

## INSTITUTIONAL ABBREVIATIONS

| | |
|---|---|
| **AMNH** | American Museum of Natural History, New York, USA |
| **GPIT** | Institut für Geologie und Paläontologie Tübingen, Tübingen, Germany |
| **KU** | University of Kansas, Lawrence, USA |
| **MB** | Museum für Naturkunde, Berlin, Germany |
| **MNHN** | Muséum National d'Histoire Naturelle, Paris, France |
| **MU** | University of Missouri, Columbia, Missouri, USA |
| **NHMW** | Naturhistorisches Museum Wien, Vienna, Austria |
| **NMMNHS** | New Mexico Museum of Natural History and Science, Albuquerque, USA |
| **OMNH** | Oklahoma Museum of Natural History, Norman, USA |
| **PEFO** | Petrified Forest National Park, Arizona, USA |
| **SMNS** | Staatliches Museum für Naturkunde Stuttgart, Stuttgart, Germany |
| **TMM** | Texas Memorial Museum, Austin, USA |
| **TTU-P** | Museum of the University of Texas Tech, Lubbock, USA |
| **UCMP** | University of California Museum of Paleontology, Berkeley, USA |
| **UMMP** | University of Michigan Museum of Paleontology, Ann Arbor, USA |
| **USNM** | National Museum of Natural History, Washington D.C., USA |
| **UW** | University of Wisconsin Geological Museum, Madison, USA |
| **YPM** | Yale Peabody Museum, New Haven, USA. |

## ACKNOWLEDGEMENTS

We thank the following people for access to, and assistance with, specimens in their care: Carl Mehling (AMNH), Oliver Rauhut (BSPG), William Simpson (FMNH), Philipe Havlik and Davit Vasilyan (GPIT), Daniela Schwarz and Thomas Schossleitner (MB), Ronan Allain (MNHN), Casey Holliday and James Schiffbauer (MU), Lorna Steel (NHMUK), Ursula Göhlich (NHMW), Eckhard Mönnig (NMC), Amanda Cantrell (NMMNHS), William Parker and Matthew Smith (PEFO), Rainer Schoch (SMNS), Kenneth Bader and Matthew Brown (TMM), Bill Mueller (TTU-P), Patricia Holroyd (UCMP), Michael Brett-Surman (USNM), Daniel Brinkman (YPM), and Robert Bronowicz and Tomasz Sulej (ZPAL). We are also grateful to Pablo Goloboff for TNT script suggestions and bug fixes, and to Octavio Mateus for useful discussion of phylogenetic characters. Finally, we thank Mark Young, Caroline Parins-Fukuchi, William Parker & Axel Hungerbühler for constructive comments that greatly improved the manuscript.

### Funding

Andrew Jones was supported by a NERC Training Grant (grant number NE|L002493|1). Richard Butler was supported by a Marie Curie Career Integration Grant (grant number 630123). The funders had no role in study design, data collection and analysis, decision to publish, or preparation of the manuscript.

## Grant Disclosures

The following grant information was disclosed by the authors:

A NERC Training: NE|L002493|1.

A Marie Curie Career Integration: 630123.

## Competing Interests

The authors declare that they have no competing interests.

## Author Contributions

- Andrew S. Jones conceived and designed the experiments, performed the experiments, analysed the data, prepared figures and/or tables, authored or reviewed drafts of the paper, approved the final draft.
- Richard J. Butler conceived and designed the experiments, authored or reviewed drafts of the paper, approved the final draft.

## Data Availability

The raw data are provided in the Supplemental Files.

## Supplemental Information

Supplemental information for this article can be found online at http://dx.doi.org/10.7717/peerj.5901#supplemental-information.

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
