# Peer review of "A new phylogenetic analysis of Phytosauria (Archosauria: Pseudosuchia) with the application of continuous and geometric morphometric character coding"

_PeerJ, doi:10.7717/peerj.5901_

## Round 0.1 · original submission · Minor Revisions

Dear authors,

I have accepted the decision of 'minor revisions' from the three reviewers.

Thank you for submitting your manuscript to PeerJ and I look forward to receiving your revision.

·

Basic reporting

It takes the reader a bit of work to figure out the specific compositions of the datasets under comparison. It might be helpful to dedicate a section that explains clearly the ways that continuous, discrete, and geometric data are combined and compared. I realize that the information is all in the paper-- this would just help when readers are trying to track down what is meant by a 'DC' vs 'DM' comparison, for instance.

As the authors are probably aware, there exists a burgeoning literature dealing with more statistically modern approaches to the analysis of morphological phylogeneties covering all of the data sources explored in this manuscript (discrete, continuous/geometric, and stratigraphic). In light of this, the exclusive focus on parsimony methods in this paper strikes as odd. To be clear, I wouldn't suggest that the authors re-analyze their data under newer methods, but the lack of awareness in the manuscript of these developments is striking. In particular, I was struck by the authors' neglect of recent development in probabilistic phylogenetic analyses of discrete (Wright and Hillis 2014, PLoS One), continuous (Parins-Fukuchi 2018, Syst. Biol.), geometric morphometric (Parins-Fukuchi 2018, BioRxiv), and stratigraphic (Stadler et al. 2017, Arxiv) data in their otherwise extremely thorough review of existing methods. As a full disclaimer, I don't generally feel good about recommending citation of my own work, but I do honestly feel that a brief review of all four of these previous studies would be appropriate given the depth with which the authors describe relevant parsimony approaches. While the approaches used by the authors in their study are sufficient for their own questions, a broader awareness of modern methods in the manuscript would be helpful in framing the relevance of this study as these data types are applied in future studies that use updated approaches.

Here are the citations:

Stadler, Tanja, et al. "The fossilized birth-death model for the analysis of stratigraphic range data under different speciation concepts." arXiv preprint arXiv:1706.10106 (2017).

Parins-Fukuchi, Caroline. "Use of Continuous Traits Can Improve Morphological Phylogenetics." Systematic biology (2017).

Parins-Fukuchi, Caroline. "Bayesian and likelihood placement of fossils on phylogenies from quantitative morphometric data." bioRxiv (2018): 275446.

Wright, April M., and David M. Hillis. "Bayesian analysis using a simple likelihood model outperforms parsimony for estimation of phylogeny from discrete morphological data." PLoS One 9.10 (2014): e109210.

Experimental design

This study is very thorough. The authors have clearly thought and planned very carefully, and are very clear in describing each step. There are a few aspects that confuse me, however. Most notably, I'm left feeling slightly puzzled regarding the choice of methods and the general statistical mindset employed by the authors. Although thorough, the deep exploration of parsimony methods, and heavy reliance on frequentist statistics can feel arcane. For instance, the sentence in line 854: "However, the agreement subtrees suggest that the amount of overlap between these hypotheses is still greater than would be expected to occur by chance" comes across as old fashioned-- do we ever expect phylogenetic hypotheses from different datasets to overlap only due to random chance? I might suggest reconsidering the reliance on this type of reasoning.

As I note in the general comments, I wouldn't suggest that the authors need to reanalyze their data using more modern methods-- the thoroughness of their parsimony analyses result in more than enough clarity to address their questions, which may or may not even be benefited by alternative methods. But some tweaking of the presentation of the logic in the manuscript might make things feel less esoteric and pigeonholed in the fading cladistic and frequentist mindsets.

Aside from the comments above, I do have two specific questions regarding the methods used:

1) I see no mention of the bootstrap as a measure of node support. I would acknowledge the bootstrap to be flawed in some respects, but it is the standard in phylogenetics, and readily interpreted by biologists of most backgrounds. The authors apparently prefer to use Bremer supports, but since this manuscript will be published in an outlet intended for a general biological audience rather than Cladistics, I think it would be really helpful to switch the support measures to bootstraps, or at least describe Bremer support clearly, along with a justification for its' use over the bootstrap.

2) I enjoyed the authors' examination of the stratigraphic fit across topologies. But since the stratigraphic fit is used by the authors as an independent comparison of the reliability of different topologies, I am left wondering why they didn't perform any simple stratocladistic analyses? They hint at a dissection of the overall stratigraphic correlation for each topology, and potentially poor support yielded by particular subclades. In lieu of a full stratocladistic optimization (which I realize is challenging given the present state of available tools), it could be really interesting to compare patterns in stratigraphic parsimony debt incurred by each of the conflicting topologies. This should be achievable given available tools, and would mesh well with the cladistic focus of the manuscript.

Despite the length of these comments, I do not intend them as barriers to the publication of this study. More generally, I found the goals and design of the study to be very clearly stated, and thoroughly and competently addressed by the analyses. I feel that the issues that I outline above could improve the reception and impact of the study among a more general audience, but at its core, this study is very well done, and most certainly worthy of publication and citation.

Validity of the findings

As I write above, this manuscript leaves me feeling confident in the the validity and thoroughness of the authors' results. My only concern in this area stems from the fact that the authors pour so much information in the results and discussion that the conclusion gets a bit buried. I would appreciate a less equivocating suggestion by the authors over their preferred resolution of the groups-- I know that they prefer the 'D' and 'DC' trees, but which aspects are most clearly resolved in these topologies? And how does the confidence that they recover in the relationships improve over previous studies, specifically? Finally, are the GM trees completely unsalvageable? Or can other insights potentially be gleaned from their use?

·

Basic reporting

Manuscript is well organized and clearly written. Hardly any typographic or grammatical errors. A very clean manuscript.

Experimental design

The methods are well described in this manuscript.

Validity of the findings

I have made comments on the annotated versions of the main text and the main supplement text. I think there needs to be a little more discussion where the results affect things such as the use of isolated squamosals for biostratigraphy and the unstable taxonomy of the Leptosuchus/Smilosuchus group.

Additional comments

Very nice study although pretty lengthy which makes it hard to follow in places. Make sure you emphasize which treatment is best recommended for moving forward in future analyses of the Phytosauria. This is done at the very end of the conclusions, but it would be beneficial to have it emphasized in the abstract and maybe at the end of the Introduction section as well.

I've also corrected a bit of the stratigraphic and locality data in the supplemental files. Also note that the catalogue number for RLG 11/07-3 is PEFO 38340.

·

Basic reporting

Jones & Butler present a remarkable, solid, and well-written study on the phylogenetic relationships among phytosaurs, the most common group of reptiles in Late Triassic deposits. The study is an original piece of work that fits well in the scope of PeerJ and adheres to the structure of the journal.

I fully endorse publication in PeerJ, but I recommend three items of improvement below to enhance the clarity of the study and to complement its goals.

Experimental design

There is a number of points in which this study stands out positively compared to previous analyses of phytosaurs (and, as a matter of fact, in comparison with many other cladistics studies of extinct archosauriforms):
The analysis is the most comprehensive with respect to taxon sampling to date.
I am particular impressed by the revised and thoroughly justified combined database presented in the supplemental material. It is safe to predict that the revised data matrix will form the base upon which researcher will rely and expand for future phylogenetic analyses of the group for years to come.
The study introduces novel types of data (continuous, geometric morphometric – the latter to my knowledge never employed in phylogenetic analyses of extinct vertebrates), and the significance and utility for phylogenetic analyses.

The authors use rigorously a number of tests (e.g., decay indices, stratigraphic congruence, implied weighting) to evaluate their findings and the validity of phylogenetic hypotheses. Employing such a variety of independent tests is extraordinary and certainly not standard procedure in our field.

The methodical part is exemplary, and in combination with the supplemental data, the analysis is fully reproducible.

The purpose of the study is fully explained in the introductory section.

Validity of the findings

I suggest three points in which the manuscript can and should be improved:

1.) In the sections “High-level taxonomy” and “low-level taxonomy”, you present an exhaustive discussion and evaluation of previous hypotheses on phytosaur relationships in the light of their data. The conclusions are well-reasoned and supported by character evaluations and various tree statistics.
However, although the conclusions on the taxonomy or the phylogenetic position of single taxa are clearly stated at the end of the paragraphs of each topic discussed, the main conclusions are difficult to retrieve for the reader without filing through half of the paper. I feel the reader would benefit from a paragraph in the “conclusions” section summarizing the more important findings, in particular those at variance with the current consensus (like the position of Protome and Rutiodon, non-monophyly of Leptosuchus and Smilosuchus, Nicrosaurus as non-mystriosuchine leptosuchomorph. or the synonymy of Redondasaurus and Machaeroprosopus), including those conclusions which are outlined in the abstract.

2.) You include a number of specimens as separate OTUs in your analysis, with the explicit intention to test their phylogenetic affinities (supplemental material, lines 1094, 1118, 1131, 1165, 1178). However, for a number of these specimens the results of the tests are not even mentioned in the text, despite the potential significance of the individuals. These include MB.R. 2747 – the only substantial remains of a Rhaetian phytosaur from Europe, PEFO 34852 – the oldest phytosaur showing three sacral vertebrae, and NMMNHS-P 4256 and P 31094 – potential Redondasaurus that if substantiated would extend the range of the taxon and question its usefulness as index fossil for the Apachean biozone (Martz & Parker 2017).
I recommend a brief discussion of these specimens in appropriate positions within the subsection “Low-level taxonomy”

3.) Sections “Previously Referred Specimens” throughout the supplemental material:

I got confused with these sections. At first I assumed that you agree with these referrals, and the sections represent a ’list of specimens’ or what you include in the hypodigm of that species, until I realized that this is actually rarely the case. However, as a reader I am much more interested to know which specimens you include in that taxon (aka, which specimens match your modified diagnosis) rather than which specimens have been erroneously or without substantiation referred to the taxon in the past. The latter should be included and discussed in a comprehensive systematic review, but I do not think it is absolutely necessary in the database for a phylogenetic analysis. As such, the list of previously referred specimens is also tor a number of taxa far from incomplete (e.g., Spielmann & Lucas 2012 referred a multitude of specimens to Redondasaurus gregorii).
I suggest to change these paragraphs to ‘Referred Specimens’ and include previous assignments that you find correct (I do appreciate that you give credit to previous workers here) and those specimens you refer to that taxon for the first time. Thus, this paragraph represents your hypothesis of the hypodigm of the taxon (which need not to be comprehensive, and of course includes also the ‘holotype’ and ‘specimen(s) used for scoring’, both of which should remain separate sections). Any information you deem important (for example, Camp’s concepts of adamanensis, lithodendrorum, and gregorii, or the exclusion of AMNH 10644 from Mystriosuchus planirostris) could go in the ‘comment’ sections.

I include two sections with some minor points or additional comments for the main text and the supplementary file; the authors are welcome to include the information if so desired:

1. Manuscript: Detailed comments

Line 284: Nicrosaurus kapffi

Lines 1192-1193: “However, the fact that Rutiodon consistently forms the sister group to Angistorhinus makes the decision of whether or not to synonymize the genera entirely arbitrary” – In addition, your analysis supports an important implication from Hungerbuehler & Sues’ hypothesis with respect to character evolution: The depression of the supratemporal fenestrae, for long regarded as the hallmark of ‘derived’ phytosaurs (Rutiodon + Leptosuchomorpha) must have actually evolved independently twice in phytosaurs.

Line 1570: “Hungerbühler et al. (2013) were unable to recover ‘Redondasaurus’ gregorii and ‘Redondasaurus’ bermani as a monophyletic group” – This is most likely because the taxa were scored after the holotypes only, which resulted in a high amount of missing data for several species included.

Lines 1617-1629 “Thickened orbital margin” – Spielmann & Lucas 2012 do not describe or expand on this character, but I assumed from their fig. 49 the character refers to the blunt (rather than sharp) and particularly the elevatation of the rims of the circumorbital bones (rather than the postorbital only) in at least some of the specimens figured. This interpretation would explain the comparison with Coburgosuchus, in which at least the dorsal rim is elevated (but not thickened).

Lines 1663-1664 “…these taxa are always monophyletic to the exclusion of species of Machaeroprosopus.” Correct, but what is with NMMNH 31094 which has been referred to Redondasaurus, and shares at least three potential synapomorphies with species of that genus (supplemental data, lines 1140-1142)

2. Supplemental Material: Detailed comments

Line 152: ‘Paleorhinus’. magnoculus

Lines 588-590: Nicrosaurus kapffi has, to my knowledge, no formally designated lectotype (nor do I think there is a need for a lectotype designation) – these specimens form the syntype series.

Line 647: “Machaeroprosopus” zunii holotype. In support of the authors’ conclusion about the validity of the species, I might add from personal inspection that the braincase almost certainly comes from a smaller individual than the postcranium, and it appears to be less derived than a basal leptosuchomorph phytosaur – thus even the holotype seems to be a chimaera.

Line 686-687: From Stocker’s description and figures (I do not know the specimen first-hand), these prongs are in the same position as two exoccipital/supraoccipital projections identified as proatlantes facets by Hungerbuehler et al. (2013: 298, fig. 24) for Machaeroprosopus lottorum. Provided these are indeed homologous structures, such prongs are present in numerous skulls referable to Machaeroprosopus and Redondasaurus (and perhaps more taxa, I have not conducted a comprehensive survey). I doubt that the exoccipital prongs are autapomorphic for Protome.

Line 713-714; “found not to be the type species of Machaeroprosopus.” - Reference needed

Line 782: As you include TTU-P 11423, the taxon also occurs in Texas.

Line 795-796: ‘tubular’, literally tube-like, implies a slender, hollow and round snout - I would not describe a low sharp-crested/triangular snout as tubular

Line 849-850: Omit the wording ‘and tubular’ in character 3.), otherwise it duplicates character 4.). As mentioned above, the word tubular aka “tube-like” implies a round, or here semi-circular, cross-section.

Line 1015; “2074.000, 2149.002, 2149.003, 2150.000” are GPIT specimens

Line 1098: “Chinle Group” – Dockum Group, to be consistent

Line 1723: “sausage-shaped” - narrow and curved? Please explain

Line 1738: “sub-temporal shelf” – please explain/describe this structure. Is this the dorsally facing facet of the jugal in character 83[1]?

Line 1750: How is this ratio calculated?

Line 1759: How is this ratio calculated?

Line 1774: Extends as ridge forward for… ? Would this be what has been called a narial crest?

Line 1776: …..of the quadrate? If so, the character makes no sense to me – the condylar area of the quadrate has two condyles, and I do not understand the character states. Please explain.

Line 1780: length of symphysis to post-symphysial mandibular length, meaning [0] short [1] long? Please clarify

---

## Round 0.2 · Minor Revisions

Dear authors,

I am very sorry for the delay in getting this back to you. I have been waiting on a review coming in, however, I have decided to proceed without it.

As you can see, there are some remaining minor comments from Reviewer 2. I look forward to receiving your revised manuscript.

·

Basic reporting

no comment

Experimental design

no comment

Validity of the findings

no comment

Additional comments

This resubmission looks great. The authors adequately addressed all of my concerns over what was already a very strong paper in the revision and rebuttal documents. I am looking forward to seeing the final published version!

·

Basic reporting

This manuscript is very clean, without noticeable typographic errors. In my first review I was critical of the presence of a section in the Introduction discussing "Weaknesses" of previous work. I am happy to see that this section has been removed and for the reasons listed by the authors in their rebuttal letter. The literature review is exhaustive and thorough and the article is well-organized. The results section is pretty dense as the section requires explaining and comparing four different trees with each other and past work. Breaking up some of the longer paragraphs and citing figures more often in the sections may help with this. In the tree choice section please iterate what the tree abbreviations are one more time for readers who struggle with remembering all of the abbreviations.

The Discussion section is also very dense with long paragraphs and sentences. For example lines 1094 to 1099 are all a single sentence. Where possible breaking apart long paragraphs will help the reader navigate through this section.

Experimental design

This is a very large exercise in phylogenetic database exploration using a group with a notoriously tortuous taxonomic history. In my previous review I criticized the manuscript for not suggesting which of the four analyses should be used by phytosaur workers to move forward. This is especially important as the authors are suggesting revisions to phylogenetic definitions of various clades. I'm glad to see the authors have added a suggestion of using the DC tree for future analyses, but also note that this recommendation is tentative. Just be aware that this makes your work difficult to use in future analyses, and instead relegates it to dataset exploration and a critique of past methodologies and scorings. As also stated because it utilizes continuous characters the DC tree is only useable with TNT and not other software packages such as PAUP*. It will be up to future workers to provide a final evaluation of the utility of the work presented here.

Validity of the findings

The use of exploratory methods creates a situation where a lot of data are generated that are not entirely consistent with each other. The conclusions are well formulated and stated, but again I hope because of the complexity of the methods and discussion that the results are useable for future analyses, especially since new phylogenetic definitions are created based on these various results.This is hard to evaluate without actually trying to use the information in future work. Overall, however, I feel that the questions asked and answered by this study are sound and useful. I believe the authors have accomplished their stated goals and will generate further discussion of phytosaur taxonomy and phylogenetic relationships.

The Supplemental materials are improved.

Additional comments

I recommend the following changes:

All uses of e.g. should be followed with a comma. This is standard for non-U.K. based journals.

Watch the prefixes for specimen numbers. I noted some differences in the conventions used. I specifically saw differences in ANHM specimens.

Line 1278: If you aren't sure that the observed character state in PEFO 31218 has not been affected by preservation, you should score it as unknown.

Line 1443: Note that the holotype specimen of M. mccauleyi is missing the end of the snout.

Any database exploration study is going to be critical of past authors. Please review and ensure that any criticisms are fair and necessary and toned down to not cause offense with any colleagues. This version is much better that the previous regarding this, but it does not hurt to review one more time to ensure you aren't offending any colleagues with your treatment of their work. Everyone expects to be criticized, and its acceptable if done in a fair manner.

---

## Round 0.3 · accepted · Accept

Dear authors,

Many thanks for your revised manuscript. After reading it, I have accepted it for publication in PeerJ.

Once again, thank you for submitting your manuscript to PeerJ and I hope you will use us again as your publication venue.

If we need to clarify any details required to move the manuscript forward, then our production staff will get in touch with you. Otherwise, a proof will be forthcoming shortly for your review.

Congratulations and thank you for your submission.

#